# Uncovering the signaling landscape controlling breast cancer cell migration identifies novel metastasis driver genes

Esmee Koedoot[1,4], Michiel Fokkelman [1,4], Vasiliki-Maria Rogkoti[1,4], Marcel Smid[2], Iris van de Sandt[1], Hans de Bont[1], Chantal Pont[1], Janna E. Klip[1], Steven Wink [1], Mieke A. Timmermans[2], Erik A.C. Wiemer[2], Peter Stoilov[3], John A. Foekens[2], Sylvia E. Le Dévédec [1], John W.M. Martens [2] & Bob van de Water[1]

Ttriple-negative breast cancer (TNBC) is an aggressive and highly metastatic breast cancer subtype. Enhanced TNBC cell motility is a prerequisite of TNBC cell dissemination. Here, we apply an imaging-based RNAi phenotypic cell migration screen using two highly motile TNBC cell lines (Hs578T and MDA-MB-231) to provide a repository of signaling determinants that functionally drive TNBC cell motility. We have screened ~4,200 target genes individually and discovered 133 and 113 migratory modulators of Hs578T and MDA-MB-231, respectively, which are linked to signaling networks predictive for breast cancer progression. The splicing factors *PRPF4B* and *BUD31* and the transcription factor BPTF are essential for cancer cell migration, amplified in human primary breast tumors and associated with metastasis-free survival. Depletion of *PRPF4B*, *BUD31* and *BPTF* causes primarily down regulation of genes involved in focal adhesion and ECM-interaction pathways. *PRPF4B* is essential for TNBC metastasis formation in vivo, making *PRPF4B* a candidate for further drug development.

[1] Division of Drug Discovery and Safety, LACDR, Leiden University, Einsteinweg 55, Leiden 2333 CC, Netherlands. [2] Department of Medical Oncology and Cancer Genomics Netherlands, Erasmus MC Cancer Institute, Erasmus University Medical Center, Rotterdam 3008 AE, Netherlands. [3] Department of Biochemistry and Cancer Institute, Robert C. Byrd Health Sciences Center, West Virginia University, Morgantown, WV 26506, USA. [4]These authors contributed equally: Esmee Koedoot, Michiel Fokkelman, Vasiliki-Maria Rogkoti. Correspondence and requests for materials should be addressed to v.d.W. (email: b.water@lacdr.leidenuniv.nl)

Breast cancer is the most prevalent cancer in women. The triple-negative breast cancer (TNBC) subtype which lacks expression of the estrogen, progesterone and HER2 accounts for ~15% of breast cancer. TNBC is the most aggressive BC subtype with an overall 5 year relapse of 40% due to primary and secondary metastatic spread[1,2]. Transcriptomics has classified four different molecular subtypes of TNBC genes[3] while proteomics studies revealed markers of disease progression[4]. More recently, genome-wide sequencing of large numbers of human breast cancers have defined the somatically acquired genetic variation in breast cancer including TNBC[5–7] as well as the genetic evolutionary programs associated with local and distant metastatic spread[8,9]. Although the roles of individual genes in breast cancer metastasis[10–13] have been described, and few pooled shRNA screens have identified regulators of cancer metastasis[14,15] a systematic analysis of the functional consequences of genetic aberrations in TNBC for progression towards metastatic disease is still lacking.

The most critical cell biological hallmark of TNBC metastasis is the increased plasticity of TNBC cells[16,17]. Many TNBC cell lines have adapted towards a mesenchymal phenotype and demonstrate a high migratory behavior in association with an increased metastatic spread[11,18,19]. Cell migration is involved in various steps of the metastatic cascade, including local invasion, intravasation, extravasation, and colonization of secondary sites[20]. Understanding the fundamentals of TNBC cell migration is critical for our comprehension of the development of metastatic disease. The control of cell migration is complex and involves components of the cell adhesion machinery as well as modulators of the actin cytoskeleton that coordinate cellular motility behavior[21,22]. The functionality of these machineries is controlled by signaling pathways and associated transcriptional programs that coordinate the expression of these cell adhesion and cytoskeletal modulators as well as their post-translational modification and activity[23]. While the signaling pathways that control TNBC proliferation have been uncovered using genome-wide cell survival screens[24,25] the role of individual cell signaling components that define TNBC motility behavior is less clear.

Here we systematically unraveled the global cell signaling landscape that functionally control TNBC motility behavior through a phenotypic imaging-based RNAi-screen to identify genes involved in the regulation of different migratory phenotypes. We discover genes including several transcriptional modulators, e.g., *PRPF4B*, *BPTF*, and *BUD31*, that define TNBC migratory programs and metastasis formation, which are associated with poor clinical outcome of breast cancer and share signaling networks underlying prognostic gene signatures for primary breast cancer.

## Results

**A high-throughput RNAi screen for TNBC cell migration.** We selected two of the most highly motile TNBC cell lines Hs578T and MDA-MB-231 for microscopy-based RNAi screening using the PhagoKinetic Track (PKT) assay (Fig. 1a, Supplementary Fig. 1, Supplementary movies 1 and 2; and described previously[26]). Briefly, cells were seeded on fibronectin containing a thin layer of discrete latex beads that are phagocytosed during cell migration, leaving behind individual migratory tracks. Track-related image-analysis included quantification of net area, major/minor axis, axial ratio, and roughness, defining the tumor cell migration behavior. In contrast to live cell migration assays, this assay can be applied in a high-throughput manner for genetic screening of TNBC cell migration[26,27]. We focused our screening effort on the complete set of cell signaling components, covering all kinases, phosphatases, (de)ubiquitinases, transcription factors, G-protein coupled receptors, epigenetic regulators, and cell adhesion-related molecules (4198 individual target genes in total,

SMARTpool siRNAs (pool of four single siRNAs per target)). Quantitative output data were normalized (robust Z-score) to mock transfected control cells. High and low Z-scores of individual parameters already showed the effect of siRNA knockdown on cell motility, i.e., low net area or low axial ratio suggests inhibition of cell migration whereas high axial ratio and high major axis indicated enhanced motility (Fig. 1c, d). Even though the quantification provided eight parameters, all the different migratory phenotypes were not fully represented by single parameters. Therefore, migratory tracks were manually curated and assigned to specific phenotypes by setting thresholds on Z-score for the most dominant parameters of each phenotype after which primary hits were determined based on these Z-scores (described in the methods section). The migratory phenotypes were visualized by principal component analysis (PCA)-based clustering (Fig. 1e, f). For all phenotypes together, we defined 2807 hits in total: 1501 primary hits for Hs578T and 1306 for MDA-MB-231. Cytoskeletal genes, which are known to be important in cell migration and might provide regulatory feedback loops, were not enriched in these primary hit lists (Supplementary Fig. 2). Importantly, there was no correlation between proliferation (number of tracks) and any of the phenotypic parameters, suggesting that hit selection was mainly based on effects on migration (Supplementary Fig. 3). However, we cannot exclude that effects on cell proliferation could contribute to the inhibition of cell migration for some candidates, in particular for those that showed a decrease in track number. We identified 129 overlapping hits showing similar effects on cell migration upon knockdown in both cell lines, suggesting these were bona-fide cell line-independent drivers of tumor cell migration. Hence we selected these overlapping hits for validation by single siRNA sequences. Additionally, to obtain a larger coverage of genes regulating cell migration that would uncover a more cell type specific migratory behavior, we also selected the top 153 hits in each cell line for validation. Only genes that have been defined as druggable were validated, resulting in validation of 451 unique targets (129 overlapping hits, 153 Hs578T unique hits and 153 MDA-MB-231 unique hits) (Supplementary Fig. 1 and Supplementary Data 1).

To validate the primary hits, we repeated the PKT screen assays with both SMARTpool and four single siRNA sequences (Fig. 2a and Supplementary Fig. 1). In total, 217 (77%) hits were validated in the Hs578T and 160 (57%) in the MDA-MB-231 (significant effects for at least two singles and SMARTpool, for Hs578T see Fig. 2b; for MDA-MB-231 see Supplementary Fig. 4; all validated genes are in Supplementary Data 2; reproducibility is shown in Supplementary Fig. 5 and Supplementary Fig. 6). The majority of validated hits was found in the phenotypic classes of reduced cell migration (Fig. 2c); 65 validated candidate genes showed inhibition of cell migration in both cell lines (Fig. 2d). This relatively low overlap of candidates cannot be attributed to cell line specific mutations, copy number alterations or differences in expression levels of the candidates (Supplementary Fig. 7, information about mutation type in Supplementary Data 3) or genes in pathways enriched for cell line specific candidates (Supplementary Fig. 8). Annotation of protein classes for each set of validated hits (Hs578T, MDA-MB-231, and overlap) showed that most of the hits were transcription factors (Fig. 2e i) also after correction for library size (Fig. 2e ii), suggesting that transcriptional regulated gene networks are critical drivers of TNBC cell migration behavior.

**Transcriptional determinants are drivers of BC migration.** Next we evaluated the effect of all validated hits on cell migration using a live microscopy cell migration assay with GFP-expressing Hs578T and MDA-MB-231 cells (Supplementary Data 4 and 5).

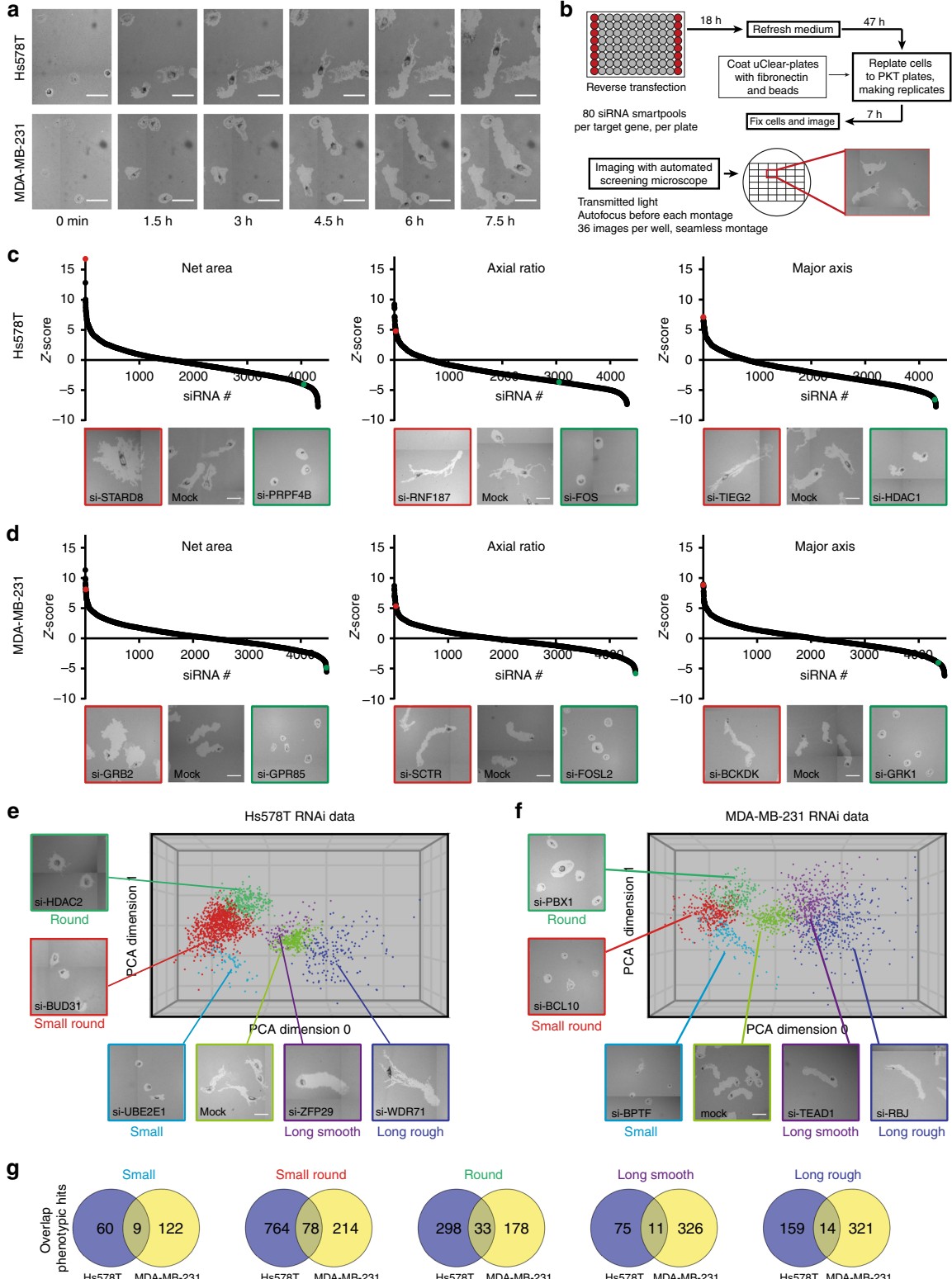

**Fig. 1** A phenotypic, imaging-based, RNAi screen identifies regulators of tumor cell migration. **a** Live cell imaging of Hs578T and MDA-MB-231 phagokinetic tracks. Scale bar is 100 μm. **b** Schematic representation of PKT screen. Transfection was performed in 96-well plates and controls were included in each plate. After 65 h, transfected cells were washed, trypsinized, diluted and seeded in PKT assay plates. Plates were fixed after 7 h of cell migration and whole-well montages were acquired using transmitted light microscopy. For each siRNA knockdown, a robust Z-score was calculated for each PKT parameter. All screening experiments were performed in technical and biological duplicates. **c** The three most dominant quantitative PKT parameters are shown for Hs578T, and **d** MDA-MB-231. Representative images of migratory tracks for genes with strong effect are shown below each graph and highlighted for enhancement (red) and inhibition (green). **e** Principal component analysis of migratory phenotypes in Hs578T, and **f** MDA-MB-231. Migratory phenotypes were identified manually and corresponding Z-scores were determined (see the methods section for more details). Only hits in each phenotypic class and mock control are plotted, and representative images of each phenotype are shown. **g** Overlap of hits in each phenotypic class in both Hs578T and MDA-MB-231

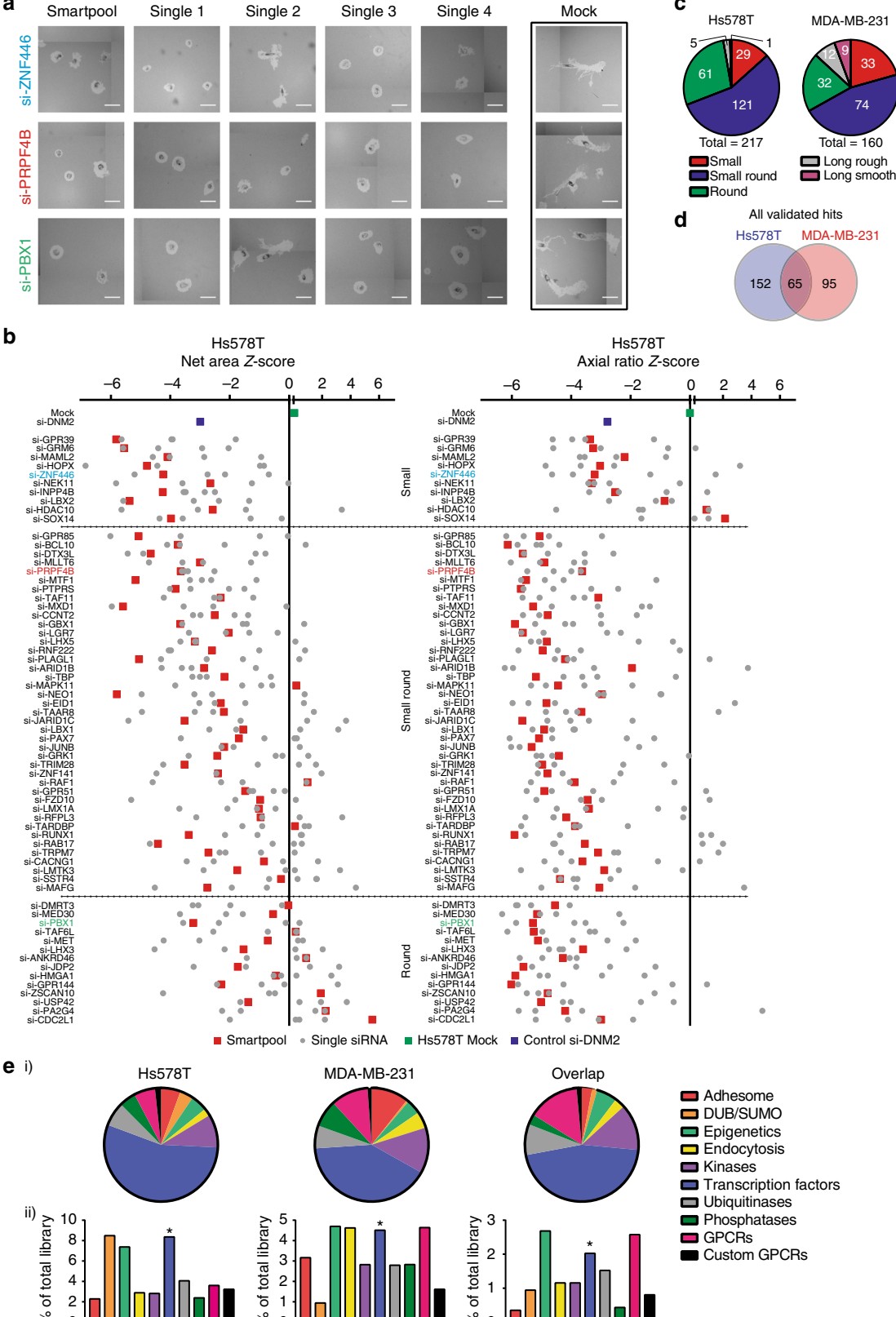

**Fig. 2** Candidate migratory gene validation by deconvolution PKT screen. **a** Representative images of valid candidate genes using four individual single siRNAs in three phenotypic classes are shown. Scale bar is 100 μm. **b** A total of 282 selected genes were tested in a deconvolution PKT screen with four single siRNAs per gene. SMARTpool and single siRNA Z-scores of Net Area and Axial Ratio are shown for the overlap hits that were validated in Hs578T. Average of biological and technical duplicates. **c** Distribution of validated candidate genes in the five phenotypic classes. **d** Overlap of validated hits between Hs578T and MDA-MB-231. **e** Distribution of the validated genes involved in tumor cell migration in Hs578T, MDA-MB-231 over the different libraries based on validated hits effective in both cell lines. Statistical significance was determined using a Fisher's exact test. *$p < 0.05$

Live cell imaging with Hs578T cells confirmed 133 of the 217 hits to inhibit cell migration. Similarly, for the MDA-MB-231 cells, 113 candidate genes (out of 160 validated hits) were confirmed to regulate cell migration. Upon knockdown, 31 PKT overlap candidates inhibited cell migration in this assay in both cell lines (Fig. 3a, b and Supplementary Movies 3–14), including various transcriptional and post-transcriptional regulators such as *RUNX1, MTF1, PAX7, ZNF141, SOX14, MXD1, ZNF446, TARDBP, TBX5, BPTF, TCF12, TCERG1, ZDHHC13, BRF1*, some of which are directly involved with splicing (*BUD31* and *PRPF4B*) or histone modification (*HDAC2* and *HDAC10*). For cell line specific validated hits, we filtered candidate genes for which the expression was associated with clinical breast cancer metastasis-free survival (MFS) in a patient dataset (the Public-344 cohort,

GSE5237, and GSE2034, Supplementary Data 6). Many of the hits associated with poor outcome inhibited cell migration in both Hs578T and MDA-MB-231 (Fig. 3a, b, see Supplementary Data 4 and 5 for all candidate genes). Combined with the overlap candidates this resulted in 43 genes that were common denominators of cell migration. Single cell migratory trajectories were plotted for genes affecting cell migration in both cell lines (Fig. 3c). Furthermore, we confirmed the general role of our candidates in random cancer cell migration by validation of several main candidates by inducible CRISPR-Cas9 knockout in a live cell migration assay (Supplementary Fig. 9), siRNA knockdown followed by live cell migration assays in two additional TNBC cell lines (Supplementary Fig. 10) and siRNA knockdown followed by a traditional scratch assay (Supplementary Fig. 11A–D), all 3 days

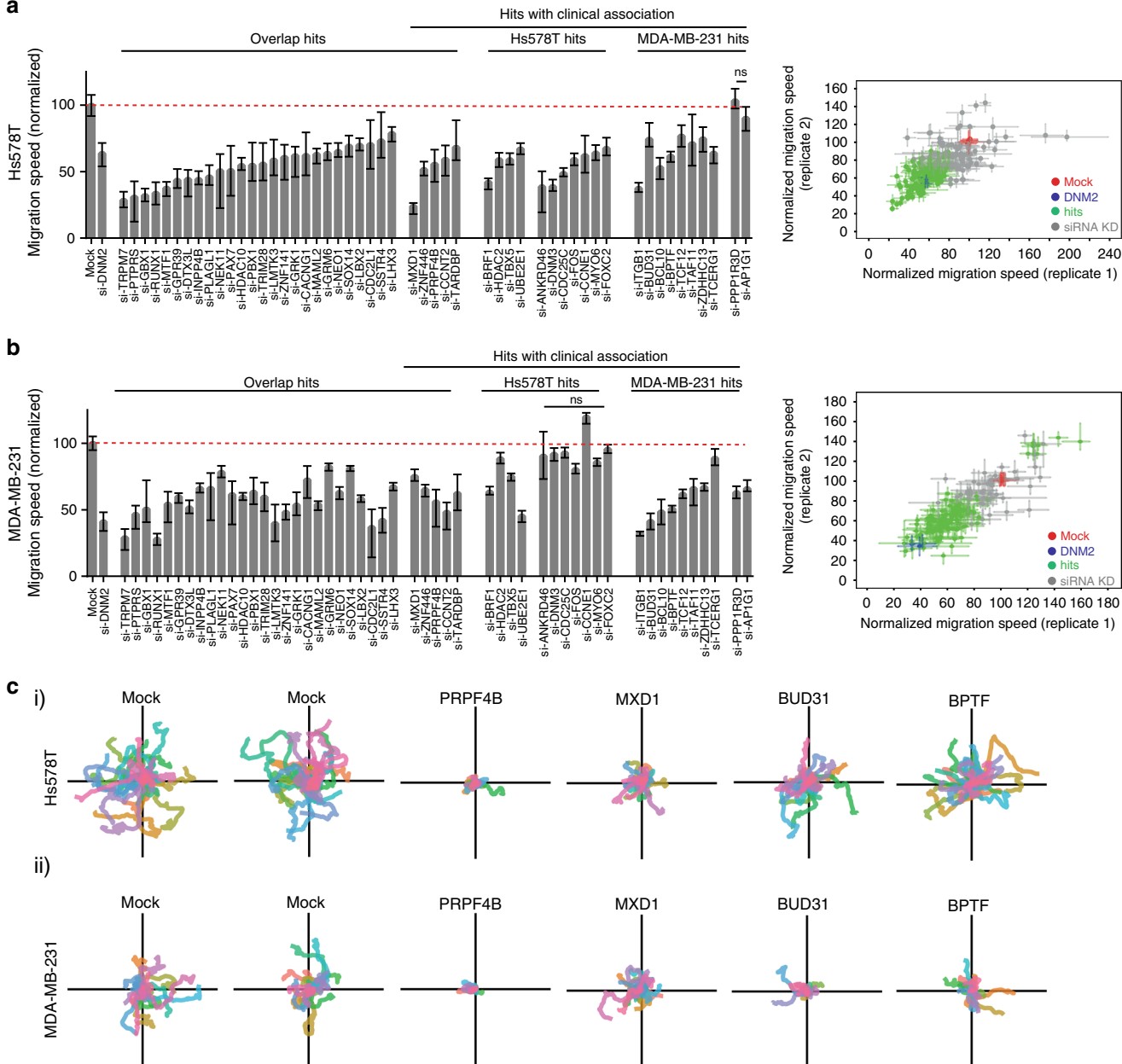

**Fig. 3** Candidate genes directly affect TNBC cell migratory behavior. **a** Quantification of single cell migration speed of Hs578T-GFP cells after knockdown of validated hits. Hs578T-GFP cells were transfected with siRNAs and cell migration was assessed by live microscopy. Hits that showed significant and consistent effects in both replicates were considered as candidate genes (right panel). Median ± 95% confidence interval is shown and cell populations were compared by Kruskal–Wallis test with Dunn's post correction test. **b** Same as in **a**, with MDA-MB-231-GFP cells. **c** Single cell trajectories of cell migration upon knockdown of *PRPF4B, MXD1, BUD31*, or *BPTF* in (i) Hs578T and (ii) MDA-MB-231 cell lines

after knockdown. All of our tested candidates were also affecting FBS-directed cell migration in MDA-MB-231 3 days after knockdown, but not per se in Hs578T (Supplementary Fig. 11E–F). This suggests that effects on random cell migration cannot always be directly extrapolated to directed cell migration, especially for the Hs578T cell line. The latter could be caused by the use of fibronectin-coated plates for live cell imaging compared to a polystyrene membrane without extracellular matrix (ECM) coating for directed cell migration or differences in the duration of the migration assays (from 7 h for the PKT assay until 22 h for the directed cell migration assays). However, altogether we demonstrated that our selected candidates robustly inhibited random cell migration in various cell lines and assays.

**Drivers of TNBC migration associate with BC progression.** To better understand the regulatory networks driving BC cell migration, we used the larger lists of our PKT validated candidate genes (217 for Hs578T and 160 for MDA-MB-231) to inform on protein-protein interaction (PPI) networks that are involved in Hs578T and MDA-MB-231 cell migration. KEGG pathway analysis of the PKT validated candidates not only confirmed the potent role of transcriptional misregulation in cancer, but also immune-related and splicing pathways in cancer cell migration (Supplementary Fig. 12A). Next, KEGG pathway analysis was performed on the first-order networks of our candidate genes and revealed that similar pathways were affecting cell migration in both cell lines, despite that the networks were constructed from different candidate genes (Fig. 4a, b, Supplementary Data 7). We identified cancer-related pathways such as pathways in cancer and focal adhesion but also immune-related pathways such as osteoclast differentiation and chemokine signaling. To further investigate the connection of our candidate genes to cell migration and invasion, we correlated our signaling networks of three established gene signatures associated with metastatic behavior and cell migration: the Human Invasion Signature (HIS)[28], the Lung Metastasis Signature (LMS)[29,30], and a 440-gene breast cancer cell migration signature. Next, these three independent gene signatures were used to generate minimum interaction PPI networks, which only contained connecting nodes and seed proteins. Both Hs578T and MDA-MB-231 networks show a solid overlap with the 440-gene signature-derived network, with 156 and 145 genes overlapping (Hs578T and MDA-MB-231, respectively) (Supplementary Fig. 12B). This notion is further strengthened by the overlap of the Hs578T and MDA-MB-231 PPI networks with the LMS and HIS signature-based networks: 58 (LMS) and 90 (HIS) genes in overlap with Hs578T network, and 53 (LMS) and 77 (HIS) genes with the MDA-MB-231 network. Furthermore, each gene-signature-derived network showed enrichment for the same KEGG pathways as the PPI networks based on our candidate genes (Supplementary Data 7). Given the high degree of overlap between these three gene signature-based networks and lists of candidate genes, we constructed a single zero-order network based on the combination of candidate genes affecting cell migration in Hs578T and MDA-MB-231 (65 genes, Fig. 4c). This revealed a sub-network linking eight transcriptional regulators of which most already have been related to cancer progression, including *HDAC2*, *BPTF*, *BRF1*, *TAF11*, *TCF12*, and *FOS*[31,32], but also a prominent role for SMADs that are normally driven by TGF-β[33]. However, TGF-β treatment showed limited effects on TNBC cell migration (Supplementary Fig. 13), suggesting that the effect of SMADs on TNBC cell migration is not dependent on TGF-β. Next we systematically investigated the effect of knockdown of the 217 PKT validated hits for morphological changes in the highly polarized Hs578T cell line by actin cytoskeleton staining, confocal imaging, and quantitative single

cell analysis (Supplementary Data 8). Hierarchical clustering grouped our PKT validated hits in nine different clusters (Fig. 4d). Both clusters 2 and 9 contained not only control knockdown samples but also many genes that affected Hs578T cell migration, suggesting that a decrease in migration does not necessarily coincide with an overall change in cell morphology. Not surprisingly, inhibition of cell migration was associated with a wide variety of cellular morphologies. For example, we observed candidates decreasing as well as increasing cell area and cell spikes (reflecting the number of cell protrusions). In vivo, loss of cell adhesion and increased motility are both prerequisites for metastasis formation. However, in vitro, our candidates could inhibit cell migration via different mechanisms such as (1) increased cell-cell adhesions, (2) decreased cell-matrix interactions, and (3) decreased actin turnover that can result in different cell phenotypes. Our combined data suggest that cell shape and motility are affected independently and indicates different genetic programs that define BC cell migration behavior.

**Drivers of cell migration are associated with BC metastasis.** To further relate our candidate genes to breast cancer progression and metastasis formation in patients, we compared our genes to three prognostic signatures (Wang's 76 genes, Yu's 50 genes and NKI-70) for breast cancer metastasis[34–36]. Despite the minimal overlap of genes, these prognostic gene signatures have many related pathways in common[34] and minimum interaction PPI networks showed a robust overlap with our Hs578T and MDA-MB-231 cell migration networks based on PKT screen candidates (217 for Hs578T and 160 for MDA-MB-231) (Fig. 5a and Supplementary Data 9). These three gene expression signatures are strongly predictive of a short time to metastasis, implying but not yet statistically proven that our candidate genes are part of biologically functional regulatory networks and pathways critical in early onset of breast cancer metastasis.

Moreover, we investigated the percentage of mutations, amplifications and deletions (together % altered) of the 43 candidate genes that affected migration in both cell lines (Fig. 3a, b) in 29 cancer types using publicly available data from The Cancer Genome Atlas (TCGA) (Fig. 5b). We identified clusters of candidate genes highly altered in multiple cancer types, among which breast cancer. This alteration rate was not related to tumor type aggressiveness (Supplementary Fig. 14). For the main factors including *BPTF*, *BUD31*, *CACNG1*, *RUNX1*, *GRK1*, *PTPRS*, *PRPF4B*, and *PBX1*, most of these genetic alterations in breast cancer are dominated by amplifications (Fig. 5c), suggesting that enhanced expression levels of these candidates might be involved in breast cancer initiation or progression. Candidate amplification rates were not increased in the more aggressive primary tumors that already metastasized at diagnosis (Supplementary Fig. 15), which might be due to the rather small group of primary tumors with developed metastases (22 tumors in total). 14 candidates among which *BUD31* and *PRPF4B* demonstrated a significant higher amplification rate in TNBC compared to the ER-positive subtype (Supplementary Fig. 16). Consequently, we also evaluated the association of the gene expression of these 43 candidate genes with MFS in the Public-344 cohort (Fig. 5d, Supplementary Data 10). Interestingly, high expression levels of both splicing factors *PRPF4B* and *BUD31* are associated with earlier metastasis formation in triple-negative and ER-positive tumors, respectively (Fig. 5d), but not in the other subtypes (Supplementary Fig. 17). Non-core splicing factor *PRPF4B* is a serine/threonine-protein kinase regulating splicing by phosphorylation of other spliceosomal components[37]. *BUD31* is a core splicing factor essential for spliceosome assembly, catalytic activity and associates with multiple spliceosome sub-complexes and has shown to be a

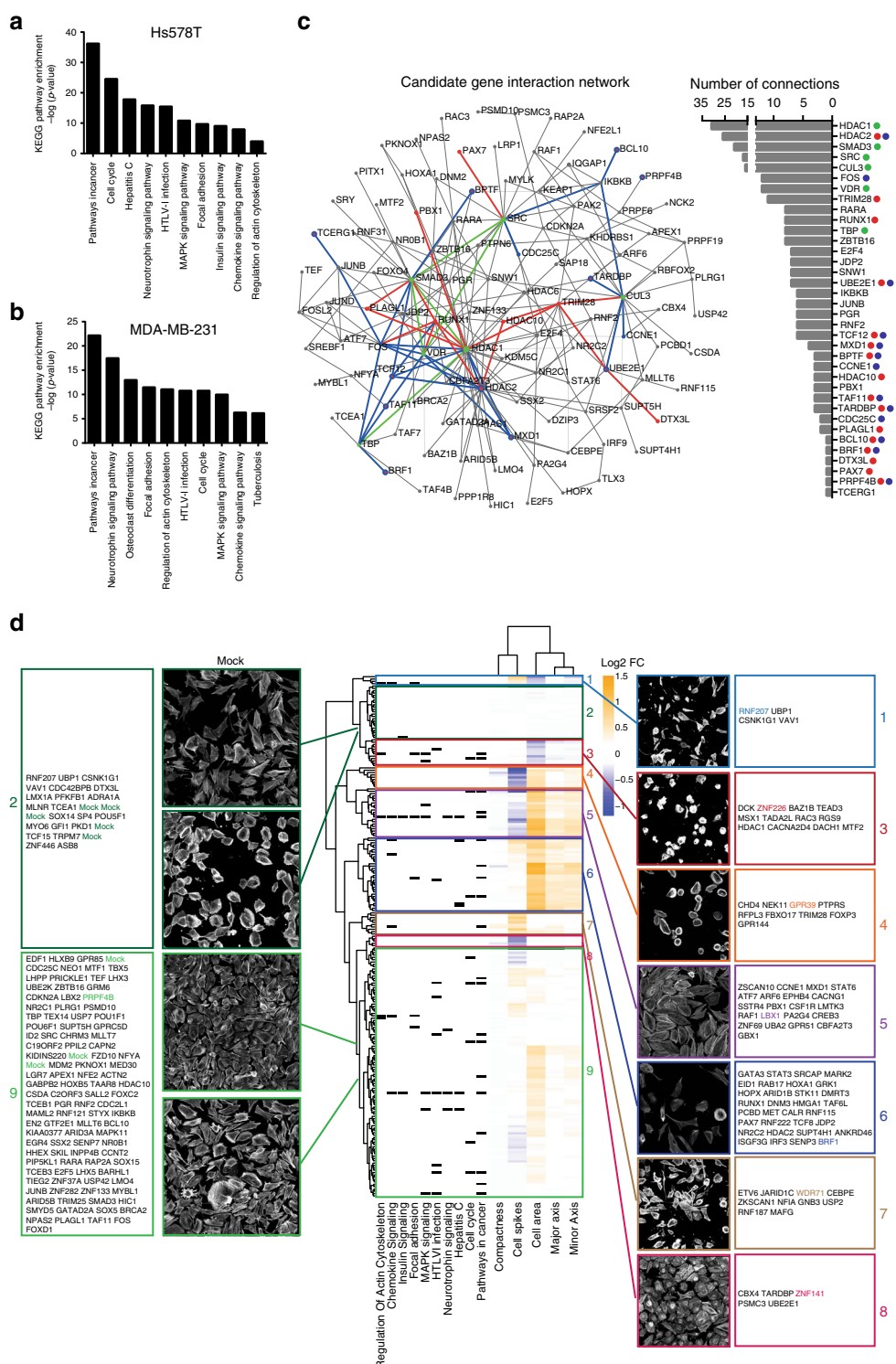

**Fig. 4** Regulatory networks drive tumor cell migration. **a** Enrichment of KEGG pathways in PPI networks generated from Hs578T candidate genes and **b** MDA-MB-231 candidate genes. NetworkAnalyst was used to generate PPI networks. **c** Zero-order interaction network of combined Hs578T and MDA-MB-231 candidate genes reveals a highly connected sub-network of clinically associated genes (in blue). Candidate genes inhibiting cell migration in both cell lines are shown in red; central hubs are highlighted in green. The degree of connectivity (number of connections) is displayed on the right. **d** Phenotype-based clustering of the PKT validated candidate genes based on morphological changes in the Hs578T cell line. Per parameter, log2 fold change (FC) compared to mock control was calculated. Clustering was performed based on Euclidean distance and complete linkage

MYC target in MYC-driven cancer cells[38]. We also identified the transcription factor *BPTF*, known for its role in chromatin remodeling and mammary stem cell renewal and differentiation[39,40], that is highly amplified in many cancer types and

significantly positively correlated to MFS in breast cancer patients irrespective of the subtype (Fig. 5d, Supplementary Fig. 17). We further focused on splicing factors *PRPF4B*, *BUD31*, and transcription factor *BPTF*, since these were newly identified

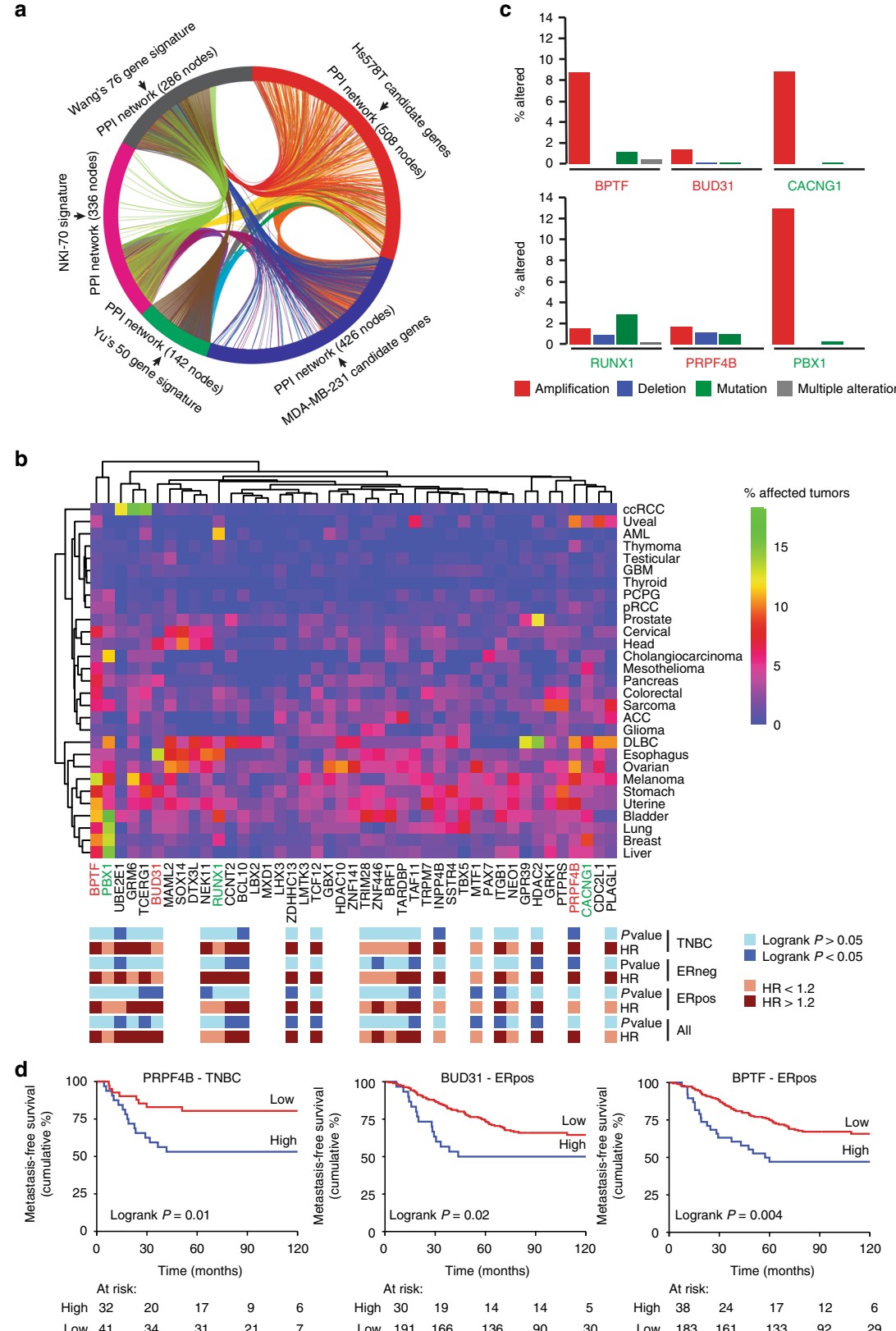

modulators of cell migration associated with BC MFS and/or highly amplified in BC.

**PRPF4B, BUD31, and BPTF modulate cell-matrix adhesion.** Next, we performed knockdown of *PRPF4B*, *BUD31*, and *BPTF* in

MDA-MB-231 and Hs578T cells, followed by next-generation sequencing (NGS)-based transcriptome analysis. For all three candidate genes, knockdown efficiency was >90% in Hs578T cells and >80% in MDA-MB-231 cells (Supplementary Fig. 18A, RT-qPCR validation Supplementary Fig. 19). *PRPF4B*, *BUD31*, and *BPTF* knockdown did not significantly affect proliferation in

**Fig. 5** Modulators of TNBC cell migration are related to BC metastasis-free survival. **a** Prognostic gene signatures were used to generate minimum interaction PPI networks and compared to our candidate TNBC cell migration gene networks. Candidate genes affecting cell migration feed into similar networks essential for BC progression and metastasis formation. **b** Hierarchical clustering (Euclidean distance, complete linkage) of genetic modifications (mutations, deletions and amplifications combined) of 43 candidate genes in 29 cancer types. Data was derived from The Cancer Genome Atlas. Annotation shows the expression of the candidates in relation to BC metastasis-free survival in different BC subtypes. P-values were calculated using Cox proportional hazards regression analysis, with gene expression values as continuous variable and metastasis-free survival as end point. Genes marked in red and blue are highlighted in **c**. Red genes were selected for further analysis. **c** Contribution of different genetic modifications to the rate in several highly mutated or amplified candidates. **d** Kaplan–Meier curves of for expression of *PRPF4B*, *BUD31*, and *BPTF* and relation to metastasis-free survival in ERpos or TNBC breast cancer. Gene expression data of lymph-node-negative BC patient cohort without prior treatment using optimal split was used to obtain the curves

these cell lines (Supplementary Fig. 20). We identified differentially expressed genes (DEGs; log2FC < −1 or > 1; adjusted *p*-value < 0.05) for si*PRPF4B*, *BUD31*, and *BPTF* (Supplementary Data 10). Notably, expression levels of other validated screen candidates available in our RNA-sequencing dataset were not specifically affected by knockdown of *PRPF4B*, *BUD31*, or *BPTF*, indicating that these genes uniquely modulate transcriptional programs that drive TNBC cell migration (Supplementary Fig. 21). Knockdown of *BUD31* had the broadest effect on gene expression and caused downregulation of 1119 genes in Hs578T and 929 in MDA-MB-231, with ~50% affected genes overlapping between the two cell lines (Supplementary Fig. 18B–D). There was limited overlap in the DEGs between *PRPFB4*, *BUD31*, and *BPTF* (Supplementary Fig. 18E). Since *PRPF4B* and *BUD31* are both splicing factors, we investigated the effects of knockdown of these candidates on alternative splicing patterns (Supplementary Fig. 22, Supplementary Data 11–13). Depletion of the core spliceosomal protein *BUD31* mainly increased intron retention (inclusion difference > 10% and *P*-adjusted < 0.01) (Supplementary Fig. 22A and C)[38]. As might be expected from a non-core splicing factor, *PRPF4B* depletion only increased a small number of introns retained (Supplementary Fig. 22B and 22C). All tested intron retention events were validated with RT-PCR (Supplementary Fig. 23), indicating that the computational pipeline we used is reliable. Although the general relation between intron retention and decreased gene expression was previously confirmed[41–43], future studies have to validate the direct causal relationships in response to *PRPF4B* and *BUD31* knockdown. A low number of genes was affected by 3' or 5' alternative splice site usage or alternative exon inclusion upon splicing factor knockdown (Supplementary Data 12), with a limited cell line overlap (Supplementary Fig. 24). This is probably caused by the insufficient sequencing depth (20 million reads compared to 100 million reads recommended) for alternative splicing analysis, prohibiting a definite overall conclusion on differential splicing events. Since si*BUD31*-induced intron retention was related to reduced gene expression, we focused on the differentially downregulated genes for further analysis. We also performed KEGG pathway over-representation analysis using the significantly downregulated genes for all hits in both cell lines separately using ConsensusPathDB[44]. Although the overlap in DEGs between different cell lines and knockdown conditions was rather limited, the ECM-receptor interaction was over-represented in all knock down conditions (Fig. 6a, Supplementary Fig. 25A). Gene set enrichment analysis (GSEA)[45] confirmed this strong down-regulation of the ECM-receptor interaction pathway (Fig. 6b, Supplementary Fig. 25B). Moreover, knockdown of *PRPF4B*, *BUD31* and *BPTF* resulted in downregulation of the focal adhesion pathway in both cell lines, except for *BPTF* in Hs578T. We also observed candidate specific responses such as immune signaling for *PRPF4B* (Fig. 6a), cell adhesion for *BPTF* and metabolic and PI3K related pathways for *BUD31* (Supplementary Fig. 25A). Also, deregulated TNF signaling was validated for all three knockdowns (Supplementary Fig. 26). Clustering of all genes

involved in ECM-receptor interaction (Fig. 6c, see Supplementary Fig. 27 for all gene names) or focal adhesion (Supplementary Fig. 28) demonstrated the involvement of many different pathway components of which some were overlapping between *PRPFB4*, *BUD31*, and *BPTF* (Figs 6c, d). A similar downregulation was observed at the protein level for several key components in both cell lines (Fig. 6e, Supplementary Fig. 29C). The effects on differential expression of cell-matrix adhesion components such as integrins and focal adhesion kinase was also reflected in the different organization of focal adhesions and the F-actin network for both *PRPFB4*, *BUD31*, and *BPTF* (Fig. 6f and Supplementary Fig. 29A–B). In summary, both splicing factors *PRPF4B* and *BUD31* as well as the transcription factor *BPTF* modulate the expression of various focal adhesion-associated proteins and ECM-interaction signaling components in association with distinct cytoskeletal reorganization and decreased BC cell migration.

**PRPF4B is essential for BC metastasis formation in vivo.** Finally, we investigated whether we could translate our in vitro findings to an in vivo mouse model for BC progression. Using our previously established orthotopic xenograft model, we predicted a decrease in BC metastasis formation upon splicing factor *PRPF4B* depletion. We selected *PRPF4B* because its depletion strongly inhibited both random and directed cell migration in both Hs578T and MDA-MB-231 cells (see Fig. 3c and Supplementary Fig. 11) and, moreover, a role for *PRPF4B* in promoting TNBC metastasis formation has not previously been demonstrated. We established stable *PRPF4B* knockdown in the metastatic MDA-MB-417.5 cell line that expresses both GFP and luciferase[12,27,29] and contains similar basal *PRPF4B* levels as its parental MDA-MB-231 cell line (Supplementary Fig. 30A-B). sh*PRPF4B* MDA-MB-417.5 cells demonstrated ~40% *PRPF4B* knockdown at RNA as well as protein level (Fig. 7a–c) and similar as siRNA knock-down, decreased wound healing and intron retention of *DGKZ* and *MAF1* (Supplementary Fig. 30C–H). sh*PRPF4B* cells showed an equal primary tumor growth compared to the two shCtrl cell lines (Fig. 7d), which ensured identical time window for tumor cell dissemination from the primary tumor and outgrowth of macro-metastasis (Supplementary Fig. 31A–B). Interestingly, *PRPF4B* was higher expressed in the borders of the primary tumor (Supplementary Fig. 32), supporting its potential role in invasion and metastasis formation. Bioluminescence imaging demonstrated that lung metastatic spread was less abundant in the *PRPF4B* knockdown group compared to control group (Supplementary Fig. 31C). Both bioluminescent imaging of the lungs ex vivo and counting of macrometastases in the ink injected right lung revealed a significant decrease in metastasis formation in mice engrafted with sh*PRPF4B* cells (Figs. 7f, g), which was also confirmed by a decreased lung weight (Supplementary Fig. 31D). Ex vivo bioluminescence imaging of the liver, spleen, heart, kidney, uterus, and axillar lymph node also showed a decreased metastatic burden by sh*PRPF4B* cells (Fig. 7e) confirmed by a decreased liver and spleen weight (Supplementary Fig. 31E and 31F). Altogether, this demonstrates that *PRPF4B*

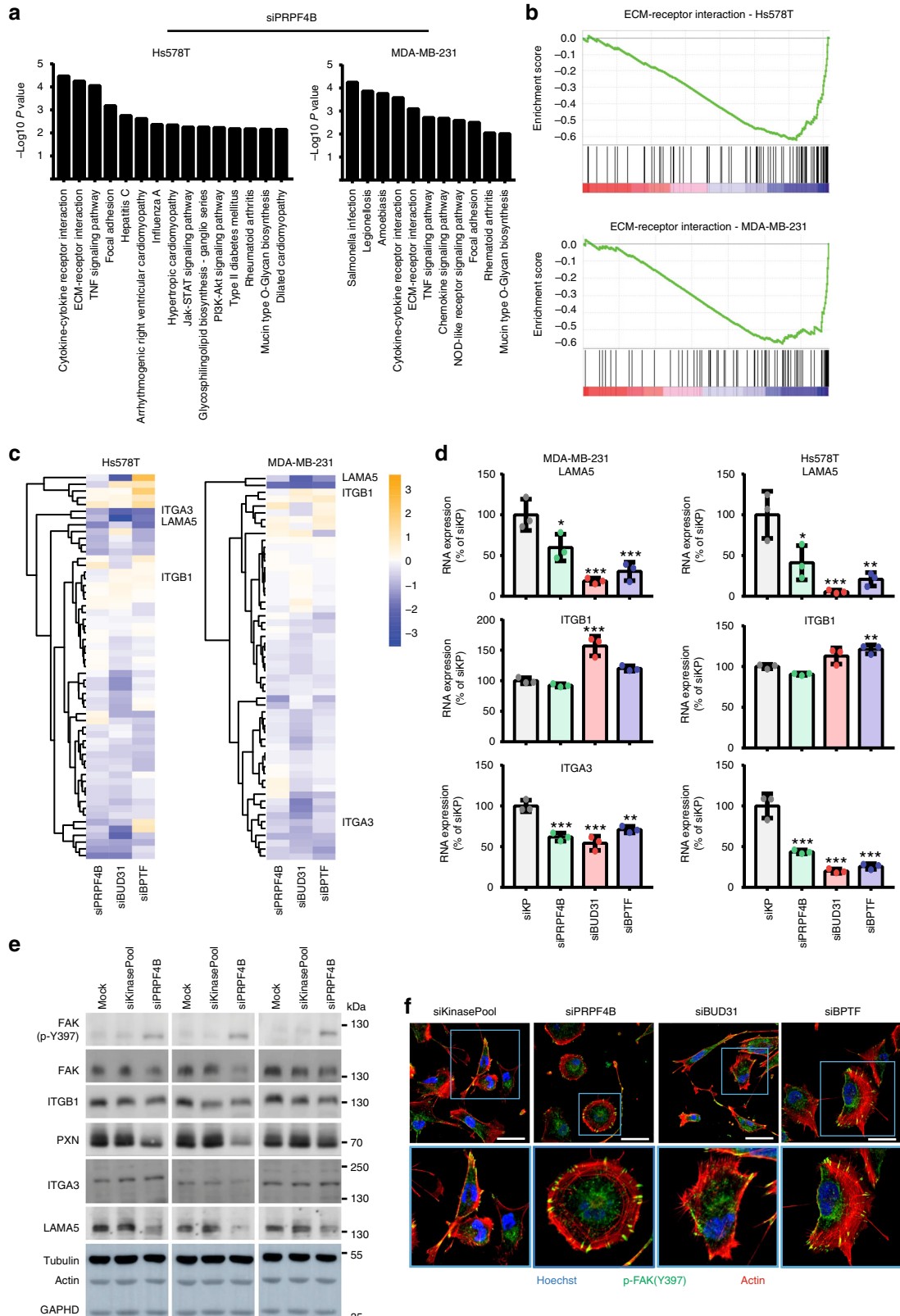

knockdown impairs general metastasis formation without showing organ-specificity.

## Discussion

TNBC is the most aggressive form of breast cancer with 40% of patients dying from metastatic disease. New insights into TNBC migration is highly needed for the identification of potential new drug targets to modulate TNBC dissemination and possibly revert metastatic disease. In the present study, we applied a multiparametric, high-content, imaging-based RNAi-screen covering ~4200 cell signaling components to unravel regulatory networks in TNBC cell migration and discovered 133 and 113 migratory

**Fig. 6** *PRPF4B, BUD31,* and *BPTF* depletion modulates ECM-receptor signaling. **a** Over-representation analysis of genes with decreased expression levels (log2FC change < −1) after PRPF4B knockdown in Hs578T and MDA-MB-231 cells using pathways annotated in the KEGG database. **b** Gene Set Enrichment Analysis (GSEA) identifies the ECM-receptor interaction pathway as significantly enriched in downregulated genes after *PRPF4B* knockdown in Hs578T and MDA-MB-231 cell lines. **c** Hierarchical clustering (Euclidean distance, complete linkage) of log2FC in expression levels of genes involved in the KEGG ECM-receptor interaction pathway after knockdown of candidates demonstrates that many genes involved in this pathway are downregulated after candidate knockdown. **d** *LAMA5, ITGB1,* and *ITGA3* expression upon depletion of *PRPF4B, BUD31* and *BPTF* (mean + s.d of three biological replicates, one-way ANOVA, *p < 0.05, **p < 0.01, ***p < 0.001). **e** Effect of *PRPF4B, BUD31* and *BPTF* depletion on levels of different ECM and focal adhesion components in MDA-MB-231 cells analyzed with western blot. **f** Effect of indicated gene depletion on focal adhesion and actin cytoskeleton organization. Hs578T cells were fixed and stained against the actin cytoskeleton, p-FAK (Y397). Scale bar is 50 μm

modulators in the Hs578T and MDA-MB-231 cell lines, respectively. Splicing factors *PRPF4B* and *BUD31* and transcription factor *BPTF* were critical for TNBC cell migration, associated with metastasis-free survival and affect genes involved in focal adhesion and ECM-interaction pathways. Moreover, *PRPF4B* knockdown reduced TNBC metastasis formation in vivo, making it an important target for future drug development.

Since our screening effort focused on a broad set of signaling components in two TNBC cell lines (Hs578T and MDA-MB-231), we were able to cover a high number of genes and networks in different migration modes. Indeed, the PKT assay allowed us to quantitatively assess different migration phenotypes, as the track morphology reveals the effect on migration, persistence and membrane activity. Enhanced cell migration proved to be difficult to validate, probably due to the increased migratory phenotype in Hs578T and MDA-MB-231 reaching a physiological ceiling. Additional screening with TNBC cell lines with lower migratory potential would provide a platform to discover the spectrum of genes that may act as suppressors of TNBC cell migration and metastasis formation. The majority of our candidate genes displayed inhibition of cell migration and are most interesting for translation to cancer metastasis. Importantly, candidate migratory regulators, including *SRPK1* and *TRPM7*, have previously been shown to impair cell migration and metastasis formation[13,27], supporting the robustness of our candidate drug target discovery strategy.

Our work provides a comprehensive resource detailing the role of individual signaling genes in cell migration. Previously, a cell migration screen in H1299 (non-small cell lung carcinoma) identified 30 candidate migration modulating genes[27]. Surprisingly, there was little overlap with our validated genes, with the exception of *SRPK1*. Similarly, little overlap in hits was found with a wound-healing screen in MCF10A cells[46]. These differences are likely due to the coverage and size of the screening libraries with the current screen covering ~4200 genes compared to ~1400 genes in the previously published data. Moreover, TNBC cell migration might be driven by different genetic program than non-small cell lung carcinoma and MCF10A cells. Also, the MCF10A screen focused on collective cell migration in epithelial cells, which is distinct from single cell mesenchymal migration in our two TNBC cell lines. Since ECM is a major component of the local tumor microenvironment, all our migration assays were performed on fibronectin-coated plates. This coating significantly increased the migratory behavior of our cells (Supplementary Fig. 33), which could also result in different candidates compared to the previous reported MCF10A screen. Moreover, none of the previously identified host-regulators of metastases in mouse[47] were validated in our screen. This might be due to the small overlap (only seven of 23 regulators were in the library) or general differences between in vitro human cell line models and in vivo mouse models. For example, our in vitro cell line models lack the tumor microenvironment containing among others immune cells or tumor-supporting fibroblasts that can modulate the metastatic response. However, the discrepancies

between these studies might also suggest that candidates from our screening approach are particular involved in the first steps in the metastatic cascade such as escaping the primary tumor and intravasation rather than later steps such as extravasation, colonization, and metastatic outgrowth.

The importance of our candidate TNBC cell migration modulators was supported by comparative bioinformatics-based network analysis, demonstrating high similarities between cell migration PPI networks in Hs578T and MDA-MB-231 and metastatic and BC prognostic signatures[35,28,29,34]. Moreover, we identified candidates that are highly amplified and/or mutated in many cancer types as well as candidates specifically related to breast cancer metastasis formation. Interestingly, two of these candidates, *PRPF4B* and *BUD31*, were splicing factors suggesting modulation of the expression of gene networks through alternative splicing. This is in line with some recent studies in which multiple splicing factors and events were related to cancer progression[48–50]. In total, 43 splicing factors were represented in our primary screen of which 11 were selected for further validation in MDA-MB-231 (Supplementary Fig. 34), again suggesting a prominent role for splicing regulation in breast cancer cell migration. Moreover, others have linked more splicing factors such as the *hnRNPs, SRSF1, SRPK1,* and *PTBP1* to breast cancer migration and metastasis formation[27,51–54], making it very interesting to systematically evaluate the role of splicing in breast cancer cell migration in future studies.

Our list of candidate genes that regulate TNBC cell migration expectedly also contributes to cancer metastasis. We selected *PRPF4B* to assess whether our in vitro screening efforts can be translated to in vivo inhibition of metastasis formation. Depletion of PRPF4B strongly inhibited migration of both Hs578T and MDA-MB-231 cells in vitro and almost completely prevented spontaneous metastasis formation from the orthotopic primary tumor to distant organs, indicating that *PRPF4B* seems essential in metastasis formation. *PRPF4B* is a pre-mRNA splicing factor kinase involved in the phosphorylation of *PRP6* and *PRP31* and splicing complex formation. Yet, in our hands depletion of *PRPF4B* showed little effect on intron retention patterns, suggesting more subtle splicing effects. However, *PRPF4B* knockdown did affect the expression of various components of the focal adhesion and ECM signaling pathways, which is likely an important contributor to the reduced migratory and metastatic behavior. *PRPF4B* could be a relevant drug target to combat TNBC dissemination and future research should focus on the development of a specific *PRPF4B* inhibitor; the X-ray structure of the catalytic domain of *PRPF4B* suggest this is feasible[55]. For other potential candidates such as *BPTF*, the correlation with metastasis-free survival added evidence that these candidate genes are likely involved in cancer metastasis. Since these associations do not prove a causal relationship, it would be very interesting to investigate these candidates in similar in vivo studies.

Our list of highly confident candidate migratory modulating genes provides ample opportunities for additional hypotheses and studies in the field of cell migration. Sixteen G-protein coupled

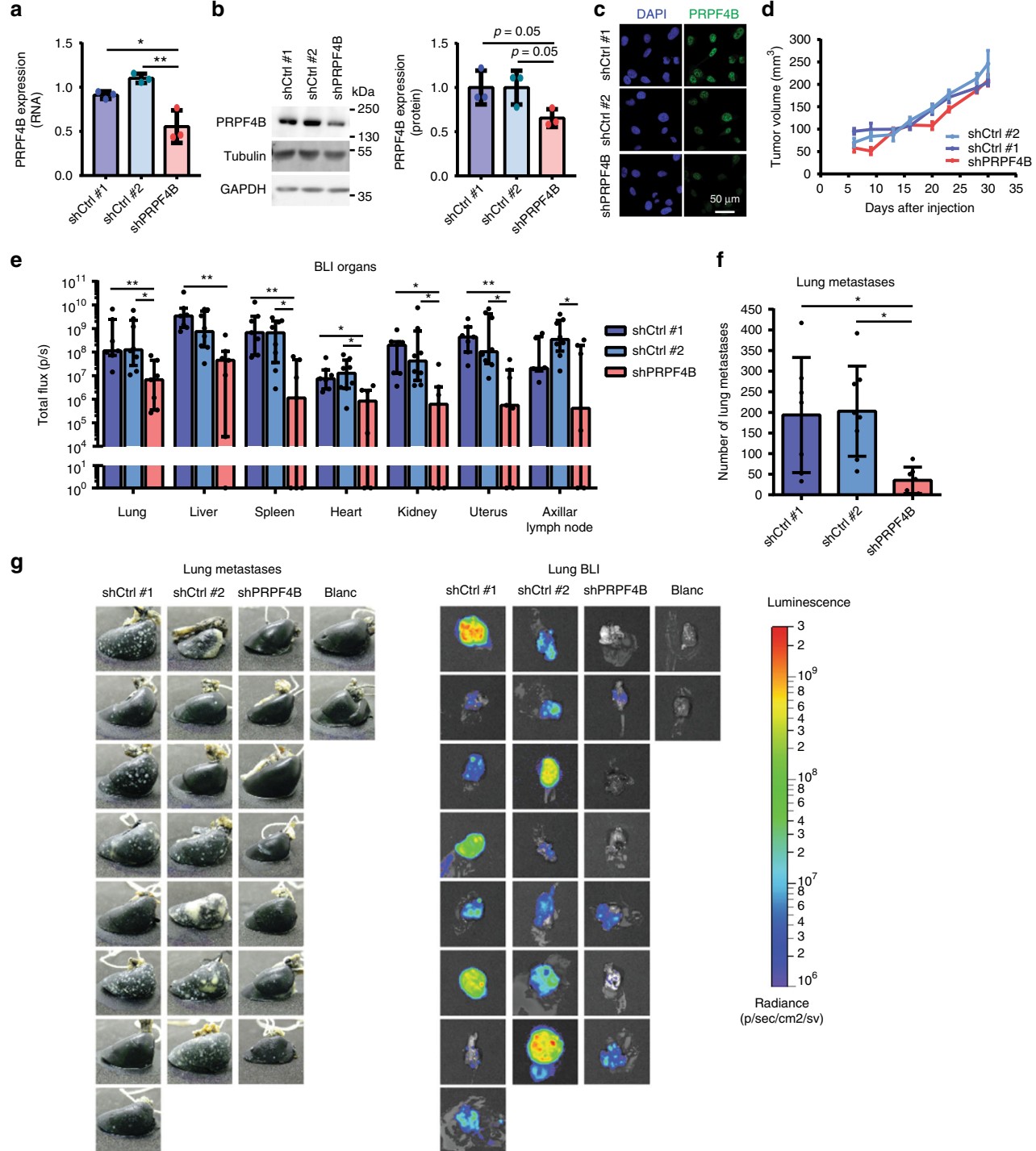

**Fig. 7** *PRPF4B* is essential for breast cancer metastasis formation in vivo. **a** mRNA and **b** protein expression of *PRPF4B* in shPRPF4B MDA-MB-417.5 cell line compared to shCtrl1 and shCrtl2 MDA-MB-417.5 cell lines. Mean + s.d. of three biological replicates. Statistical significance was determined using one-way ANOVA correcting for multiple testing. *p < 0.05, **p < 0.01. **c** Immunofluorescent staining for *PRPF4B* (green) and DNA (DAPI; blue) in shCtrl1, shCtrl2 and shPRPF4B MDA-MB-417.5 cell lines. Scale bar = 50 µm. **d** Tumor growth of sh*PRPF4B*, shCtrl1, and shCtrl2 MDA-MB-417.5 cells engrafted in the mammary fat pad of Rag2$^{-/-}$ Il2rg$^{-/-}$ mice (n = 7–8 animals per group). **e** Dissemination of sh*PRPF4B*, shCtrl1 and shCtrl2 MDA-MB-417.5 cells in the lung, liver, spleen, kidney, heart, uterus, and axillar lymph node as determined by ex vivo bioluminescent analysis (total flux (p/s)) of different target tissues (n = 7–8 animals per group). **f** Number of lung metastases for sh*PRPF4B*, shCtrl1, and shCtrl2 injected mice as determined by macroscopic evaluation of lungs injected with ink (n = 7–8 animals per group). **g** Images of ink injected lungs (left) and bioluminescent signal in the lungs (right) of all individual mice. Blanc indicates mice that did not receive MDA-MD-417.5 cells. Groups were compared by Kruskal–Wallis test with Dunn's post correction test (luminescence) or ANOVA (metastasis count). *p < 0.05, **p < 0.01

receptors were defined, which is particularly relevant as the pathway of GPCR-signaling is one of top over-represented pathways in ER-negative tumors[34]. Depletion of several ubiquitinases and proteasome components (*DTX3L, UBE2E1, RNF31, RNF115, USP2, USP42, PSMC3,* and *PSMD10*) were also found to inhibit tumor cell migration. Protein homeostasis and proteasome function have recently been suggested as a target for proliferation and growth in basal-like TNBC[56]. Transcription factors and regulators make up the largest group of candidate metastasis genes. Some of these factors are part of a large interactive network including *HDAC2, BPTF, BRF1, TAF11, TCF12,* and *FOS*. The downstream targets of these transcription factors are potentially driving BC cell migration, and/or other biological processes that are critical for metastasis formation. Indeed *BPTF* knockdown affected focal adhesion and ECM components in a similar fashion as *PRPF4B* and *BUD31* knockdown did.

In conclusion, in the present study we used imaging-based phenotypic screening to identify candidate metastatic genes for TNBC that have translational relevance. Understanding the gene networks that are controlled by the various candidate genes provides further insights in the biological programs that define BC cell migration behavior and lead to important drug targets to combat cancer metastasis.

## Methods

**Transient siRNA-mediated gene knockdown.** Human siRNA libraries were purchased in siGENOME format from Dharmacon (Dharmacon, Lafayette, CO, USA). Transient siRNA knockdown was achieved by reverse transfection of 50 nM single or SMARTpool (containing four different siRNAs) siRNA in a 96-well plate format using the transfection reagent INTERFERin (Polyplus, Illkirch, France) according to the manufacturer's guidelines. Medium was refreshed after 20 h and transfected cells were used for various assays between 65 and 72 h after transfection. Mock and/or siKinasePool were used as negative control. For the validation screen, four different single siRNAs and one SMARTpool siRNA mix (same four single siRNAs) were tested independently. For all other experiments, SMARTpool siRNAs were used.

**Stable shRNA-mediated gene knockdown.** MDA-MB-417.5 cells were transduced with lentiviral shRNA constructs coding for a non-targeting control sequences shCtrl #1 (SHC002), shCtrl #2 (SHC202V) or a sequence targeting the coding region of *PRPF4B* (target sequence: GCTTCACATGTTGCGGATAAT (TRCN0000426824)) (Mission/Sigma–Aldrich, Zwijndrecht, The Netherlands). The cells were selected by puromycin (sc-108071, Santa Cruz Biotechnology, Heidelberg, Germany). Knockdown efficiency was verified by RT-qPCR, western blot, and immunofluorescent staining.

**Phagokinetic track (PKT) assay.** PKT assays were performed as described before[26,57]. For each transfection, duplicate bead plates were generated (technical replicates); transfection of each siRNA library was also performed in duplicate (independent biological replicate). Procedures for transfection, medium refreshment and PKT assay were optimized for laboratory automation by a liquid-handling robot (BioMek FX, Beckman Coulter).

**PKT imaging and analysis.** Migratory tracks were visualized by acquiring whole-well montages (6 × 6 images) on a BD Pathway 855 BioImager (BD Biosciences, Franklin Lakes, NJ, USA) using transmitted light and a ×10 objective (0.40 NA). A Twister II robotic microplate handler (Caliper Life Sciences, Hopkinton, MA, USA) was used for automated imaging of multiple plates. Montages were analyzed using WIS PhagoTracker[26]. Migratory tracks without cells or with more than one cell were excluded during image analysis. Quantitative output of PhagoTracker was further analyzed using KNIME. Wells with <10 accepted tracks were excluded. Next, data were normalized to mock to obtain a robust Z-score for each treatment and each parameter. After normalization, an average Z-score of the four replicates was calculated. Knockdowns with <3 images were removed, as well as knockdowns with <60 accepted tracks for Hs578T and <150 accepted tracks for MDA-MB-231. Phenotypic classes were determined based on Z scores of the track parameters: small (net area Z score < −4), small round (net area < 8000, axial ratio < 1.7, net area Z score < −1, axial ratio Z-score < −3), big round (net area > 8000, axial ratio < 1.7, net area Z-score > 1, axial ratio Z-score < −4), long rough (axial ratio > 2.4, major axis > 200, roughness > 5, axial ratio Z-score > 1, major axis Z-score > 1), and long smooth (axial ratio > 2.1, major axis > 180, roughness < 5).

All PKT screening data will be available in the Image Data Resource database upon publication and are currently available upon request.

**Live single cell migration assay and analysis.** Hs578T-GFP and MDA-MB-231-GFP cells were transfected with siRNAs as described above and after 65 h, knockdown cell suspensions were seeded in fibronectin-coated black 96-well glass plates (SensoPlate, Greiner Bio-One, Frickenhausen, Germany). As controls, siRNA targeting the GTPase dynamin 2 (DNM2) was used for reduced cell migration and mock was used as transfection control. For the TGF-β and EGF stimulation experiments, cells were seeded in fibronectin-coated black 96-well glass plates (SensoPlate, Greiner Bio-One, Frickenhausen, Germany). Cells were starved in serum-deprived RPMI for 2 h after which cells were stimulated with 10 ng/ml TGF-β (Immunotools) or 100 ng/ml EGF (Peprotech). Imaging was directly started after treatment. Live microscopy was performed on a Nikon Eclipse Ti microscope using a ×20 objective (0.75 NA, 1.00 WD), 2 × 2 binning and 2 × 2 montageTwo positions per well were selected and GFP images were acquired every 12 min for a total imaging period of 12 h using NIS software (Nikon, Amsterdam, The Netherlands). Image analysis was performed using CellProfiler (Broad Institute)[58]. For live cell migration, images were segmented using an in-house developed watershed masked clustering algorithm[59], after which cells were tracked based on overlap between frames. Tracking data was organized and analyzed using in-house developed R-scripts to obtain single cell migration data. Only data originating from cells that were tracked for a minimum of 2 h were used. Two negative control wells with low and high cell densities, comparable to the knockdown populations, were selected for statistical comparison, and knockdowns were required to be statistically significant compared to both controls.

**Imaging-based phenotypic screen.** Hs578T cells were fixed and permeabilized in 1% formaldehyde and 0.1% Triton X-100 in PBS and blocked in 0.5% bovine serum albumin (BSA, A6003, Sigma–Aldrich) in PBS. Cells were stained with Rhodamine Phalloidin (R415, Molecular Probes) and imaged using an Nikon Eclipse TE2000-E inverted confocal microscope (Nikon Instruments, Amsterdam, The Netherlands) using a ×20 Plan Apo objective, 408, and 561 nm lasers. ×2 digital zoom, 2 × 2 stitching images were captured at four positions per well. Nuclei and actin cell body were detected by CellProfiler (Broad Institute) and parameters were measured using the measure object size and shape module in CellProfiler. Images with more than 150 cells were filtered out. Using KNIME, nuclei without a clear cell body were rejected and single cell data were normalized to the median of 2 mock control wells per plate. For the heatmap, all features were mock normalized and clustering was performed on complete linkage and Euclidean distance.

**Next-generation sequencing.** RNA was collected 72 h after knockdown using the RNeasy plus mini kit (Qiagen) according to the manufacturer's guidelines. DNA libraries were prepared with the TruSeq Stranded mRNA Library Prep Kit and sequenced according to the Illumina TruSeq v3 protocol on an Illumina HiSeq2500 sequencer. $20 \times 10^6$ 100 base pair paired-end reads were generated and alignment was performed using the HiSat2 aligner (version 2.2.0.4) against the human GRCh38 reference genome. Gene expression was quantified using the HTseq-count software (version 0.6.1) based on the ENSEMBL gene annotation for GRCH38 (release 84). Count data were normalized and log2 fold changes and adjusted *P*-values were calculated using the DESeq2 package[60]. Calculated log2 fold changes were used to perform ranked GSEA[45]. DEGs were selected by effect size (log2 fold change bigger than 1 or smaller than −1) and adjusted *p*-value (smaller than 0.05) and used for over-representation analysis for KEGG pathways using ConsensusPathDB[44].

For the intron retention analysis, RNA-seq reads were mapped to the current human genome (GRCh38) using Hisat 2[61]. Differential intron retention analysis was carried out in R using DexSeq package[62,63]. In DexSeq the difference of intron inclusion were determined based on the counts from the intron and the counts from the two adjacent exons. The sizes of the exons were limited to 100nt immediately adjacent to the intron to reduce artifacts deriving from alternative promoters, alternative splice sites and alternative poly-adenylation sites. Deferentially retained introns were selected by effect size (relative change in inclusion more than 0.1) and statistical significance of the change (adjusted *p*-value less than 0.01). Alternative exon inclusion was analyzed using the rMATS package version 3.0.8[64]. Differentially spliced exons were selected by effect size (relative change in inclusion more than 0.1) and statistical significance of the change (FDR < 0.05).

RNA sequencing data are available in Sequence Read Archive with accession number SRP127785.

**Network analysis.** Protein annotation of the primary hits was retrieved from QIAGEN's Ingenuity Pathway Analysis (IPA, QIAGEN Redwood City, USA). Protein-protein interaction (PPI) networks were generated separately for all signatures using NetworkAnalyst (www.networkanalyst.ca)[65]. Candidate genes were used as seed proteins to construct first-order, minimum interaction and zero-order networks based on the InnateDB Interactome. KEGG pathway analysis was performed on the first-order PPI networks. The connection between multiple PPI networks was visualized by a Chord diagram using NetworkAnalyst.

**Cell culture.** Hs578T (ATCC-HBT-126) and MDA-MB-231 (ATCC-HBT-26) were purchased from ATCC. MDA-MB-417.5 (MDA-LM2) was kindly provided by Dr. Joan Massagué. All cell lines were grown in RPMI-1640 medium (Gibco,

ThermoFisher Scientific, Breda, The Netherlands) supplemented with 10% FBS (GE Healthcare, Landsmeer, The Netherlands), 25 IU/ml penicillin, and 25 µg/ml streptomycin (ThermoFisher Scientific) at 37 °C in a humidified 5% $CO_2$ incubator. When not stated otherwise, experiments were performed in this full RPMI medium. Stable GFP-expressing Hs578T and MDA-MB-231 cells were generated by lentiviral transduction of pRRL-CMV-GFP and selection of GFP positive clones by FACS. For live cell imaging, phenol red-free culture medium was used. All cell lines used in this study are not listed in the database of commonly misidentified cell lines maintained by ICLAC and were tested for mycoplasma contamination. The authenticity of all the cell lines was checked by short tandem repeat (STR) profiling using the PowerPlex® 16 System (Promega, Madison, WI, USA) including fifteen STRs and one gender discriminating locus using 10 ng of genomic DNA isolated with QIAamp DNA Mini Kit (Qiagen, Hilden, Germany) for the multiplex PCR. The authenticity of the cell lines was assessed based on the source STR profiles of the American Type Culture Collection (ATCC) and the Deutsche Sammlung von Mikroorganismen und Zellkulturen (DSMZ).

**Antibodies and reagents.** Rabbit anti-PRPF4B (8577, Cell Signaling Technology), mouse anti-Tubulin (T-9026, Sigma–Aldrich), mouse anti-Paxillin (610052, BD Biosciences), mouse anti-ITGB1 (610467, BD Biosciences), mouse anti-FAK (610087, BD Biosciences), mouse anti-N-cadherin (610920, BD Biosciences), mouse anti-E-cadherin (610181, BD Biosciences), rabbit anti-p-FAKY397 (446–24ZG, Thermo Fisher), and mouse anti-Vimentin (ab8069, Abcam) were all commercially purchased. All antibodies were used in a 1:1,000 dilution. Rabbit anti-ITGA3 and rabbit anti-Laminin5 were kindly provided by A. Sonnenberg (NKI, Amsterdam, The Netherlands). Anti-mouse and anti-rabbit horseradish peroxidase (HRP) conjugated secondary antibodies were purchased from Jackson ImmunoResearch.

**Cell migration scratch assay.** Hs578T and MDA-MB-231 cells were transfected with siRNAs as described above and after 65 h, knockdown cell were counted and seeded (25,000 cells/well for Hs578T, 35,000 cells/well for MDA-MB-231, 35,000 cells/well for MDA LM2) in a fibronectin-coated black 96-well screenstar plate (Greiner Bio-One, Frickenhausen, Germany). Six hour after seeding cells were stained with Hoechst 33342 (1:10,000) for 1 h, scratches were made using a pipet tip and the medium was refreshed. Live microscopy was performed on a Nikon Eclipse Ti microscope using a ×4 objective. Images were acquired every hour for a total imaging period of 22 h using NIS software (Nikon, Amsterdam, The Netherlands).

**Boyden chamber assay.** Hs578T and MDA-MB-231 cells were transfected with siRNAs as described above. 65 h after knockdown, cells were starved in serum-deprived medium for 6 h. After starvation, 50,000 cells were plated in 0.3% FBS in medium in ThinCert inserts (8 µm pore size, Greiner) which were placed in a 24-wells plate filled with 600 µl medium containing 10% FBS or 0.3% FBS as a negative control. After 22 h, the culture medium in the wells was replaced by 450 µl serum-free medium containing 8 µM Calcein-AM (Sanbio/Caymen) and incubated for 45 min at 37 °C. Inserts were transferred to a freshly prepared 24-well culture plate containing 500 µl pre-warmed trypsin-EDTA per well and incubated for 10 min at 37 °C. 200 µl of the trypsin-EDTA solution was transferred into a black flat bottom 96-well plate (µclear plate, Greiner) and fluorescence was measured with a FLUOstar OPTIMA plate reader (BMG labtech, Offenburg, Germany) using an excitation wavelength of 485 nm and emission wavelength of 520 nm.

**Inducible CRISPR-Cas9 knockout.** Inducible Cas9 cell lines were obtained by transduction of MDA-MB-231 cell lines with lentiviral the Edit-R inducible lentiviral Cas9 plasmid (Dharmacon). Cells were selected using 2 µg/ml blasticidin and grown single cell after which a clone was selected that was fully Cas9 inducible; from no called MDA-MB-231 ind-Cas9. sgRNAs were obtained from the human Sanger Arrayed Whole Genome Lentiviral CRISPR Library (Sigma–Aldrich) (sgPRPF4B #1: ATGCCAGCCCCATCAATAGATGG, sgPRPF4B #2: GGAGCAGATCACGCTTGCGAAGG, sgBUD31 #1: ACAAAAACCTGATTGCAAAATGG, sgBUD31 #2: ATGAGAACTTGTGCTGCCTGCGG, sgBPTF #1: CCGGATGACATCAATTGAAAGAG, sgBPTF #2: AAACGATGCAGCAAGCGACATGG). Inducible knockout cell lines were obtained by lentiviral transfection of the sgRNAs into the MDA-MB-231 ind-Cas9 cell line after which the cells were selected using 1 µg/ml puromycin. Cells were treated for 72 h with 1 µg/ml freshly prepared doxycycline and western Blot and live cell migration assays were performed as described above.

**Immunofluorescence.** Cells were fixed and permeabilized 72 h after knockdown by incubation with 1% formaldehyde and 0.1% Triton X100 in PBS and blocked with 0.5% w/v BSA in PBS. Cells were incubated with the primary antibody in 0.5% w/v BSA in PBS overnight at 4 °C and incubated with the corresponding secondary antibodies and 1:10,000 Hoechst 33258 for 1 h at room temperature. Cells were imaged with a Nikon Eclipse Ti microscope and 60x oil objective.

**Western blotting.** Cell lysis and western blotting was performed as described before (Zhang, 2011). Blots were visualized using the Amersham Imager 600 (GE Healthcare). At least two biological replicates were performed per experiment. Tubulin was used as a loading control. Uncropped gel images can be found in the Source Data file.

**PCR.** Forty-eight hour after plating stable knockdown cell lines, total RNA was extracted using RNeasy plus mini kit (Qiagen) followed by cDNA synthesis using the RevertAid H minus first strand cDNA synthesis kit (Thermo Fisher Scientific) both according to the manufacturer's protocol. RT-qPCR was performed with the SYBR Green PCR master mix (Thermo Fisher Scientific) on a 7500 Fast Real-Time PCR machine (Applied Biosystems/Thermo Fisher Scientific). Conventional PCRs were performed using MyTaq Red Mix (Bioline) according to the manufacturer's protocol. The following primers were used: PRPF4B forward: 5'-CCGAGGAGT CAGGAAGTTCA-3', PRPF4B reverse: 5'-TCTTTTCAGAATTAGCATCTTC CAT-3'; GAPDH forward: 5'-CTGGTAAAGTGGGATATTGTTGCCAT-3', GAPDH reverse: 5'-TGGAATCATATTGGAACATGTAAACC-3', β-actin forward: 5'-TCAAGATCATTGCTCCTCCTGAG-3', β-actin reverse: 5'-ACATCTGCTGGAAGGTGGACA-3', CSF1 forward: 5'-CCCTCCCACGA-CATGGCT-3', CSF1 reverse: 5'-CCACTCCCAATCATGTGGCT-3', CSF2 forward: 5'-GCCCTGGGAGCATGTGAATG-3', CSF2 reverse: 5'-CTGTTTCATTCATCTCAGCAGCA-3', IL6 forward: 5'-TCAATATTA GAGTCTCAACCCCCA-3', IL6 reverse: 5'-GAAGGCGCTTGTGGGAGAAGG-3', MMP3 forward: 5'-CACTCACAGACCTGACTCGG-3', MMP3 reverse: 5'-AGT CAGGGGGAGGTCCATAG-3', PIK3CD forward: 5'-GTCCCCTGGGCAACTGTC-3', PIK3CD reverse: 5'-GCCTGACTCCT TATCGGGTG-3', BUD31 forward: 5'-CATTCAGACACGGGACACCA-3', BUD31 reverse: 5'-ATGATGCGGCCCACTTCC-3', BPTF forward: 5'-GGCATCTTGCAAAGTGAGGC-3', BPTF reverse: 5'-TATGGGCCTGTAAG GAACGG-3', DGKZ intron 7 forward: 5'-TGCTCGTGGTGCAAGCA-3', DKGZ intron 7 reverse: 5'-AGCATGAAGCAGGACACCTT-3', POMGNT1 intron 20 forward: 5'-TGCCTCCATATCTGGGACCT-3', POMGNT1 intron 2 reverse: 5'-GTGACTGAGGGTGGCTTCTT-3', MAF1 intron 4 forward: 5'-ATCTGCCTGGCTGAATGTGAC-3', MAF1 intron 4 reverse: 5'-GGATCT GAGTCCAAGTCTGGGT-3', CDCA5 intron 1 forward: 5'-AGTTATGTCTGG GAGGCGAA-3', CDCA5 intron 1 reverse: 5'-TCAGAGCCTGATTTCCGCT-3', CDCA5 intron 2 forward: 5'-AGCGGAAATCAGGCTCTGA-3', CDCA5 intron 2 reverse: 5'-AGACGATGGGCTTTCTGACT-3', SRRT intron 18 forward: 5'-AGGCCAGGGAGGTTATCCT-3', SRRT intron 18 reverse: 5'-TTGGGTCTCCACGAACCAT-3', ELAC2 intron 19 forward: 5'-AGATTGAT CAGTTCGCTGTTGC-5', and ELAC2 intron 19 reverse.

Relative gene expression was calculated after correction for GAPDH and β-actin expression using the 2ΔΔCt method.

**Orthotopic mouse model for metastasis assessment.** $1 \times 10^6$ MDA-MB-417.5 shCtrl #1, shCtrl #2, or shPRPF4B cells diluted in 100 µL matrigel (9.2 mg/ml, 354230, batch 4321005, Corning, Amsterdam, The Netherlands) were injected in the fourth mammary fat pad of 7–9-week-old female Rag2−/− Il2rg−/− mice ($n = 8$ per group: coefficient of variation = 0.2, effect size = 30%, power π = 0.90 bij α = 0.05). Primary breast tumors were surgically removed when they reached the size of $7 \times 7$ mm. Next, bioluminescent imaging was used to follow metastasis formation over time. Mice were sacrificed 50 or 51 days after surgery and metastasis formation of all organs was assessed by bioluminescent imaging followed by weighing the lungs, liver, and spleen. The right lung was injected with ink in order to count the number of lung macrometastases. Animals that passed away due to unknown reasons before the end of the study were removed from the analysis. Animals were randomly distributed over the different groups. The experiment was performed without blinding.

**Breast cancer patient gene expression profiles.** Gene expression data of a cohort of 344 lymph node-negative BC patients (221 estrogen receptor-positive (ER-positive) and 123 estrogen receptor-negative (ER-negative)), who had not received any adjuvant systemic treatment, were used and is available from the Gene Expression Omnibus (accession no. GSE5327 and GSE2034). Clinical characteristics, treatment details and analysis were previously described[34,35,66–68]. Stata (StataCorp) was used to perform Cox proportional hazards regression analysis, with gene expression values as continuous variable and MFS as end point.

**Statistical analysis.** Sample sizes were based on previously published similar experiments. When not indicated, all experiments were performed in biological triplicates. Normality of migration measurements and in vivo data were tested using Kolmogorov–Smirnov's test, d'Agostino and Pearson's test and Shapiro–Wilk's test using GraphPad Prism 6.0 (GraphPad Software, San Diego, CA). A data set was considered normal if found as normal by all three tests. Data sets following a normal distribution were compared with Student's t-test (two-tailed, equal variances) or one-way ANOVA (for comparison of more than two groups) using GraphPad Prism 6.0. Data sets that did not follow a normal distribution were compared using Mann–Whitney's test or a non-parametric ANOVA

(Kruskal–Wallis with Dunn's multiple comparisons post-test) using GraphPad Prism 6.0. Results were considered to be significant if $p$-value $< 0.05$.

**Study approval**. The study involving human BC patients was approved by the Medical Ethical Committee of the Erasmus Medical Center Rotterdam (Netherlands) (MEC 02.953). This retrospective study was conducted in accordance with the Code of Conduct of the Federation of Medical Scientific Societies in the Netherlands (www.federa.org). Mouse experiments and housing were performed according to the Dutch guidelines for the care and use of laboratory animals (UL-DEC-11244).

## Data availability
All datasets used to generate the results presented in this study are publicly available. RNA sequencing data are available in Sequence Read Archive with accession number SRP127785. Images are publicly available as resource datasets in the Image Data Resource (IDR) [https://idr.openmicroscopy.org] under accession number idr0022[69]. Raw data from Figs. 3a, b, 6d, 7a, b, 7d–f and Supplementary Figs. 9B, 10, 11B, 11D–F, 13, 18A, 19B, D, 23A, B, 26, 30B, 30D–H, 31A–F, 32B, C, 33 are provided in a source data file. All other remaining data are available within the article or supplemental data.

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

## Acknowledgements

This project was supported by the EU FP7 Systems Microscopy Network of Excellence (grant no. 258068), the ERC Advanced grant Triple-BC (grant no. 322737), and the Dutch Cancer Society project (grant nr 2011–5124).

## Author contributions

M.F., E.K., V.M.R., S.E.L.D., and B.v.d.W. conceived and designed the experiments. M.F., E.K., V.M.R., S.E.L.D., I.v.d.S., C.P., M.A.T., and J.E.K. performed the experiments. E.K. and M.F. analyzed the data. E.A.C.W., M.S., J.A.F., and J.W.M.M. provided the clinical data. P.S. performed the intron retention analysis. H.d.B. provided technical support for the microscopes. S.W. arranged image reformatting and upload to the Image Data Resource. E.K. and M.F. wrote the manuscript. B.v.d.W., S.E.L.D., J.A.F., and J.W.M.M. reviewed and corrected the manuscript.

## Additional information

**Competing interests:** The authors declare no competing interests.

