## [Peer Review File · Nature Communications]

Reviewers' comments:

Reviewer #1, Expertise: TNBC, metastasis
(Remarks to the Author):

This is an interesting manuscript on triple-negative breast cancer, looking at motility pathways to provide new insights into translatable pathways. TNBC is an orphan disease with no molecular therapeutics and a dearth of broadly expressed translatable pathways. Focusing on motility, the authors use an incredible array of bioinformatic analyses to distill down to a handful of genes, one of which is shown to regulate metastasis *in vivo*.

I have two major comments: First, this manuscript is written in a manner that a biologist cannot easily understand. I dug through the supplementary data, and still did not understand what some of the data meant. Second and more importantly, there is too little *in vivo* and mechanistic data. One gene is confirmed in one metastasis assay with only the barest of endpoints. Other types of motility assays are not performed.

Specific comments are listed below:

1. It is difficult from the main text to understand the screening assay. Is this directed or random motility? Would the results change if it is the other? How were the parameters selected, how did you come up with long smooth and long rough? A reader should not have to dig too deep into supplemental data to understand the basics of this assay.
2. Is there a difference between cell shape and motility? Fig. 4D actually breaks down the genes by cell morphology. This is discussed briefly later in the manuscript, but the question remains.
3. Given a 7 hr assay, what is the contribution of cell proliferation? Many labs use a 4 hr assay for MDA-MB-231 cells to eliminate that aspect. Cell cycle was a class of proteins in both cell lines (Fig. 4).
4. "we focused our screening effort on the complete set of cell signaling components, covering all kinases, phosphatases, (de)ubiquitinases, transcription factors, G-protein coupled receptors, epigenetic regulators and cell adhesion-related molecules (4198 individual target genes in total)." Does this include the cytoskeletal proteins? If not, why? Could they not provide a number of feedback regulatory loops?
5. Can the authors test some of these genes in traditional Boyden chamber or scratch migration assays so that the field can understand the overlap?
6. I am unsure of the significance of the interaction map on Fig. 4C. It looks like red lines go different places than blue or green, but apparently "connect" as shown on the graph to the right? The main connectors are HDACs. Wouldn't that be the case for sets of genes associated with many phenotypes?
7. How different would this look if motility was directed to a particular attractant? The SMADs are listed prominently in Fig. 4. Is TGF- β driving this or are SMADs crucial to random migration?
8. On Fig. 5, Kaplan-Meier curves are shown for genes and particular subtypes of breast cancer. Were they the only subtypes with significant associations?
9. Does it matter if gene knockdowns are performed on cells in tissue culture versus on ECM, as in a Boyden chamber assay?
10. Many gene signatures in cancer and metastasis are replete with ECM alterations. Were other classes of altered genes examined?
11. This manuscript is a tour-de-force in bioinformatics, but is lean on functional data. *In vivo* data for one gene and one cell line are presented. There is no mechanistic evaluation- for instance, were the primary tumors more or less invasive? Where was PRPF4B expressed- the center of the tumor versus the invasive front? How about a tail vein assay where initial extravasation is omitted

and an impact on colonization can be seen? At a minimum, two models should be shown in vivo, and the other genes should be tested.

Reviewer #2, Expertise: RNAi screen
(Remarks to the Author):

Enclosed is a review of the manuscript. Overall it's a solid experiment, quite an interesting one as well. There are some limitations as pointed out in the critique, however, the manuscript left very good impression of the work. Additional experiments at the lab bench are minimal, however, a reanalysis of the TCGA data, applying elements of experimental design, is heavily advisable

Review of "Uncovering the signaling landscape controlling breast cancer cell migration identifies novel metastasis drivers" by Fokkelman, et al.

The article presented by Fokkelman et al. is an elegant experiment where an exhaustive screen of 4200 target genes involved in cell signaling was carried out to ascertain their impact on cell migration related traits. PKT = phagokinetic track assay. Through multiple rounds of experimental validation by SMARTpool, siRNA, live-cell microscopy in GFP-expressing cells, and the analysis of RNA-seq gene expression data the authors consistently replicated their initial findings by RNAi, culminating in an in-vivo experiment that clearly exemplifies their top candidate PRPF4B as a driver of metastasis in breast cancer.

Comments and Suggestions:

This study made good use of high-content imaging, coupled with siRNA screening to measure the phenotypic effects on cell migration and morphology as it relates to metastatic breast cancer. There are a number of issues that need to be addressed from the point of view of high-content imaging and informatics.

1. In the identification of hits for migration using the PKT assay, there needs to be some analysis of viability as a result of gene expression knockdown. This would ideally be the result of measuring the cell number following siRNA treatment and prior to seeding in bead plates, or from published data in these cell lines.
2. The principal component analysis performed on the small number of measures is probably insufficient to describe differences if viability measures are not included.
3. This type of screen has been performed previously for morphology and migration, so this aspect of the study is not particularly novel.

1: Pascual-Vargas P, Cooper S, Sero J, Bousgouni V, Arias-Garcia M, Bakal C. RNAi screens for Rho GTPase regulators of cell shape and YAP/TAZ localisation in triple negative breast cancer. *Sci Data*. 2017 Mar 1;4:170018. doi:

10.1038/sdata.2017.18. PubMed PMID: 28248929; PubMed Central PMCID: PMC5332010.

2: Taniguchi H, Hoshino D, Moriya C, Zembutsu H, Nishiyama N, Yamamoto H, Kataoka

K, Imai K. Silencing PRDM14 expression by an innovative RNAi therapy inhibits stemness, tumorigenicity, and metastasis of breast cancer. *Oncotarget*. 2017 Jul 18;8(29):46856-46874. doi: 10.18632/oncotarget.16776. PubMed PMID: 28423353;

PubMed Central PMCID: PMC5564528.

3: Williams SP, Gould CM, Nowell CJ, Karnezis T, Achen MG, Simpson KJ, Stacker SA. Systematic high-content genome-wide RNAi screens of endothelial cell migration and morphology. *Sci Data*. 2017 Mar 1;4:170009. doi: 10.1038/sdata.2017.9. PubMed PMID: 28248931

The results of the study show a bias where the validation experiments preferentially replicated targets conferring a reduced cell migration phenotype. The bias could arise from limitations in the methods as they only test knockout and knockdown expression of initial targets deemed significant. The other alternative is from limitations in the cell lines chosen. Assuming the latter, it is possible that knockdown and knockout of targets resulting in increased cell motility were masked by the variants that increase cell migration already present in Hs578T and MDA-MB-231. Thus bringing the phenotype to a physiological ceiling, and consequently reducing the sensitivity to identified targets that upon inhibition increase cell migration. The use of a lower motility cell line in this study would provide a platform with an increased sensitivity to identify suppressors of cell migration i.e. inhibition of the target increases cell migration as the genetic background would be devoid of cell migration promoting alleles. These limitations should be discussed in greater detail because they open opportunities for additional screens that can elucidate in the landscape of metastasis drivers.

There was little discussion on the validated targets from the cell-line specific genes. These cell line specific genes may operate through an epistatic interaction with mutated genes specific to the cell line. Were there loss of function (LOF) or activating mutations, or copy number alterations on genes in the pathways related to the cell line specific targets?

There is a leap of causality in the manuscript by specifically highlighting that genes affecting cell migration traits will drive a metastatic phenotype in the tumor. This was primarily based on the gene expression correlations of validated targets to genomic alteration in different tumor types. However, there is no statistical test or any other evidence to each gene on their metastatic potential, outside of the in-vivo validation of PRPF4B. These arguments should be further supported or discussed in greater detail.

The use of external data sources to identify supporting evidence for candidate targets is not well described in the methods. There was no description of how the cancer types were partitioned to identify supporting evidence for each gene. For example, cancer types can be divided by their metastatic potential, as highly metastatic and non-metastatic, then carry out a burden test between the two groups asking if the frequency of LOF, activating, or other genetic alterations different between the two groups? Or for differential gene expression; where the candidate genes over- or under-expressed in highly metastatic tumor types vs non-metastatic?

Furthermore, clinical data for the patients on TCGA is available. This contains the AJCC Metastasis Pathologic stages, metastatic site, and metastatic tumor, which can be used to partition the samples to perform burden and differential expression comparisons. There is no mention of the project# to access TCGA data.

Additional data from whole genome sequencing data of 560 genomes is publicly available by Nik-Zainal et al. (2016), ENA:EGAS00001001178. It would be worthwhile to investigate if the burden tests proposed above can also be carried out using this data set. The independent cohort would strengthen the author's findings. Lastly, Van der Weyden et al. (2017) carried out an in-vivo screen for host-regulators of metastases in the mouse. Where any of the 217 primary hits present in their significant host-regulators?

The analysis of Human Invasive Signatures, Lung Metastasis Signatures and the list of 440 genes from Rogkoti et al, (unpublished) is controversial because the results are enriched in the unpublished data set. It is unclear the extent of the strength of evidence this data set is providing relative to the rest. Do the results change if this set is taken out?

A careful revision for clarity is needed in the results section. The manuscript conveys the complete scope of the experiment. However, the results sections are not woven together in a concise

manner making them difficult to follow. The section titled "Transcriptional determinants are critical drivers of BC migratory phenotypes" was the most difficult to read as the list is narrowed down in the validation process. These sections are missing a details such as the sum total of 2807 primary hits (1501 + 1306), and the exact number of candidates used in the validations in each of the two cell lines as 298 targets, composed of the 145 overlapping and 153 cell-line specific (or 451 unique targets composed of 129 overlapping + 153 Hs578T + 153 MDA-MB-231). The sections "Functional drivers of tumor cell migration partner in networks predictive for BC progression" and "Modulators of cell migration are associated with BC metastasis-free survival" continue to narrow down the initial list the lists of 298 targets per cell line (or the 451 unique targets) to those that validated in SMARTpool, siRNA, and GFP+ live cell imaging. However, the numbers are not consistently clear throughout these sections.

Specific observations

Page 5 line 22 One would wish to see the key hits of the focused screen validated by shRNA and sgRNA CRISPR-Cas9 methodologies. These cross-checks are critical for molecular target confirmation experiments and rather straightforward to do.

Page 6, line 7 How much of the "reduced cell migration" was due to cytotoxicity rather than specifically impacting candidate genes involved in the process of cell migration? A generalized cytotox counterscreen would be appropriate, even in the same two cell backgrounds.

Page 8, second paragraph The tie-in between morphological changes and migration, to differential genetic backgrounds, particularly with respect to cell protrusions (clusters 4, 5, 7 and 8) is rather weak. One may consider a RNAseq of cluster 7 versus the other three groups (4, 7, 8) to tease-out these differences.

Page 9, second paragraph: Why was GRK1 not mentioned in the text amongst the "top key factors"? Also, two of the top three hits were splicing factors, yet the "spliceosome" per se was not one of the focused libraries (see Suppl Fig 1, upper left).

Page 12 and Fig 7A. Related to the "essentiality" of PRPF4B: Why did this gene hit in particular rise in importance above the other quality hits? The 40% knockdown (KD) is statistically significant but on a pure level of KD basis, rather unimpressive. This may suggest that in addition to impairing the function of "general metastasis", there may be other splicing factors or off-target effects at play. The line of reasoning, and questions about it, continue in the first paragraph on page 15.

Page 13 The initial sentence and paragraph of the Discussion should recapitulate the most important findings of the results. This does not occur until the second paragraph, in a way hiding the main results. Reorganizing the Discussion to flow from specific findings to broad discussion would improve the readability of this section and emphasize the main results in concise manner.

Page 19, line 18 "microscope using" (make into separate the words)

Page 28 typo in the Science reference #24.

Page 29, reference #49 Was this cited? Regrets if I did not see it in the text.

Page 31, Figure 1G Not much overlap between the hits from both cells lines, which makes one wonder if a third cell line is needed to clarify the results? See also page 40 Suppl Figure 3A for more non-similarity between Hs578T and MDA-MB-231.

Page 32 In Figure 2 and Supplementary Figure 2, the plots of panel B obfuscate the message of reproducibility. Adding a panel with xy-scatters plots between Single siRNA and SMARTpool Z-scores would help visualize the overall reproducibility of the screens.

Page 33 The quantification of cell migration regarding BUD31 and BPTF (re: inhibition) are less impressive than PRPF4B and MXD1 and beg further validation using CRISPR and shRNAs.

Page 34, Fig 4 Not clear why “osteoclast differentiation” and “tuberculosis” appear in the network analysis. Also, in 4D, where is the cluster 1 data (amongst the nine clusters)?

Page 35 Figure 5. Does the hierarchical clustering based on gene expression also cluster highly metastatic and non-metastatic tumor types?

Page 38 Supplementary Figure 1. Add number of targets for each of the siRNA libraries. The bottom half of the flowchart shows three target gene inputs and two output lists, one for each cell line. Consolidating the three input diagrams (left of the text Primary hits and selection of top hits) to only Hs587T (top 153 cell line specific + 129 overlapping hits); MDA-MB-231 (top 153 cell line specific + 129 overlapping hits) would make this figure have two inputs and two outputs, as opposed to three inputs and two outputs. Furthermore, the 129 overlapping genes don't match the text, which lists 145 significant hits.

Page 42 Supplementary Figure 5 would benefit from the addition of volcano plots (Log2 Fold Change vs $-\text{Log}_{10}P$) to illustrate the overall results of the differential expression analysis. Plots can be annotated with the current information being presented on panels B and C.

References for Specific Observations section

Nik-Zainal et al. Nature. 2016 Jun 2;534(7605):47-54.

Van der Weyden et al. Nature. 2017 Jan 12;541(7636):233-236.

Reviewer #3, Expertise: Splicing factors, breast cancer
(Remarks to the Author):

Fokkelman et al. describe the findings from a large-scale phenotypic imaging-based RNAi-screen to identify genes involved in the regulation of migratory phenotypes in two human triple negative breast cancer (TNBC) cell lines. Uncovering the determinant of metastatic dissemination and progression is highly needed to treat and prevent aggressive breast tumors such as TNBC. This study discovered novel regulators of cell migration in TNBC including the transcriptional modulator BPTF and the splicing factors PRPF4B and BUD31. Using in vitro and in vivo models, Fokkelman et al. further demonstrate the reducing PRPF4B levels inhibits cell migration and decreases the number of metastatic lesions, and could constitute a promising drug target for metastatic cancer. Findings from this study thus contribute to improving our understanding of the regulators of TNBC cell migration and highlight the role of splicing regulators in tumor progression.

However several questions should be addressed prior to publication, and should result in an enhanced manuscript.

Major comments:

1) The major weakness of this study is the lack of data measuring cell viability/proliferation following knock down of target genes. While the various assays used by Fokkelman et al. allow a thorough analysis of the different migratory phenotypes, none of them can distinguish an effect on cell viability/proliferation from an effect on cell migration. If a gene knockdown negatively impacted cell viability and promoted cell cycle arrest or cell death, the cells would not exhibit a migratory behavior in the time frame of the experiment, and would presumably be scored as negatively impacting migration. This should be addressed by the authors, or at least discussed,

especially given that a number of studies demonstrated that knockdown of RNA-binding proteins, and specifically splicing factors, blocks cell cycle progression and leads to cell death. In addition, the author achieve a PRPF4B knockdown efficiency of ~40% using shRNA in xenograft experiments, compared to an >80% efficiency using siRNAs. This suggests that cells that exhibit a high PRPF4B knockdown efficiency are eliminated when creating stable cell lines and do not give rise to tumors in xenograft experiments; highlighting the potential link between knockdown efficiency and cell viability.

2) Overall the description of the siRNA screen and downstream computational analysis need a more detailed description of the methodology, time points, and statistical analysis within the main text.

- Please add brief description of the siRNA library, e.g., how many siRNAs per target.

- The authors state "In total, 217 hits [from the primary screen] were validated in the Hs578T and 160 in the MDA-MB-231...". Please describe how many validated/tested, how many siRNA per target, what are the negative and positive controls and add statistics for the validation rate.

- The authors state "Annotation of protein classes for each set of validated hits (Hs578T, MDA-MB-231, and overlap) showed that most of the hits were transcription factors (Figure 2E i) also after correction for library size (Figure 2E ii)...". Please provide statistical analysis.

- The authors state "we used the larger lists of our PKT validated candidate genes to inform on protein-protein interaction (PPI) networks that are involved in Hs578T and MDA-MB-231 cell migration... [...] ". Please describe how many genes tested vs. number in background and type of statistical analysis.

- Regarding the TCGA analysis, the authors state "we identified clusters of candidate genes highly altered in multiple cancer types, amongst which breast cancer". Please describe how many genes tested vs. number in background and type of statistical analysis.

- RNA-seq analysis: please provide read depth and number of biological replicates, as well as time frame of the experiment.

3) The authors identify that BUD31 and PRP4B are each amplified in ~2% of breast tumors, whereas BPTF is amplified in ~8% tumors and correlate their expression with clinical phenotypes (Figure 5). This analysis would be greatly enhanced if the authors (i) focused their analysis specifically on TNBC vs. other breast cancer subtypes using available RNA-seq and DNA-seq data, (ii) analyzed paired primary and metastatic tumor samples using available RNA-seq and DNA-seq data, (iii) analyzed changes in RNA expression in addition to amplifications and mutations to determine if additional epigenetic or transcriptional control mechanisms can impact BUD31 PRP4B or BPTF expression even in absence of copy number changes or mutations. Finally, given that solid tumors exhibit a large number of genomic alterations, the analysis would be strengthened if the author compared TCGA data for negative candidates from their screen.

4) The authors analyzed the RNA-seq data to identify changes in splicing events following BUD31 or PRP4B siRNA-mediated KD, and focused specifically on intron retention events. We suggested using a computational pipeline more appropriate for intron retention detection, a challenging task that often generates many false positive, as well as including RT-PCR validation for several of these events. In addition, it is surprising that the authors were "unable to detect any changes in exon inclusion or 3' or 5' alternative splice site usage after knockdown of either PRPF4B or BUD31"; we suggest re-analyzing the data with computational pipeline dedicated to splicing analysis (e.g. MISO, rMATs, etc) that would allow to identify additional splicing event types, such as cassette exon, alternative acceptors and donors, to ensure that no changes are indeed detected. The author further focus on the differential genes decreased upon siRNA knockdown,

stating that these will be a direct consequence of increased intron retention – a logical assumption that should however be analyzed more thoroughly.

5) The authors further focus on the role of PRPF4B in vitro and in vivo using MDA-MB231 and its lung metastatic variant, MDA-MB-417.5. A few questions need to be addressed or at least discussed: (i) siRNA KD of PRPF4B in MDA-MB231 has an effect of the cell migratory phenotype, yet there are very few changes in splicing or gene expression compared to other siRNA (Supp. Figure 5 and Supp Figure 7) thus providing little explanation on what mediated the phenotype; (ii) is there a difference in PRPF4B expression MDA-MB231 and its lung metastatic variant, MDA-MB-417.5?; (iii) what is the effect of shPRPF4B KD on cell migration in MDA-MB-417.5?; (iv) an analysis of the splicing events in MDA-MB-417.5 shRNA PRPF4B used for the xenograft experiments would likely allow to identify the splicing events that are relevant to metastatic progression in vivo.

6) The authors demonstrate that shPRPF4B KD decreases the number of metastatic nodules in the lung as determined by macroscopic evaluation, as well as the total bioluminescence flux. Yet, in figure 7E and 7E, a few shPRPF4B animals still exhibit a similar number of metastasis to control animals. Have the authors collected tissues from these animals and verified if these metastatic lesions still express the shPRPF4B and exhibit lower levels of PRPF4B, or alternatively have escape the shRNA KD or silenced the shRNA? A more detailed analysis may uncover that the author underestimate the effect of PRPF4B KD.

7) A number of small molecule inhibitors of the splicing machinery have been tested in cancer cell lines. Please discuss if any of these have been shown to affect cell migration. How specific is the effect of PRPF4B and BUD31 vs. global inhibition of splicing, e.g. were other splicing factors tested in the screen?

Minor comments:

1) Introduction: please cite the following papers performing screens to identify genes involved in metastatic progression in breast cancer cell lines <https://www.ncbi.nlm.nih.gov/pubmed/29414946> <https://www.ncbi.nlm.nih.gov/pubmed/25855289>

2) Please add data for RNAi efficiency at the RNA and protein levels for at least BTPF, BUD31, PRPF4B in each of the cell lines. Please specific how many siRNA have been tested for each target.

3) The author use a platform that allows a very thorough analysis of the different migratory phenotypes that would be of great interest to the readers, yet the description of these phenotypes is very spare. Please expand the description in the text and supplemental figures. In addition, the author state that “a decrease in migration does not necessarily coincide with an overall change in cell morphology.” Please describe if the screen is corrected for differences in cell size. One can assume that larger cells would migrate further than smaller cells.

4) The author focus on a set of genes from the initial screen, including “some of which are directly involved with splicing (BUD31 and PRPF4B)”. However, we could not find a reference to BUD31 in the main figures Fig1-2 or corresponding supplemental material. Please explain or correct this discrepancy. Please add the phenotypes for BUD31 and BPTF to Figure 1.

5) BPTF Please cite the following paper relevant to BPTF in breast cancer: <https://www.ncbi.nlm.nih.gov/pubmed/28579392>

6) The authors state “Clustering of all genes involved in ECM receptor interaction (Fig. 6C, see Suppl. Fig. 9 for all gene names) or focal adhesion (Suppl.Fig. 10) demonstrated the involvement of many different pathway components of which some were overlapping between PRPF4B, BUD31

and BPTF (Fig. 6C and 6D); a similar downregulation was observed at the protein level for several key components in both cell lines (Fig. 6E, Suppl. Fig. 11C).” Please describe the components the authors are referring to and how they potentially contribute to increased metastatic dissemination.

7) The author state that “Our combined data indicate that the various candidate hits differentially affect cell morphology and migratory phenotypes, indicative of different genetic programs that define BC cell migration behavior”. Please discuss the correlation between cell adhesion vs. cell migration and its effect of metastasis in vivo? Discuss the different steps of the metastatic cascade, and how the differences in migratory phenotypes relate to them.

8) The authors state “The effects on differential expression of cell matrix adhesion components was also reflected in the different organization of focal adhesions”. Please describe the components the authors are referring to and how they potentially contribute to increased metastatic dissemination.

9) In the discussion: “Also decreased levels of PRPF4B almost completely eradicated spontaneous metastasis formation from the orthotopic primary tumor to distant organs...” Please replace “eradicated” with a more appropriate term, e.g. “prevented”

10) In the discussion, please add a paragraph about the link between splicing regulation and cell migration and metastatic potential.

Reviewer #1, Expertise: TNBC, metastasis

(Remarks to the Author):

This is an interesting manuscript on triple-negative breast cancer, looking at motility pathways to provide new insights into translatable pathways. TNBC is an orphan disease with no molecular therapeutics and a dearth of broadly expressed translatable pathways. Focusing on motility, the authors use an incredible array of bioinformatic analyses to distill down to a handful of genes, one of which is shown to regulate metastasis in vivo.

I have two major comments: First, this manuscript is written in a manner that a biologist cannot easily understand. I dug through the supplementary data, and still did not understand what some of the data meant. Second and more importantly, there is too little in vivo and mechanistic data. One gene is confirmed in one metastasis assay with only the barest of endpoints. Other types of motility assays are not performed.

We thank the reviewers for these remarks. Regarding the first comment, we went critically through the manuscript and updated and clarified the supplementary data information files. Changes in the manuscript are marked in yellow.

Regarding the second comment, our manuscript provides a legacy dataset on signaling components that modulate TNBC cell migration. In our opinion this dataset is unique in its kind and of high value for the broader research community. We acknowledge the comment that we have limited in vivo data, but, with all respect and as the reviewer may understand, validating all candidate hits is an enormous effort in itself, beyond the scope of this manuscript. However, as part of the revision we created stable knockdown cell lines for the transcription factor BPTF and tested its role in metastasis formation in vivo. The role of BUD31 has already been studied in vivo in a similar model elsewhere (Hsu et al, 2015). Moreover, we assessed the role of other main targets in other motility assays such as a Boyden Chamber and scratch assays. We anticipate that these new data provide further support for the validity of our screening data and validity on the impact of our data.

- Hsu et al. The spliceosome is a therapeutic vulnerability in MYC-driven cancer. Nature. 2015 Sep 17;525(7569):384-8.

Specific comments are listed below:

1. It is difficult from the main text to understand the screening assay. Is this directed or random motility? Would the results change if it is the other? How were the parameters selected, how did you come up with long smooth and long rough? A reader should not have to dig too deep into supplemental data to understand the basics of this assay.

We thank the reviewer for these suggestions and clear questions. Below, we would like to separately address these questions. Moreover, we adapted the materials and methods and results section to clarify the used methods in the main text.

- i. Is the phagokinetic track assay based on directed or random motility?

The phagokinetic track (PKT) assay has been used and published before by our lab (Fokkelman et al, 2016; van Roosmalen et al, 2015) and is an assay to detect random motility; we have published the details of these methods (van Roosmalen et al, 2011). Cells are seeded in a plate coated with beads,

in normal RPMI medium supplemented with FBS without any barrier. The beads are phagocytosed when the cells are migrating, resulting in tracks of which we analyze the phenotype. Candidates were validated with a live random cell migration assay (Figure 3), in which single cells are imaged and tracked over time.

- Fokkelman et al. Cellular adhesion screen identifies critical modulators of focal adhesion dynamics, cellular traction forces and cell migration behaviour. *Sci Rep.* 2016 Aug 17;6:31707.
- Van Roosmalen et al. Tumor cell migration screen identifies SRPK1 as breast cancer metastasis determinant. *J Clin Invest.* 2015 Apr;125(4):1648-64.
- van Roosmalen et al. Functional screening with a live cell imaging-based random cell migration assay. *Methods Mol Biol.* 2011;769:435-48.

ii. Would the results change if it is the other?

As suggested by the reviewer, we also tested the role of some of our main hits in directed cell migration by performing Boyden chamber assays in both Hs578T and MDA-MB-231 (Suppl. Fig. 10). Cells were starved for 6 hours in serum-free medium, plated in 0.3% FBS in a Boyden Chamber and directed cell migration towards 10% FBS medium was tested. The complete method is described in the supplemental methods section. In MDA-MB-231 knockdown of PRPF4B, BUD31 and BPTF all decreased directed cell migration, while in Hs578T only PRPF4B affected directed migration towards FBS. Therefore, we cannot claim that the results from our random cell migration screen can be translated to directed cell migration. Yet, knockdown of our main candidate PRPF4B inhibited cell migration irrespective of the mode of motility, suggesting that PRPF4B inhibits the migration machinery components that are critical for cell migration. Moreover, PRPF4B also affected spontaneous metastasis formation *in vivo*. These results are discussed in greater detail on page 8 and 14.

Supplemental Figure 10. Candidate knockdown effects on cell migration in scratch and Boyden Chamber assays. (E) Directed cell migration to FBS of candidate knockdowns in MDA-MB-231 using a Boyden Chamber assay. Migration is normalized to siKP knockdown. Bars show mean+sd of 2 biological replicates. (F) Same as in E, but for Hs578T. Significance was determined using ANOVA correcting for multiple testing. * $p < 0.05$, ** $p < 0.01$, *** $p < 0.001$.

- iii. How were the parameters selected, how did you come up with long smooth and long rough?

Phenotypic track parameters clearly related to migration were selected for analysis: net area, major and minor axis, axial ratio and the roughness. The net area and major axis of the track do provide information about the area covered and distance travelled by the cell. These are static measurements that are in general related to the dynamic migration speed of the cell. However, sometimes we observed that knockdown of candidate genes stopped random cell migration in conjunction with cell flattening. In this case, we can still observe an increase in net area, but we also observe a decrease in axial ratio because the tracks were more rounded. By taking along the axial ratio, we can discriminate between these two forms of migration inhibition. The roughness parameter does provide information about the migration mode. Especially the Hs578T cells form many protrusions when migrating, thereby creating tracks with a rough outline. Some knockdowns inhibited protrusion formation while not affecting migration, resulting in a smooth track phenotype. The definition of the track parameters is discussed in greater detail in the results and methods sections (page 5 and 22).

2. Is there a difference between cell shape and motility? Fig. 4D actually breaks down the genes by cell morphology. This is discussed briefly later in the manuscript, but the question remains.

Yes, there is a difference between track shape, motility and cell shape. Using the PKT assay, we measured the track shape as a measure of cell motility. In general these two are positively correlated (61% and 71% of the PKT candidates were validated in a random cell migration assay for Hs578T and MDA-MB-231, respectively, page 7). To investigate a correlation between decreased motility and cell shape, we performed a phenotypic screen using all PKT screen candidates (Figure 4). Here we did not observe a single cellular phenotype related to inhibition of migration, suggesting that cell shape and cellular motility are affected independently (discussed on page 10).

3. Given a 7 hr assay, what is the contribution of cell proliferation? Many labs use a 4 hr assay for MDA-MB-231 cells to eliminate that aspect. Cell cycle was a class of proteins in both cell lines (Fig. 4).

We choose to use a 7 hour assay to obtain a large enough window to detect significant differences in track areas for both MDA-MB-231 and Hs578T. Since these are two fast proliferating cell lines, some of the cells are indeed likely dividing during this assay period and this might affect the track area. To correct for this, we automatically removed all tracks containing multiple cells from the analysis during the image processing. In this way, we ensured that proliferation was not affecting the measured track area. Next to this, one can imagine that knockdown of cell cycle related genes results in stress and thereby affect migration. To determine whether proliferation was a confounder in our screen, we used the number of tracks per well (measure for proliferation) and plotted this against the used phenotypic track parameters (Suppl. Fig. 3). We did not observe any correlation between the track number and any of the phenotypic parameters, suggesting that candidates we selected in our primary screen were mainly based on effects on migration. This has been implemented in the manuscript on page 6, Suppl. Fig. 3.

Supplementary Figure 3. Proliferative effects of migration candidates primary PKT screen. (A) Relation between track parameter Z-scores and number of tracks imaged per well in MDA-MB-231. (B) Relation between track parameter Z-scores and number of tracks imaged per well in Hs578T. (C) Comparison of number of tracks for non-hits and hits in MDA-MB-231. (D) Comparison of number of tracks for non-hits and hits of the primary screen in Hs578T.

4. “we focused our screening effort on the complete set of cell signaling components, covering all kinases, phosphatases, (de)ubiquitinases, transcription factors, G-protein coupled receptors, epigenetic regulators and cell adhesion-related molecules (4198 individual target genes in total).” Does this include the cytoskeletal proteins? If not, why? Could they not provide a number of feedback regulatory loops?

Many cytoskeletal genes were included in our primary screen, some of which are modulators of the actin cytoskeletal, e.g. RhoGTPases, GEFs and GAPs and part of the sub-library of ‘cell adhesion-related molecules’. In Suppl. Fig. 2, we highlighted the effect of knockdown of genes in the KEGG pathway ‘Regulation of the actin cytoskeleton’ in our primary screen. Although some of the

cytoskeletal genes affect track phenotype and cell migration (mainly in Hs578T, Suppl. Fig. 2A) and some of them were selected as migration candidates out of the primary screen, we observed no significant difference in phenotypic parameters comparing cytoskeletal genes with non-cytoskeletal genes (Suppl. Fig. 2B-C). Still, some of these cytoskeletal genes can be involved in regulatory feedback loops for our candidates. For example, various cell adhesion-related genes were downregulated upon knockdown of PRPF4B, BPTF and BUD31 and also affected cell migration of our cells in the primary screen (Figure 6D). This is included in the manuscript on page 6.

Supplemental Figure 2. Cytoskeletal genes in the primary PKT screen. (A) Primary screen Z-scores of genes in the KEGG pathway “Regulation of the actin cytoskeleton”. Red = hit in Hs578T, blue = hit in MDA-MB-231, green = hit in both cell lines. (B) Z-score distribution of genes that are in the “Regulation of the actin cytoskeleton” pathway or other genes for Hs578T. (C) Same as in B, but for MDA-MB-231.

5. Can the authors test some of these genes in traditional Boyden chamber or scratch migration assays so that the field can understand the overlap?

According to the reviewer’s suggestion, as a proof of confidence that our candidate genes that affect cell migration in other assay conditions, we tested some of our main candidates in traditional Boyden chamber and scratch migration assays. Knockdown of all our tested candidates, except for CCNT2 in Hs578T, also inhibited cell migration in scratch migration assays (Supp. Fig. 10). For the directed cell migration (discussed before in point 1), all tested candidates did inhibit directed cell migration in MDA-MB-231, while only PRPF4B knockdown also affected directed cell migration in Hs578T (Supp.

Fig. 10). This supports that our screen provides a robust list of candidates that inhibit random cell migration independent of the assay used, but at this moment cannot be used to predict effects on directed cell migration in Hs578T cells. Importantly, these new experiments provide additional support for the effect of PRPF4B on cell motility of TNBC cell lines. These results are implemented in the manuscript on page 8 and 14.

Supplemental Figure 10. Candidate knockdown effects on cell migration in scratch and Boyden Chamber assays. (A) Wound size 0 and 20 hours after scratch preparation in MDA-MB-231 after candidate knockdown. (B)

Quantification of A. Bars represent mean+sd, n = 4. (C) Wound size 0 and 6 hours after scratch preparation in Hs578T after candidate knockdown. (D) Quantification of A. Bars represent mean+sd, n = 4. (E) Directed cell migration to FBS of candidate knockdowns in MDA-MB-231 using a Boyden Chamber assay. Migration is normalized to siKP knockdown. Bars show mean+sd of 2 biological replicates. (F) Same as in E, but for Hs578T. Significance was determined using ANOVA correcting for multiple testing. * p < 0.05, ** p < 0.01, *** p < 0.001.

6. I am unsure of the significance of the interaction map on Fig. 4C. It looks like red lines go different places than blue or green, but apparently “connect” as shown on the graph to the right? The main connectors are HDACs. Wouldn't that be the case for sets of genes associated with many phenotypes?

The line color in the interaction map on Figure 4C is purely chosen for visualization. All interactions are displayed in grey, except for the connections between central hubs (green), candidates in both cell lines (red) or clinically relevant genes (blue). When a clinically relevant gene was connected with a hub, the line color was randomly picked. We do not claim that HDAC connectors are unique for our phenotype. Indeed, given that HDACs are involved in chromatin remodeling and likely affect the expression of many genes, it is not unlikely that they are involved in other phenotypes.

7. How different would this look if motility was directed to a particular attractant? The SMADs are listed prominently in Fig. 4. Is TGF- β driving this or are SMADs crucial to random migration?

All migration assays in this study were performed in RPMI-1640 medium (Gibco) supplemented with 10% FBS, 25 IU/ml penicillin and 25 μ g/ml streptomycin without addition of extra growth factors, and suggested that SMADs are crucial for random cell migration. To exclude the role of TGF- β in this process, we now have exposed MDA-MB-231 and Hs578T cells to 10 ng/mL TGF- β in starved conditions (serum-free medium). In contrast to EGF treatment, TGF- β treatment did not increase migration speed (Suppl. Fig. 12) in both MDA-MB-231 and Hs578T cell lines. This suggests that migration is not dependent on TGF- β and that SMAD knockdown in itself is modulating random cell migration, likely through modulating SMAD-depending transcriptional programs of which some affect cell motility. These results are discussed on page 9 of the manuscript. Methods are described on page 23.

Supplemental Figure 12. Effect of TGF-beta treatment on TNBC cell migration. Random cell migration speed of MDA-MB-231 or Hs578T cells after 2 hours starvation, followed by EGF or TGF treatment. Average and standard deviation of two independent replicates are shown. Significance was determined using ANOVA correcting for multiple testing. * $p < 0.05$.

8. On Fig. 5, Kaplan-Meier curves are shown for genes and particular subtypes of breast cancer. Were they the only subtypes with significant associations?

According to the reviewer's suggestion, we investigated the associations for the main candidates PRPF4B, BUD31 and BPTF for all tumors irrespective of the subtype as well as for the subtypes complementary to Fig. 5 (Suppl. Fig. 17). For PRPF4B, gene expression is related to metastasis-free survival taking into account all tumors (Suppl. Fig. 17) and TNBC (Fig. 5), but not in ER positive tumors (Suppl. Fig. 17), suggesting that PRPF4B is mainly important in metastasis formation in TNBC. BUD31 expression is significantly associated with metastasis-free survival in all tumors and ER positive tumors, but not in ER negative tumors (Fig. 5 and Suppl. Fig. 17), suggesting that BUD31 is mainly important in metastasis formation in ER positive tumors. BPTF is associated with metastasis-free survival independent of the subtype (Fig. 5 and Suppl. Fig. 17). These results are discussed in the manuscript on page 11.

Supplemental Figure 17. Candidate modulators of TNBC cell migration are related to distant metastasis-free survival in breast cancer patients. Kaplan Meier curves for expression of PRPF4B, BUD31 and BPTF and relation to metastasis-free survival in All, ERpos or ERneg breast cancer patients.

9. Does it matter if gene knockdowns are performed on cells in tissue culture versus on ECM, as in a Boyden chamber assay?

All our migration experiments were performed on plates coated with fibronectin as discussed in the results and methods section (page 5 and 23). To investigate the role of the ECM in the migratory behavior of the Hs578T and MDA-MB-231 cell lines, we performed live cell migration assays for cells plated on normal tissue culture plates or tissue culture plates coated with collagen or fibronectin. Plate coating with fibronectin and collagen significantly increased the migration speed in both MDA-MB-231 and Hs578T cell lines (Suppl. Fig. 31). This suggests that the fibronectin coating we used in our screen could enlarge the number of candidates by increasing the screening window. Moreover candidates specifically affecting ECM migratory pathways might only be selected when tissue plate coating is applied, thereby resulting in different candidate lists. Since ECM is a major component of the local tumor microenvironment (Lu et al, 2012), we believe that plate coating in vitro might be important to increase clinical translation of the identified candidates. This is further discussed in the manuscript on page 17.

Supplemental Figure 31. Migration speed on different cell coatings in MDA-MB-231 and Hs578T. Average + stdev of 2 biological replicates, significance was calculated using ANOVA with multiple testing correction (* $p < 0.05$, ** $p < 0.01$).

- Lu et al. The extracellular matrix: a dynamic niche in cancer progression. *J Cell Biol.* 2012 Feb 20;196(4):395-406.

10. Many gene signatures in cancer and metastasis are replete with ECM alterations. Were other classes of altered genes examined?

Next to ECM alterations, also TNF- α signaling was significantly downregulated upon knockdown of PRPF4B, BPTF and BUD31 (Fig. 6A and Suppl. Fig. 24). Expression levels of CSF1, CSF2, IL6, MMP3 and PIK3CD were validated with RT-qPCR in knockdown as well as control conditions in MDA-MB-231 (Suppl. Fig. 24), suggesting that multiple altered gene signatures identified by next generation sequencing can be validated using RT-qPCR. These results are discussed in the manuscript on page 13.

Supplemental Figure 24. Candidate knockdown results in deregulated TNF-alpha signaling. Effect of candidate knockdown on TNF-alpha signaling components CSF1, CSF2, IL6, MMP3 and PIK3CD in next generation sequencing (top) and RT-qPCR (bottom). Mean + stdev of 3 biologically independent replicates. Significance was determined using ANOVA correcting for multiple testing. * $p < 0.05$, ** $p < 0.01$, *** $p < 0.001$.

11. This manuscript is a tour-de-force in bioinformatics, but is lean on functional data. In vivo data for one gene and one cell line are presented. There is no mechanistic evaluation- for instance, were the primary tumors more or less invasive? Where was PRPF4B expressed- the center of the tumor versus the invasive front? How about a tail vein assay where initial extravasation is omitted and an impact on colonization can be seen? At a minimum, two models should be shown in vivo, and the other genes should be tested.

We did not evaluate the invasiveness of the primary tumor in detail. When surgically removing the primary tumor, we did not notice a difference in tumor cell invasion, neither was there a significant difference in the number of tumors growing in the muscle layer. To identify the differences in PRPF4B expression levels across the tumor, we stained cross-sections of the control primary tumor for PRPF4B expression. Interestingly, the number of PRPF4B positive cells as well as the PRPF4B intensity is higher at the border compared to the middle of the tumor, suggesting that PRPF4B might play a role in tumor invasion (Supp. Fig. 30). This has been discussed in the manuscript on page 14.

Supplemental Figure 30. PRPF4B expression in primary mouse tumors. (A) PRPF4B expression in the middle and border of primary breast tumors from of three representative mice injected with LM2 shCtrl cells. Percentage of PRPF4B positive cells (B) and PRPF4B intensity (C) in the middle and border of these tumors. Paired samples show the same color. Significance was determined using a Wilcoxon test. * $p < 0.05$, ** $p < 0.01$.

Next the reviewer proposed to study the role of other main candidates in vivo. Given the RNAseq analysis for PRPF4B, BUD31 and BPTF, we focused on these targets. We already demonstrated the effect of PRPF4B in our previous submission. The role of BUD31 in breast cancer metastasis formation has already been demonstrated in the LM2 cells by others (Hsu et al, 2015). Therefore we tested the in vivo capability of BPTF knockdown to inhibit metastasis formation. For two out of four different shRNA against BPTF we successfully generated stable BPTF knockdown LM2 cell lines that showed decreased cell migration (see figure below). Unexpectedly, these cell lines did behave different in vivo. Whereas LM2 shBPTF #2 demonstrated decreased numbers of lung metastases (see figure below), LM2 shBPTF #1 did not demonstrate this effect. Remarkably, for LM2 shBPTF #1 cells the BLI photon flux was almost ~ 100 higher than for LM2 shBPTF #2 and shCtrl cells, which was associated with increased spleen weight (see figure below). This is suggestive of possible enhanced systemic pro-inflammatory responses. In all the experiments we have performed in the past with this LM2 in vivo metastasis model, we have never seen such a response for any of the targets. Currently we can only speculate about potential off-target effects. Given the uncertainty of the conclusion of these experiments we decided not to include this data in the manuscript resubmission.

BPTF knockdown cell line and metastasis formation. (A) BPTF expression in LM2 sh cell lines. (B) Wound closure of LM2 sh cell lines. (C) Number of lung metastases in mice injected with LM2 sh cell lines. Total flux of BLI measurements in mice injected with LM2 sh cell lines in the (D) lungs, (E) spleen and (F) lymph node.

- Hsu et al. The spliceosome is a therapeutic vulnerability in MYC-driven cancer. Nature. 2015 Sep 17;525(7569):384-8.

Reviewer #2, Expertise: RNAi screen

(Remarks to the Author):

Enclosed is a review of the manuscript. Overall it's a solid experiment, quite an interesting one as well. There are some limitations as pointed out in the critique, however, the manuscript left very good impression of the work. Additional experiments at the lab bench are minimal, however, a reanalysis of the TCGA data, applying elements of experimental design, is heavily advisable.

Review of "Uncovering the signaling landscape controlling breast cancer cell migration identifies novel metastasis drivers" by Fokkelman, et al. The article presented by Fokkelman et al. is an elegant experiment where an exhaustive screen of 4200 target genes involved in cell signaling was carried out to ascertain their impact on cell migration related traits. PKT = phagokinetic track assay. Through multiple rounds of experimental validation by SMARTpool, siRNA, live-cell microscopy in GFP-expressing cells, and the analysis of RNA-seq gene expression data the authors consistently replicated their initial findings by RNAi, culminating in an in-vivo experiment that clearly exemplifies their top candidate PRPF4B as a driver of metastasis in breast cancer.

Comments and Suggestions:

This study made good use of high-content imaging, coupled with siRNA screening to measure the phenotypic effects on cell migration and morphology as it relates to metastatic breast cancer. There are a number of issues that need to be addressed from the point of view of high-content imaging and informatics.

1. In the identification of hits for migration using the PKT assay, there needs to be some analysis of viability as a result of gene expression knockdown. This would ideally be the result of measuring the cell number following siRNA treatment and prior to seeding in bead plates, or from published data in these cell lines.

The reviewer is correct that effects on viability can affect the screening results. In our initial screening analysis we took effects on viability into account. We anticipated that genes that strongly affect viability and/or proliferation as a consequence of 3 day siRNA treatment before the migration assay would result in a limited number of individual cell migration tracks in the PKT assay, i.e. tracks of the remaining viable cells. To remove the genes that caused extensive inhibition of proliferation and/or cell death, we removed the wells with very low track numbers (< 60 for Hs578T and < 150 for MDA-MB-231) for further analysis. However, we agree that the remaining results could still be influenced by effects on proliferation. Therefore, to address the effect of knockdowns on proliferation, we used the number of tracks per well as a measure for proliferation and plotted this against the used phenotypic track parameters (Suppl. Fig. 3). We did not observe any correlation between the track number and any of the phenotypic parameters, suggesting that candidates we selected in our primary screen were mainly based on effects on cell migration. This has been implemented in the manuscript on page 6, Suppl. Fig. 3.

Supplementary Figure 3. Proliferative effects of migration candidates primary PKT screen. (A) Relation between track parameter Z-scores and number of tracks imaged per well in MDA-MB-231. (B) Relation between track parameter Z-scores and number of tracks imaged per well in Hs578T. (C) Comparison of number of tracks for non-hits and hits in MDA-MB-231. (D) Comparison of number of tracks for non-hits and hits of the primary screen in Hs578T.

2. The principal component analysis performed on the small number of measures is probably insufficient to describe differences if viability measures are not included.

We used PCA to highlight the differences in track phenotypes in one overview by a combination of multiple parameters. We have not used the viability (i.e. track number) in this PCA, since this is not a descriptor of the phenotype. As discussed above, overall cell health could affect the cellular phenotype, but again, for this graph, the knockdowns heavily affecting proliferation were already removed (low track/image number). We already have evaluated that differences in indirect measures of viability (i.e. track number) do not affect a specific phenotypic parameter. Since our goal for this PCA was to highlight the different migratory phenotypes, we chose to leave out viability measurements. For all further filtering steps in our secondary and tertiary screenings we removed knockdowns heavily affecting cell number. Important to note is that knockdown of our main

candidate gene tested in vivo, PRPF4B, did not affect primary tumor growth but did reduce metastasis formation.

3. This type of screen has been performed previously for morphology and migration, so this aspect of the study is not particularly novel.

1: Pascual-Vargas P, Cooper S, Sero J, Bousgouni V, Arias-Garcia M, Bakal C. RNAi screens for Rho GTPase regulators of cell shape and YAP/TAZ localisation in triple negative breast cancer. *Sci Data*. 2017 Mar 1;4:170018. doi: 10.1038/sdata.2017.18. PubMed PMID: 28248929; PubMed Central PMCID: PMC5332010.

2: Taniguchi H, Hoshino D, Moriya C, Zembutsu H, Nishiyama N, Yamamoto H, Kataoka K, Imai K. Silencing PRDM14 expression by an innovative RNAi therapy inhibits stemness, tumorigenicity, and metastasis of breast cancer. *Oncotarget*. 2017 Jul 18;8(29):46856-46874. doi: 10.18632/oncotarget.16776. PubMed PMID: 28423353; PubMed Central PMCID: PMC5564528.

3: Williams SP, Gould CM, Nowell CJ, Karnezis T, Achen MG, Simpson KJ, Stacker SA. Systematic high-content genome-wide RNAi screens of endothelial cell migration and morphology. *Sci Data*. 2017 Mar 1;4:170009. doi: 10.1038/sdata.2017.9. PubMed PMID: 28248931

We think the novelty of this study is that the screen is specifically focused on uncovering new targets for triple-negative breast cancer (TNBC) in combination with conducting a genome-wide cell signaling screen covering ~4,200 target genes. Since TNBC cell lines migrate significantly faster than luminal breast cancer cell lines (Rogkoti et al, manuscript in preparation), inhibiting migration in the TNBC subtype might be of ultimate therapeutic interest. Pascual-Vargas et al only focused on the role of RhoGEFs and RhoGAPs in TNBC; we took a genome-wide approach to also identify targets that were not linked to migration before. Taniguchi et al only investigated one migration target: PR domain zinc finger protein 14 (PRDM14). Williams et al performed a genome-wide RNAi screen in endothelial cells, but these cells show a completely different mode of migration compared to TNBC (Pandya et al, 2017; Michaelis, 2014) and would therefore be hard to directly extrapolate to breast cancer. Moreover, Williams et al investigated collected cell migration using a wound healing assay, while our study focused on single cell migration.

- Pandya et al. Modes of invasion during tumour dissemination. *Mol Oncol*. 2017 Jan;11(1):5-27.
- Michaelis. Mechanisms of endothelial cell migration. *Cell Mol Life Sci*. 2014 Nov;71(21):4131-48.

The results of the study show a bias where the validation experiments preferentially replicated targets conferring a reduced cell migration phenotype. The bias could arise from limitations in the methods as they only test knockout and knockdown expression of initial targets deemed significant. The other alternative is from limitations in the cell lines chosen. Assuming the latter, it is possible that knockdown and knockout of targets resulting in increased cell motility were masked by the variants that increase cell migration already present in Hs578T and MDA-MB-231. Thus bringing the phenotype to a physiological ceiling, and consequently reducing the sensitivity to identified targets that upon inhibition increase cell migration. The use of a lower motility cell line in this study would

provide a platform with an increased sensitivity to identify suppressors of cell migration i.e. inhibition of the target increases cell migration as the genetic background would be devoid of cell migration promoting alleles. These limitations should be discussed in greater detail because they open opportunities for additional screens that can elucidate in the landscape of metastasis drivers.

The reviewer is correct in the analysis. We purposely have used TNBC cell lines that are highly motile to identify genes that in both cell lines demonstrate an inhibition of cell migration upon knockdown. We therefore indeed may have missed genes that can even further speed up migration of these cells if they are reaching their maximal capacity. In the context of understanding cancer metastasis, including slow migrating cell lines would create the opportunity to identify genes that act as suppressors of cell motility. Nevertheless, our ultimate attempt is to identify genes that can be used as drug targets to inhibit cancer dissemination. Therefore, we decided that including low migrating cells would impact on the complexity of our study and affect the overall clarity of the results. The limitations of this bias in our study are discussed in greater detail on page 16.

There was little discussion on the validated targets from the cell-line specific genes. These cell line specific genes may operate through an epistatic interaction with mutated genes specific to the cell line. Were there loss of function (LOF) or activating mutations, or copy number alterations on genes in the pathways related to the cell line specific targets?

This is a highly interesting suggestion from the reviewer. According to the reviewer's suggestion we investigated the mutation status, copy number variation and RNA expression levels in MDA-MB-231 and Hs578T cell lines for the PKT screen validated candidates. We did not observe an enrichment for mutations or copy numbers or differences in RNA expression levels comparing cell line specific candidates to candidates effective in both cell lines (Suppl. Fig. 7). Thus the cell line specific candidates might be related to cell line-specific dependencies or differences in migration modes, but cannot directly be explained by general differences in mutations, copy numbers or RNA expression levels of candidate genes. These results are discussed in the manuscript on page 6.

Supplemental Figure 7. Validated hits in Hs578T and MDA-MB-231 and cell line specific mutations, copy number alterations and differences in RNA expression levels. Candidates are shown in cluster order, color indicates their mutation or copy number states in at least one of the cell lines (blue = mutated, yellow = copy

number gain, brown = high copy number gain, light green = copy number loss, dark green = homozygous copy number loss).

There is a leap of causality in the manuscript by specifically highlighting that genes affecting cell migration traits will drive a metastatic phenotype in the tumor. This was primarily based on the gene expression correlations of validated targets to genomic alteration in different tumor types. However, there is no statistical test or any other evidence to each gene on their metastatic potential, outside of the in-vivo validation of PRPF4B. These arguments should be further supported or discussed in greater detail.

The reviewer is correct that not all of our candidate genes may be important for a metastatic phenotype of the tumor, because naturally there is a higher level of biology in the tumor microenvironment that will determine metastasis formation. Indeed we have only validated PRPF4B in our hands. Yet others have established that some of our strong candidate genes do affect metastasis formation in vivo, e.g. ITGB1, TIAM1 and BUD31 (Yin et al, 2016; Xu et al, 2016; Hsu et al, 2015). The correlation with metastasis free survival adds additional evidence that our candidate genes are likely involved in cancer metastasis in patients, but these associations do not prove a causal relationship. We have softened the conclusions in our manuscript (page 19).

- Yin et al. β 1 Integrin as a Prognostic and Predictive Marker in Triple-Negative Breast Cancer. *Int J Mol Sci.* 2016 Sep; 17(9): 1432.
- Xu et al. The fibroblast Tiam1-osteopontin pathway modulates breast cancer invasion and metastasis. *Breast Cancer Res.* 2016; 18: 14.
- Hsu et al. The spliceosome is a therapeutic vulnerability in MYC-driven cancer. *Nature.* 2015 Sep 17;525(7569):384-8.

The use of external data sources to identify supporting evidence for candidate targets is not well described in the methods. There was no description of how the cancer types were partitioned to identify supporting evidence for each gene. For example, cancer types can be divided by their metastatic potential, as highly metastatic and non-metastatic, then carry out a burden test between the two groups asking if the frequency of LOF, activating, or other genetic alterations different between the two groups? Or for differential gene expression; where the candidate genes over- or under-expressed in highly metastatic tumor types vs non-metastatic? Furthermore, clinical data for the patients on TCGA is available. This contains the AJCC Metastasis Pathologic stages, metastatic site, and metastatic tumor, which can be used to partition the samples to perform burden and differential expression comparisons. There is no mention of the project# to access TCGA data.

We thank the reviewer for these suggestions. For the Kaplan Meier curves and clinical evidence, tumors were divided based on subtype determined by immunohistochemical stainings of the estrogen receptor, progesterone receptor and HER2 receptor of primary tumor material using the following datasets as indicated in the methods section on page 26: GSE5327 and GSE2034. We choose for this dataset to exclude potential confounders such as treatment and lymph node status, while still having a big patient cohort to perform statistics.

According to the reviewer's suggestions, we examined CN variations, mutations and RNA expression levels in primary tumors with and without distant metastases at diagnosis for the candidates shown in Figure 5B. We did not observe a significant difference in RNA expression levels, mutations and deletions (Suppl. Fig. 15). Yet, interestingly, there were many candidates bearing extra amplifications in the primary tumors that already metastasized to distant organs (Suppl. Fig. 15). Although the group of primary tumors with metastases was rather small (22 tumors in total), this suggests that amplifications of these genes might be related to metastasis formation. This has been further discussed in the manuscript on page 11.

Supplemental Figure 11. Candidate alterations in primary tumors with and without metastasis at diagnosis (A) Log₂ RNA expression levels of candidates in primary tumors with (green) or without (red) distant metastases at diagnosis. (B) Candidate mutation rate (i), amplification rate (ii) and deletion rate (iii) in primary tumors with (green) or without (red) metastases at diagnosis.

Additional data from whole genome sequencing data of 560 genomes is publicly available by Nik-Zainal et al. (2016), ENA:EGAS00001001178. It would be worthwhile to investigate if the burden tests proposed above can also be carried out using this data set. The independent cohort would

strengthen the author's findings. Lastly, Van der Weyden et al. (2017) carried out an in-vivo screen for host-regulators of metastases in the mouse. Were any of the 217 primary hits present in their significant host-regulators?

As suggested by the reviewer we have carefully looked into the Zainal et al. dataset. Unfortunately, the 560 genomes dataset lacks the appropriate clinical data to carry out these analyses. The relapse information was only available for a small subset of the patients and the follow-up time was short in general. Dividing the patient cohort based on 5-year relapse results in 8 samples without and 59 samples with relapse, which is too little to perform statistical analysis.

Van der Weyden et al. Identified 23 validated host-regulators of metastases in the mouse. 7 out of these 23 genes were in our library (IRF1, RNF10, PIK3CG, IRF7, BACH2, ARHGEF1 and FBXO7). None of these genes was a validated candidate in our screen. This is not an unexpected observation given the different cell types used by Van der Weyden and the complexity of underpinning the exact biological step that is the critical biological determinant affected by each of these 7 genes. This underscores the importance to use phenotypic screens to unravel the various components that drive critical cancer cell phenotypes that are hallmarks of cancer progression. We have discussed these differences in our revised manuscript on page 17.

The analysis of Human Invasive Signatures, Lung Metastasis Signatures and the list of 440 genes from Rogkoti et al, (unpublished) is controversial because the results are enriched in the unpublished data set. It is unclear the extent of the strength of evidence this data set is providing relative to the rest. Do the results change if this set is taken out?

The results for the other signatures will not change if the 440 gene set is left out of the analysis. The first order PPI networks were generated separately for all signatures after which the overlap for all possible combinations of PPI networks was examined. The chord diagrams in Figure 5A and Suppl. Fig. 11B show the overlap between the different PPI networks. This is explained in more detail in the methods section on page 25.

A careful revision for clarity is needed in the results section. The manuscript conveys the complete scope of the experiment. However, the results sections are not woven together in a concise manner making them difficult to follow. The section titled "Transcriptional determinants are critical drivers of BC migratory phenotypes" was the most difficult to read as the list is narrowed down in the validation process. These sections are missing a details such as the sum total of 2807 primary hits (1501 + 1306), and the exact number of candidates used in the validations in each of the two cell lines as 298 targets, composed of the 145 overlapping and 153 cell-line specific (or 451 unique targets composed of 129 overlapping +153 Hs578T + 153 MDA-MB-231). The sections "Functional drivers of tumor cell migration partner in networks predictive for BC progression" and "Modulators of cell migration are associated with BC metastasis-free survival" continue to narrow down the initial list the lists of 298 targets per cell line (or the 451 unique targets) to those that validated in SMARTpool, siRNA, and GFP+ live cell imaging. However, the numbers are not consistently clear throughout these sections.

We thank the reviewer for pointing out this unclarity. In the section titled "Transcriptional determinants are critical drivers of BC migratory phenotypes " we incorporated the total number of

hits and further specified the numbers used for validation (page 5 and 6). We also incorporated and explained these numbers in the paragraphs “Functional drivers of tumor cell migration partner in networks predictive for BC progression” and “Modulators of cell migration are associated with BC metastasis-free survival”.

Specific observations

Page 5 line 22 One would wish to see the key hits of the focused screen validated by shRNA and sgRNA CRISPR-Cas9 methodologies. These cross-checks are critical for molecular target confirmation experiments and rather straightforward to do.

According to the reviewer’s suggestions, we generated a doxycycline inducible CRISPR-Cas9 cell line from the MDA-MB-231 cell line. Lentiviral sgRNAs against PRPF4B, BPTF and BUD31 were introduced in this cell line and the effects on migration were tested 72 hours after Cas9 induction. More details about the procedure can be found in the supplemental methods section. Three days after Cas9 induction, the live cell migration speed was reduced for both sgRNAs for all three tested targets, validating the role of BPTF, BUD31 and PRPF4B in cell migration (Suppl. Fig. 8). These results are discussed on page 7.

Supplemental Figure 8. Candidate CRISPR-Cas9 knockout and effect on live cell migration. (A) PRPF4B knockout efficiency 48 hours after Cas9 induction. (B) Migration speed of MDA-MB-231 ind-Cas9 72 hours after doxycycline exposure. Experiment was performed in biological triplicates, significance was calculated using ANOVA with multiple testing correction. * $p < 0.05$, ** $p < 0.01$, *** $p < 0.001$

Page 6, line 7 How much of the “reduced cell migration” was due to cytotoxicity rather than specifically impacting candidate genes involved in the process of cell migration? A generalized cytotox counterscreen would be appropriate, even in the same two cell backgrounds.

We have taken cytotoxicity into account in our analysis of the screening data. The number of tracks in our PKT assay is a direct representative of the cytotoxicity of siRNA knockdown and reflecting both effects on cell proliferation and survival. In our screening assays we have determined all the tracks per siRNA treatment from replicate experiments. Hence, as such our PKT screen setup indirectly already integrates a cytotoxicity counter screen. In our screening setup we removed all wells showing less than 150 tracks and 60 tracks from further analysis for MDA-MB-231 and Hs578T, respectively. As already discussed above, we could not observe a correlation between track number and migration

phenotypic parameters. Moreover, the stable knockdown cell lines for PRPF4B did not affect proliferation *in vivo* (Supp. Fig. 29). Although we cannot exclude that some of our candidates reduced cell migration due to some effects on overall cell health, we have confidence that the majority of our candidates is particularly affecting cell migration. These results have been discussed in greater detail on page 6 in the manuscript.

Page 8, second paragraph The tie-in between morphological changes and migration, to differential genetic backgrounds, particularly with respect to cell protrusions (clusters 4, 5, 7 and 8) is rather weak. One may consider a RNAseq of cluster 7 versus the other three groups (4, 7, 8) to tease-out these differences.

We thank the reviewer for this suggestion and agree that the effect on cell protrusions is maybe not the strongest. Furthermore, migratory behavior seemed to be very weakly related to cellular phenotype in this cell line as pointed out on page 9 and 10. This can also be observed by examining cluster 9, where many candidates strongly inhibiting migration in the PKT assay cluster together with the negative controls. Altogether, we feel that the integration of the RNAseq data with the complex relation between migratory behavior and cell phenotype is out of the scope of our already extensive study and might be better addressed in a separate manuscript.

Page 9, second paragraph: Why was GRK1 not mentioned in the text amongst the “top key factors”? Also, two of the top three hits were splicing factors, yet the “spliceosome” *per se* was not one of the focused libraries (see Suppl Fig 1, upper left).

We thank the reviewer for pointing out this inconsistency. GRK1 is now also mentioned in the text (page 10). Indeed, the spliceosome was not one of the focused libraries and therefore only partly represented in our screen. Therefore it is even more striking that multiple splicing factors were identified as main hits in the study, resulting in the decision to focus on these factors.

Page 12 and Fig 7A. Related to the “essentiality” of PRPF4B: Why did this gene hit in particular rise in importance above the other quality hits? The 40% knockdown (KD) is statistically significant but on a pure level of KD basis, rather unimpressive. This may suggest that in addition to impairing the function of “general metastasis”, there may be other splicing factors or off-target effects at play. The line of reasoning, and questions about it, continue in the first paragraph on page 15.

Since splicing factors were over-represented in the top candidates and PRPF4B has not been related to TNBC cell migration before (in contrary to BUD31) (Hsu et al, 2015) we decided to select this factor for *in vivo* validation. We agree that 40% knockdown on itself is not very convincing. However, this relatively low knockdown efficiency still results in decreased metastasis formation. Moreover, we confirmed high transient PRPF4B knockdown efficiency (Suppl. Fig. 19) and Cas9 knockout efficiency (Supp. Fig. 8), both resulting in decreased migration speed in *in vitro* experiments. Altogether, this confirmed the role of PRPF4B in migration and metastasis formation.

- Hsu et al. The spliceosome is a therapeutic vulnerability in MYC-driven cancer. Nature. 2015 Sep 17;525(7569):384-8.

Page 13 The initial sentence and paragraph of the Discussion should recapitulate the most important findings of the results. This does not occur until the second paragraph, in a way hiding the main results. Reorganizing the Discussion to flow from specific findings to broad discussion would improve the readability of this section and emphasize the main results in concise manner.

We thank the reviewer for this suggestion. The discussion section has been reorganized and the main results are now discussed in the first paragraph, after which the results are broadly discussed.

Page 19, line 18 “microscope using” (make into separate the words)

We thank the reviewer for noticing this mistake. The spelling error has been corrected.

Page 28 typo in the Science reference #24.

We thank the reviewer for noticing this mistake. The spelling error has been corrected.

Page 29, reference #49 Was this cited? Regrets if I did not see it in the text.

Reference 49 has been cited in the Materials and Methods section, sub-header Phagokinetic track (PKT) assay, page 21.

Page 31, Figure 1G Not much overlap between the hits from both cells lines, which makes one wonder if a third cell line is needed to clarify the results? See also page 40 Suppl Figure 3A for more non-similarity between Hs578T and MDA-MB-231.

We agree with the reviewer that hits in the primary screen show quite specific effects in each cell lines. Since we purposely selected two highly motile TNBC cell lines that have two different forms of migratory behavior, the large difference in primary hits was not unexpected; we have now pointed this out in the discussion (page 17). We consciously choose these cell lines to capture the diversity in TNBC migration patterns and identify candidates important in a broader TNBC spectrum. To clarify the results in different TNBC cell lines, we performed live cell migration assays for our main targets PRPF4B, BUD31 and BPTF in two different TNBC cell lines (HCC1806 and HCC38, Suppl. Fig. 9). Also in these two additional cell lines all tested candidates significantly reduced migration speed compared to control knockdown, suggesting that the candidates we identified are involved in TNBC cell migration irrespective of the cell line or migration mode. These results were implemented in the results section on page 8.

Supplemental Figure 9. Effect of PRPF4B, BUD31 and BPTF knockdown on live cell migration in HCC1806 and HCC38. Mean + sdev of two biological replicates. ANOVA with multiple testing correction was used for statistical analysis. * $p < 0.05$, ** $p < 0.01$, *** $p < 0.001$.

Page 32 In Figure 2 and Supplementary Figure 2, the plots of panel B obfuscate the message of reproducibility. Adding a panel with xy-scatters plots between Single siRNA and SMARTpool Z-scores would help visualize the overall reproducibility of the screens.

In Figure 2 and Supplementary Figure 4 (was Suppl. Fig. 2 in the initial submission) we show the effect of single and smartpool siRNAs of validated hits. For a candidate to be selected, at least 2 of the single siRNAs should have the same effect as the smartpool, which is visualized in these plots. Following the reviewers suggestion, we also created xy-scatter plots between single siRNA and smartpool Z-scores and xy-scatter plots of the reproducibility of the biological replicates (Suppl. Fig. 5A-B). In general we observed a positive correlation of the smartpool and singles Z-scores (cor ~ 0.5). However, there is also quite some variation in terms of response size, a known characteristic of single siRNAs (Falkenburg et al, 2014). This variation is not due to technical variation, since the correlation between biological replicates is rather high (cor ~ 0.9, Suppl. Fig. 5C-D). We refer to these results on page 6 of the manuscript.

- Falkenburg et al. Genome-wide functional genomic and transcriptomic analyses for genes regulating sensitivity to vorinostat. *Sci Data*. 2014; 1: 140017.

Supplemental Figure 5. Reproducibility of single siRNA PKT validation screen (A) Correlation of Z-score of smartpool and single siRNAs in Hs578T. (B) Correlation of Z-score of smartpool and single siRNAs in MDA-MB-231. (C) Correlation of biological replicates in Hs578T. (D) Correlation of biological replicates in MDA-MB-231.

Page 33 The quantification of cell migration regarding BUD31 and BPTF (reinhhibition) are less impressive than PRPF4B and MXD1 and beg further validation using CRISPR and shRNAs.

As discussed before, also the less inhibitory candidates such as BPTF and BUD31 were validated using inducible CRISPR-Cas9 knockout demonstrating that the used candidate selection method was robust. (Suppl. Fig. 8, page 7 of the manuscript).

Page 34, Fig 4 Not clear why “osteoclast differentiation” and “tuberculosis” appear in the network analysis. Also, in 4D, where is the cluster 1 data (amongst the nine clusters)?

We thank the reviewer for pointing out this inconsistency. Cluster 1 data has been added to Figure 4D.

Osteoclast differentiation and tuberculosis were KEGG pathways enriched in the first order PPI network for MDA-MB-231 candidates. These two pathways have a high number of common members and therefore the same candidates are responsible for over-representation of these pathways in the PPI networks. Next to the immune-related component in these pathways, also many AKT and MAPK family members are involved. Both of these gene sets demonstrated to be involved in cell migration in our screen (Figure 4A-B), which could also explain the enrichment of the osteoclast differentiation and tuberculosis pathways in the network analysis. This has been implemented in the manuscript on page 8.

Page 35 Figure 5. Does the hierarchical clustering based on gene expression also cluster highly metastatic and non-metastatic tumor types?

The hierarchical clustering in Figure 5 is based on % copy number variations and mutations, as indicated on page 10. In this clustering, we observe that a subset of these genes is altered in multiple cancer types (DLBC, esophagus, ovarian, melanoma, stomach, uterine, bladder, lung, breast and liver cancer). For the different cancer types we compared the estimated number of new cases with the estimated number of deaths in the USA (Cancer Facts & Figures 2018, Supp. Fig. 14). We do not observe a significant difference between cancer types with a high death rate compared to cancer types with a low death rate, suggesting that candidate alterations are not restricted by tumor type aggressiveness. We implemented these results in the manuscript on page 10.

A

B

Supplemental Figure 14. Tumor type aggressiveness (A) % of deaths compared to new cases for different cancer types estimated for 2018 in the United States. Color indicates the alteration rate for the different candidates. Blue = low alteration rate, pink = high alteration rate. (B) Same as A, now ordered by alteration rate.

Page 38 Supplementary Figure 1. Add number of targets for each of the siRNA libraries. The bottom half of the flowchart shows three target gene inputs and two output lists, one for each cell line. Consolidating the three input diagrams (left of the text Primary hits and selection of top hits) to only Hs587T (top 153 cell line specific + 129 overlapping hits); MDA-MB-231 (top 153 cell line specific + 129 overlapping hits) would make this figure have two inputs and two outputs, as opposed to three inputs and two outputs. Furthermore, the 129 overlapping genes don't match the text, which lists 145 significant hits.

We thank the reviewer for pointing out these inconsistencies. Indeed 129 overlapping genes were selected for validation. This number is based on the initial 145 overlapping genes, followed by elimination of the non-drugable targets and genes that were represented in multiple libraries. We adjusted Suppl. Fig. 1 and the main text on page 6.

Supplemental Figure 1. RNAi PKT screen setup. Transfection of up to 10 siRNA library plates per run was performed by automated liquid handling (BioMek). Transfections were performed in duplicate, on different days with separately grown cell cultures. Transfected cells were washed with PBS, trypsinized, diluted and resuspended into single cell suspension, before being seeded in duplicate PKT assay plates (technical replicate). All steps were optimized for automated liquid handling. Whole well montages (6x6) were acquired on a BD Pathway BioImager using transmitted light, and a robotic arm (Twister II, Caliper) placed and removed the PKT assay plates on the microscope. PKT images were analyzed using PhagoTracker software as described previously (32, 33). Quantitative output was normalized to mock control (robust Z-score) using KNIME. Visual inspection of images led to the identification of migratory phenotypes, which were subsequently used for supervised clustering of hits by means of principal component analysis and plotted in a 3D phenotypic space

(Fig. 1E,F). Primary hits were selected in two ways: hits that showed overlap between the two cell lines for each migratory phenotype (129 hits) and the top hits affecting cell migration within each cell line (153 hits in Hs578T, 153 hits in MDA-MB-231). Primary hits were validated by deconvolution screens, evaluating the effect of SMARTpool and single siRNA sequences in PKT assays as before. Hits were considered validated if the SMARTpool showed consistent results and at least 2 of 4 single siRNA sequences showed the same phenotype. Ultimately, 217 hits were validated in the Hs578T cells and 160 hits in the MDA-MB-231.

Page 42 Supplementary Figure 5 would benefit from the addition of volcano plots (Log₂ Fold Change vs $-\text{Log}_{10}P$) to illustrate the overall results of the differential expression analysis. Plots can be annotated with the current information being presented on panels B and C.

We thank the reviewer for the suggestion. Volcano plots are now implemented in Supplementary Figure 18.

Supplemental Figure 18. Effect of PRBF4B, BUD31 and BPTF depletion on gene expression. (A) qRT-PCR of knockdown efficiency of siPRBF4B, siBUD31 and siBPTF used for next generation sequencing in Hs578T and MDA-MB-231 cells. Data are normalized using the $\Delta\Delta CT$ method normalized to actin and tubulin levels. (B) Vulcano plots for Hs578T. Significant up- and down-regulated genes ($L2FC > 1$ or < -1 and $P\text{-adjusted} < 0.01$) are shown in green and red, respectively. (C) Same as in B for MDA-MB-231. (D) Overlap of DEGs in Hs578T and MDA-MB-231. (E) Overlap of DEGs comparing different knockdown conditions.

References for Specific Observations section

Nik-Zainal et al. Nature. 2016 Jun 2;534(7605):47-54.

Van der Weyden et al. Nature. 2017 Jan 12;541(7636):233-236.

Reviewer #3, Expertise: Splicing factors, breast cancer

(Remarks to the Author):

Fokkelman et al. describe the findings from a large-scale phenotypic imaging-based RNAi-screen to identify genes involved in the regulation of migratory phenotypes in two human triple negative breast cancer (TNBC) cell lines. Uncovering the determinant of metastatic dissemination and progression is highly needed to treat and prevent aggressive breast tumors such as TNBC. This study discovered novel regulators of cell migration in TNBC including the transcriptional modulator BPTF and the splicing factors PRPF4B and BUD31. Using in vitro and in vivo models, Fokkelman et al. further demonstrate the reducing PRPF4B levels inhibits cell migration and decreases the number of metastatic lesions, and could constitute a promising drug target for metastatic cancer. Findings from this study thus contribute to improving our understanding of the regulators of TNBC cell migration and highlight the role of splicing regulators in tumor progression.

However several questions should be addressed prior to publication, and should result in an enhanced manuscript.

Major comments:

1) The major weakness of this study is the lack of data measuring cell viability/proliferation following knock down of target genes. While the various assays used by Fokkelman et al. allow a thorough analysis of the different migratory phenotypes, none of them can distinguish an effect on cell viability/proliferation from an effect on cell migration. If a gene knockdown negatively impacted cell viability and promoted cell cycle arrest or cell death, the cells would not exhibit a migratory behavior in the time frame of the experiment, and would presumably be scored as negatively impacting migration. This should be addressed by the authors, or at least discussed, especially given that a number of studies demonstrated that knockdown of RNA-binding proteins, and specifically splicing factors, blocks cell cycle progression and leads to cell death. In addition, the author achieve a PRPF4B knockdown efficiency of ~40% using shRNA in xenograft experiments, compared to an >80% efficiency using siRNAs. This suggests that cells that exhibit a high PRPF4B knockdown efficiency are eliminated when creating stable cell lines and do not give rise to tumors in xenograft experiments; highlighting the potential link between knockdown efficiency and cell viability.

We thank this reviewer for raising this important concern about cell viability. This point was also raised by reviewer 1 and 2. Our screening set up indirectly takes into account the cytotoxicity of siRNA in our cell lines. Thus, 72 hr after transfection we trypsinize and replat the cells to the PKT assay plates. In each well we then determine the number of cell migration tracks, which is a representative of the cell number after 72 hr siRNA treatment. Therefore, we could investigate whether proliferation was a major confounder in our screen. We used the number of tracks per well (measure for proliferation) and plotted this against the different phenotypic track parameters (Suppl. Fig. 3). We did not observe any correlation between the track number and any of the phenotypic parameters, suggesting that the candidates we selected in our primary screen were particularly based on effects on migration. We did not investigate effects of siRNA candidate knockdown on TNBC cell proliferation longer than 72 hr. As for PRPF4B, our stable shPRPF4B cell lines only showed 40% knockdown. At this stage we cannot conclude that the 40% knockdown efficiency in the shRNA

stable knockdown cell lines was caused by limited effectivity of the shRNA constructs or loss of cells that showed sustained high knockdown levels of PRPF4B. Regardless, the established shPRPF4B cell line did grow equally fast as the shCtrl cell lines in vivo while decreasing metastasis formation, suggesting that there is a window in which PRPF4B knockdown only effects cell migration. Since we did not perform these extensive evaluations for all the other candidates, we cannot exclude that effects on cell proliferation might be a contributing factor to the inhibition of cell migration for some of the candidates, in particular for those candidate genes that show a decrease in track number. We have addressed this issue manuscript on page 6.

2) Overall the description of the siRNA screen and downstream computational analysis need a more detailed description of the methodology, time points, and statistical analysis within the main text.

- Please add brief description of the siRNA library, e.g., how many siRNAs per target.

For the primary screen, we combined 4 single siRNAs (SMARTpool) per target. In the validation screen, four single siRNAs were tested separately in addition to the SMARTpool. This information is shown in the manuscript on page 5 and 6.

- The authors state "In total, 217 hits [from the primary screen] were validated in the Hs578T and 160 in the MDA-MB-231...". Please describe how many validated/tested, how many siRNA per target, what are the negative and positive controls and add statistics for the validation rate.

For both MDA-MB-231 and Hs578T 282 candidates were selected for validation (page 6). We used 4 singles and SMARTpool siRNA for validation (page 6) and candidates were considered validated when at least 2 out of 4 singles and the SMARTpool showed significant effects on the migratory phenotype (page 6). We used mock (no siRNA added) as a negative control and siDNM2 was used as a positive control (page 6). In total we validated 217 hits for Hs578T (77%) and 160 hits for MDA-MB-231 (57%) (page 6). Moreover, the correlation between the smartpool Z-scores of primary and validation screen was highly significant (Suppl. Fig. 6), confirming that our results are highly reproducible. We refer to these results on page 6 of the manuscript.

Supplemental Figure 6. Pearson correlation of Z-scores from primary screen and validation screen for net area and axial ratio track parameters in both Hs578T and MDA-MB-231.

- The authors state “Annotation of protein classes for each set of validated hits (Hs578T, MDA-MB-231, and overlap) showed that most of the hits were transcription factors (Figure 2E i) also after correction for library size (Figure 2E ii)...”. Please provide statistical analysis.

Among the candidates, transcription factors were significantly over-represented (Fisher’s exact test). Statistical significance was added to Figure 2E.

- The authors state “we used the larger lists of our PKT validated candidate genes to inform on protein-protein interaction (PPI) networks that are involved in Hs578T and MDA-MB-231 cell migration... [...] “. Please describe how many genes tested vs. number in background and type of statistical analysis.

The purpose of this analysis, was to investigate the relation between our set candidates and previously published metastatic and migratory networks. Looking at the original candidate lists, we identified only a very limited overlap, due to the low number of genes represented in all datasets. To investigate whether the same pathways were involved in these different datasets, we expanded our gene lists by defining complexes using PPI interactions resulting in many commonly affected pathways (Fig. 5A). Determining statistical significance appears to be difficult, since all of these

datasets are different regarding both the number of candidates and the number of genes in the background/the initial number of genes in the screen. Moreover, the number of interactors is highly depending on each specific target gene; some genes have many interactors, others very few. Altogether, we think that these PPI networks confirm a relation between the migratory candidates we identified and the metastatic genes identified by others, implying that our genes are part of biologically functional networks involved in metastasis formation. However, we cannot make a conclusion about the significance of this overlap (implemented in the manuscript on page 10).

- Regarding the TCGA analysis, the authors state “we identified clusters of candidate genes highly altered in multiple cancer types, amongst which breast cancer”. Please describe how many genes tested vs. number in background and type of statistical analysis.

In this initial clustering, we tested the 43 live cell migration validated candidates among the different cancer types shown in Figure 5B. The purpose of this analysis was to observe the variability in alteration rate across different cancer subtypes and candidates and define the candidate genes that may bear higher translational relevance. Next, according to the reviewer’s suggestion (see point 3), we also investigated these alteration rates in different primary tumor subtypes based on hormone receptor status or metastatic status. Here we found a significant enrichment in the TNBC subtype, suggesting that the alteration rates we observe in Figure 5B are related to the aggressiveness of the primary tumor.

- RNA-seq analysis: please provide read depth and number of biological replicates, as well as time frame of the experiment.

For these experiments, we sequenced 20.10^6 reads (100 base pairs, paired end) of candidate and control knockdown 72 hours after siRNA transfection, as described in the methods section (page 24). This information has now also been added to the results section on page 11.

3) The authors identify that BUD31 and PRP4B are each amplified in ~2% of breast tumors, whereas BPTF is amplified in ~8% tumors and correlate their expression with clinical phenotypes (Figure 5). This analysis would be greatly enhanced if the authors (i) focused their analysis specifically on TNBC vs. other breast cancer subtypes using available RNA-seq and DNA-seq data, (ii) analyzed paired primary and metastatic tumor samples using available RNA-seq and DNA-seq data, (iii) analyzed changes in RNA expression in addition to amplifications and mutations to determine if additional epigenetic or transcriptional control mechanisms can impact BUD31 PRP4B or BPTF expression even in absence of copy number changes or mutations. Finally, given that solid tumors exhibit a large number of genomic alterations, the analysis would be strengthened if the author compared TCGA data for negative candidates from their screen.

As suggested by the reviewer, we now also investigated the RNA expression levels and mutation, amplification and deletion rates for the 43 candidates shown in Figure 5B also specifically in TNBC and estrogen receptor (ER) positive subtypes. Interestingly 14 candidates were significantly amplified in the TNBC subtype including BUD31 and PRPF4B (Suppl. Fig. 16). BPTF is an exception and mainly (not significantly) amplified in the ER positive subtype, which could explain the strong relation to metastasis formation observed in ER positive breast cancer patients (Figure 5D). We agree with the

reviewer that solid tumors generally exhibit large numbers of genomic alterations. Therefore, we randomly selected 43 negative candidates from the screen and compared their amplification rates in TNBC and ER positive tumors (Supp. Fig. 16C). We only identified 3 negative candidates with significantly altered amplification rates, suggesting that our positive screen candidates truly enriched for TNBC amplifications.

The comparison between metastatic and primary tumor samples is rather difficult because of the limited number of metastatic samples (7) available in the TCGA database. However, as suggested by reviewer 2, we compared the mutation, amplification and deletion rate between primary tumors with and without metastases. In accordance with the subtype specific effects, we identified increased amplification rates for the candidates in the aggressive tumors that already metastasized (Suppl. Fig. 15). However, these results would need further validation with increased sample size, since DNA-seq data was available for only 16 tumors with metastasis at diagnosis.

These results have been implemented in the manuscript on page 11.

Supplemental Figure 15. Candidate alterations in primary tumors with and without metastasis at diagnosis (A) Log₂ RNA expression levels of candidates in primary tumors with (green) or without (red) distant metastases at diagnosis. (B) Candidate mutation rate (i), amplification rate (ii) and deletion rate (iii) in primary tumors with (green) or without (red) metastases at diagnosis.

Supplemental Figure 16. Candidate alterations in ERpos and TNBC primary breast tumors (A) Log₂ RNA expression levels of candidates in ERpos and TNBC primary tumors. (B) Candidate mutation rate (i), amplification rate (ii) and deletion rate (iii) in ERpos and TNBC primary tumors. (C) Amplification of randomly

selected non-hits in ERpos and TNBC primary tumors. Significance was calculated using the Fisher's exact test using the Benjamini-Hochberg method to correct for multiple testing. * $p < 0.05$, ** $p < 0.01$, *** $p < 0.001$.

4) The authors analyzed the RNA-seq data to identify changes in splicing events following BUD31 or PRP4B siRNA-mediated KD, and focused specifically on intron retention events. We suggested using a computational pipeline more appropriate for intron retention detection, a challenging task that often generates many false positive, as well as including RT-PCR validation for several of these events. In addition, it is surprising that the authors were "unable to detect any changes in exon inclusion or 3' or 5' alternative splice site usage after knockdown of either PRPF4B or BUD31"; we suggest re-analyzing the data with computational pipeline dedicated to splicing analysis (e.g. MISO, rMATS, etc) that would allow to identify additional splicing event types, such as cassette exon, alternative acceptors and donors, to ensure that no changes are indeed detected. The author further focus on the differential genes decreased upon siRNA knockdown, stating that these will be a direct consequence of increased intron retention – a logical assumption that should however be analyzed more thoroughly.

In the original analysis, we already analyzed the data with a computational pipeline dedicated for alternative splicing analysis (rMATS). We overlooked that this information was not added to the methods section and we incorporated this in the revised manuscript (page 25). As suggested by the reviewer, we performed RT-PCR experiments to validate intron inclusion events induced by PRPF4B and BUD31 knockdown (Suppl. Fig. 22). We were able to validate all tested intron retention events, indicating that the computational pipeline we used is reliable for this purpose. These results are implemented in the manuscript on page 12.

The relation between intron retention and reduced gene expression amongst others due to nonsense-mediated decay has extensively been demonstrated in the literature (Bergeron et al, 2015; Wong et al, 2013; Braunschweig et al, 2014). Therefore, we think it is very likely that gene downregulation is a direct consequence of intron retention. Although we did observe a strong relation between intron retention events and gene downregulation, we agree with the reviewer that more information would be needed to validate a direct causal relationship in this study. We discussed these limitations in the manuscript on page 12.

- Bergeron et al. Regulated Intron Retention and Nuclear Pre-mRNA Decay Contribute to PABPN1 Autoregulation. *Mol Cell Biol.* 2015 Jul;35(14):2503-17.
- Wong et al. Orchestrated intron retention regulates normal granulocyte differentiation. *Cell*, 154 (2013), pp. 583-595.
- Braunschweig et al. Widespread intron retention in mammals functionally tunes transcriptomes. *Genome Res.* 2014 Nov;24(11):1774-86.

Supplemental Figure 22. Validation of intron retention events in MDA-MB-231 (A) RT-PCR validation of PRPF4B knockdown induced alternative splicing events in MDA-MB-231 detected by NGS. (B) RT-PCR validation of BUD31 knockdown induced alternative splicing events in MDA-MB-231 detected by NGS. Mean + stdev of three biological replicates. Significance is determined using a student's t-test. * $p < 0.05$, ** $p < 0.01$, *** $p < 0.001$.

5) The authors further focus on the role of PRPF4B in vitro and in vivo using MDA-MB231 and its lung metastatic variant, MDA-MB-417.5. A few questions need to be addressed or at least discussed: (i) siRNA KD of PRPF4B in MDA-MB231 has an effect of the cell migratory phenotype, yet there are very few changes in splicing or gene expression compared to other siRNA (Supp. Figure 5 and Supp Figure 7) thus providing little explanation on what mediated the phenotype; (ii) is there a difference in

PRPF4B expression MDA-MB231 and its lung metastatic variant, MDA-MB-417.5?; (iii) what is the effect of shPRPF4B KD on cell migration in MDA-MB-417.5? (iv) an analysis of the splicing events in MDA-MB-417.5 shRNA PRPF4B used for the xenograft experiments would likely allow to identify the splicing events that are relevant to metastatic progression in vivo.

We agree with the reviewer that knockdown of PRPF4B resulted in significantly less splicing differences than BUD31 knockdown. This was expected since BUD31 is a component of the core spliceosome, essential for both assembly and catalytic activity of the splicing reaction (Hsu et al, 2015). PRPF4B is a non-core component regulating mRNA splicing via phosphorylation of other spliceosome components (Corkery et al, 2015), and thereby probably less essential for global splicing activity. Although the splicing changes were maybe less abundant, they could be validated with RT-PCR (Suppl. Fig. 22). Next, we also validated some of these alternative splicing events in the LM2 shPRPF4B cell line, suggesting that similar mechanisms are involved in this cell line (Suppl. Fig. 28E-H). Furthermore, basal PRPF4B levels were similar in MDA-MB-231, LM2 shCtrl#1 and shCtrl #2 (Supp. Fig. 28A-B). Wound healing was significantly decreased upon stable PRPF4B knockdown in the LM2 cell line (Supp. Fig. 28C-D), again confirming the important role of PRPF4B in breast cancer cell migration. These results are discussed in the manuscript on page 14.

Supplemental Figure 28. Stable PRPF4B knockdown affects RNA splicing and cell migration. (A) PRPF4B expression in Hs578T, MDA-MB-231 and LM2 cell lines. (B) Quantification of G. PRPF4B expression levels were normalized to tubulin expression levels. Mean of all cell lines is equal to 1. (C) Scratch assay of stable LM2 knockdown cell lines. (D) Quantification of E. Mean + stdev of 15 measurements in 2 biological replicates. DGKZ (E), POMGNT1 (F), MAF1 (G) or CDCA5 (H) intron retention in stable knockdown cell lines. Mean + stdev of 3 biological replicates. Significance was determined using ANOVA correcting for multiple testing. * $p < 0.05$, ** $p < 0.01$, *** $p < 0.001$.

- Hsu et al. The spliceosome is a therapeutic vulnerability in MYC-driven cancer. *Nature*. 2015 Sep 17;525(7569):384-8.
- Corkery et al. PRP4K is a HER2-regulated modifier of taxane sensitivity. *Cell Cycle*. 2015;14(7):1059-69.

6) The authors demonstrate that shPRPF4B KD decreases the number of metastatic nodules in the lung as determined by macroscopic evaluation, as well as the total bioluminescence flux. Yet, in figure 7E and 7E, a few shPRPF4B animals still exhibit a similar number of metastasis to control animals. Have the authors collected tissues from these animals and verified if these metastatic lesions still express the shPRPF4B and exhibit lower levels of PRPF4B, or alternatively have escape the shRNA KD or silenced the shRNA? A more detailed analysis may uncover that the author underestimate the effect of PRPF4B KD.

We thank the reviewer for these suggestions. Unfortunately we did not collect tissue containing metastatic lesions we could use to investigate this in greater detail. We agree with the reviewer that such an analysis could add to the study and in future experiments, we will biobank the metastatic organs. It may therefore well be that the expression of PRPF4B is higher in metastasis than in the primary tumor and that cells with higher levels of PRPF4B were able to disseminate, home and grow out as metastatic lesions. Regardless of the potential of such an underestimation of the effect of PRPF4B, the lack of this data does not affect the current conclusions in the manuscript.

7) A number of small molecule inhibitors of the splicing machinery have been tested in cancer cell lines. Please discuss if any of these have been shown to affect cell migration. How specific is the effect of PRPF4B and BUD31 vs. global inhibition of splicing, e.g. were other splicing factors tested in the screen?

Indeed, the last decade there has been more attention for the development of splicing inhibitors in the context of cancer treatment (Lee and Abdel-Wahab, 2016; Salton et al, 2016; Agrawal et al, 2018). The majority of the studies using small molecule inhibitors focused on the effect on cancer proliferation (Salton, 2016; Iwai et al, 2018; Kotake et al, 2007). Furthermore, treatment with the SF3B1 inhibitor Pladienolide B was shown to, next to inhibiting proliferation, effectively inhibit cell migration in prostate cancer cells (Jiménez-Vacas et al, 2018). However, to our knowledge, the relation between migration inhibition and small molecule inhibitors of the spliceosome in the context of (triple negative) breast cancer has not yet been addressed. In total, there were 43/244 splicing factors (Hegele et al, 2012) represented in the primary screen (Suppl. Fig. 32). Mainly in MDA-MB-231 cells, there seems to be a prominent role for splicing in breast cancer cell migration with 11 out of 43 factors being selected for further validation. Various of these factors were filtered out later in the selection procedure due to stricter cut-offs. Yet, it would indeed be highly relevant in future studies to systematically evaluate the role of splicing in breast cancer cell migration taking into account all 244 splicing factors. These results are discussed in the manuscript on page 18.

- Lee SC, Abdel-Wahab O. Therapeutic Targeting of Splicing in Cancer. *Nat Med.* 2016 Sep 7;22(9):976-86.
- Salton M and Misteli T. Small molecule modulators of pre-mRNA splicing in cancer therapy. *Trends Mol Med.* 2016 Jan;22(1):28-37.
- Agrawal et al. Targeting splicing abnormalities in cancer. *Curr Opin Genet Dev.* 2018 Feb;48:67-74.
- Iwai et al. Anti-tumor efficacy of a novel CLK inhibitor via targeting RNA splicing and MYC-dependent vulnerability. *EMBO Mol Med.* 2018 Jun;10(6).

- Pawellek et al. , Identification of Small Molecule Inhibitors of Pre-mRNA Splicing. J Biol Chem. 2015 Mar 6;290(10):6005.
- Jimenez-Vacas et al. Inhibition of alternative splicing using the spliceosome inhibitor Pladienolide B reduces aggressiveness of prostate cancer cells in vitro. Endocrine Abstracts (2018) 56 P653.
- Hegele et al. Dynamic protein-protein interaction wiring of the human spliceosome. Mol Cell. 2012 Feb 24;45(4):567-80.
- Kotake, 2007, Splicing factor SF3b as a target of the antitumor natural product pladienolide. Nat Chem Biol. 2007 Sep;3(9):570-5.

Supplemental Figure 32. Effect of splicing factors in the primary PKT screen in Hs578T and MDA-MB-231.

Minor comments:

1) Introduction: please cite the following papers performing screens to identify genes involved in metastatic progression in breast cancer cell lines <https://www.ncbi.nlm.nih.gov/pubmed/29414946> <https://www.ncbi.nlm.nih.gov/pubmed/25855289>

We thank the reviewer for these suggestions and we cited these papers in the introduction (page 3).

2) Please add data for RNAi efficiency at the RNA and protein levels for at least BTPF, BUD31, PRPF4B in each of the cell lines. Please specify how many siRNA have been tested for each target.

As described in the methods and now also in the results section (page 5 and 6); we used SMARTpool siRNA during the primary screen. Here, 4 single siRNAs for the same target are combined to increase knockdown efficiency. To confirm the SMARTpool effects, in the validation screen we used both SMARTpool siRNA and the 4 singles separately. As described in the results section, we selected only candidates in which the effect observed in the primary screen could be reproduced by the SMARTpool and at least 2 singles. For all other experiments, SMARTpool siRNAs were used. Due to antibody availability, only PRPF4B knockdown efficiency was validated using western blot (Supp. Fig. 19). However, knockdown efficiencies of BPTF, BUD31 and PRPF4B were efficient as shown by next generation sequencing (Supp. Fig. 18) and RT-qPCR (Suppl Fig. 19), confirming the on-target activity of the used siRNAs. This has been implemented in the manuscript on page 12.

Supplemental Figure 19. Candidate siRNA knockdown efficiency in Hs578T and MDA-MB-231. (A) PRPF4B knockdown efficiency on protein level in Hs578T. (B) PRPF4B, BUD31 and BPTF knockdown efficiency on RNA level in Hs578T based on RNAseq data. (C) PRPF4B knockdown efficiency on protein level in MDA-MB-231. (D)

PRPF4B, BUD31 and BPTF knockdown efficiency on RNA level in MDA-MB-231 based on RNAseq data. Experiments were performed in biological triplicates, significance was calculated using student's t-test. *** p < 0.001

3) The author use a platform that allows a very thorough analysis of the different migratory phenotypes that would be of great interest to the readers, yet the description of these phenotypes is very spare. Please expand the description in the text and supplemental figures. In addition, the author state that “a decrease in migration does not necessarily coincide with an overall change in cell morphology.” Please describe if the screen is corrected for differences in cell size. One can assume that larger cells would migrate further than smaller cells.

We thank the reviewer for these suggestions. We did expand the description of the phenotypes in the results section on page 5 and the methods section on page 22 and updated the legend of Figure 1 and Supplementary Figure 1. In the primary screen, we did not correct for differences in cell size. We agree with the reviewer that gene knockdown could potentially lead to a decrease in migration accompanied with an increase in cell size, resulting in a similar or even bigger and round track area. This typically relates to an increased minor axis and equal or decreased major axis resulting in a round track phenotype. Because these knockdowns do impair migration and are therefore of interest, we also selected tracks with a round phenotype. This has been implemented in the manuscript on page 5.

4) The author focus on a set of genes from the initial screen, including “some of which are directly involved with splicing (BUD31 and PRPF4B)”. However, we could not find a reference to BUD31 in the main figures Fig1-2 or corresponding supplemental material. Please explain or correct this discrepancy. Please add the phenotypes for BUD31 and BPTF to Figure 1.

We thank the reviewer for noticing this inconsistency. The phenotype of BPTF was already shown in Figure 1F. BUD31 has now been added to Figure 1E.

5) BPTF Please cite the following paper relevant to BPTF in breast cancer: <https://www.ncbi.nlm.nih.gov/pubmed/28579392>

We thank the reviewer and implemented the paper in the manuscript (page 11).

6) The authors state “Clustering of all genes involved in ECM receptor interaction (Fig. 6C, see Suppl. Fig. 9 for all gene names) or focal adhesion (Suppl. Fig. 10) demonstrated the involvement of many different pathway components of which some were overlapping between PRPF4B, BUD31 and BPTF (Fig. 6C and 6D); a similar downregulation was observed at the protein level for several key components in both cell lines (Fig. 6E, Suppl. Fig. 11C).” Please describe the components the authors are referring to and how they potentially contribute to increased metastatic dissemination.

In Figure 6, we are mainly referring to integrins and its interactors (laminins and collagen) and Focal Adhesion Kinase (FAK). A prominent role for integrins has been demonstrated in various steps of the

metastatic cascade amongst which migration and invasion, metastasis and anchorage-independent growth and metastatic colonization (Hamidi et al, 2018) . Many studies have related increased integrin adhesions to EMT and cancer cell dissemination (Yilmaz et al, 2009). Moreover, integrin signaling can activate actin contraction (Martinez-Rico et al, 2010) and regulate E-cadherin internalization and cell adhesion, affecting cancer cell movement (Canel et al, 2010). These results have been implemented in the manuscript on page 13.

- Hamidi, 2018, Every step of the way: integrins in cancer progression and metastasis
- Yilmaz, M. & Christofori, G. EMT, the cytoskeleton, and cancer cell invasion. *Cancer Metastasis Rev.* 28, 15–33 (2009).
- Martinez-Rico, C., Pincet, F., Thiery, J. & Dufour, S. Integrins stimulate E-cadherin-mediated intercellular adhesion by regulating Src-kinase activation and actomyosin contractility. *J. Cell Sci.* 123, 712–722 (2010).
- Canel, M. et al. Quantitative in vivo imaging of the effects of inhibiting integrin signaling via Src and FAK on cancer cell movement: effects on E-cadherin dynamics. *Cancer Res.* 70, 9413–9422 (2010)

7) The author state that “Our combined data indicate that the various candidate hits differentially affect cell morphology and migratory phenotypes, indicative of different genetic programs that define BC cell migration behavior”. Please discuss the correlation between cell adhesion vs. cell migration and its effect of metastasis in vivo? Discuss the different steps of the metastatic cascade, and how the differences in migratory phenotypes relate to them.

Generally the metastatic cascade can be separated in three phases: invasion, intravasation and extravasation. In order to invade the surrounding environment, tumor cells need to lose their cell-cell contacts (Martin et al, 2009). As a next step, cells need to become motile via differences in cell matrix interactions (Martin et al, 2013). Due to angiogenesis, tumor cells can adhere to the endothelial membrane and intravasate into the blood. Altogether, this suggests that loss of adhesion and increased motility are both prerequisites for metastasis formation; there will be no migration without loss of cell-cell interactions and increased motility is essential to reach the blood vessel and intravasate. However, these programs are partly controlled by different complexes: cell-cell junctions for adhesion and cell-matrix interactions and actin turnover for motility. Taking this into consideration, our candidates could inhibit in vitro cell migration via different mechanisms: 1) increased cell-cell adhesions, 2) decreased cell-matrix interactions and 3) decreased actin turnover that can also result in multiple cell phenotypes. Decreased cell-matrix interactions will decrease the area, while decreased actin turnover might more affect the focal adhesions and spikes. In this way, it is impossible to connect migratory behavior directly to cell phenotype. This has been discussed in greater detail in the manuscript on page 9-10.

- Martin TA, Jiang WG. Loss of tight junction barrier function and its role in cancer metastasis. *Biochim Biophys Acta.* 2009;1788:872–91. <http://dx.doi.org/10.1016/j.bbmem.2008.11.005>
- Martin, 2013, *Cancer Invasion and Metastasis: Molecular and Cellular Perspective*

8) The authors state “The effects on differential expression of cell matrix adhesion components was

also reflected in the different organization of focal adhesions”. Please describe the components the authors are referring to and how they potentially contribute to increased metastatic dissemination.

Here, again we refer to the differential expression of integrins and focal adhesions as shown before. This has now been added to the manuscript on page 13.

9) In the discussion: “Also decreased levels of PRPF4B almost completely eradicated spontaneous metastasis formation from the orthotopic primary tumor to distant organs...” Please replace “eradicated” with a more appropriate term, e.g. “prevented”

We thank the reviewer for this suggestion. We replaced ‘eradicated’ to ‘prevented’ in the discussion on page 19.

10) In the discussion, please add a paragraph about the link between splicing regulation and cell migration and metastatic potential.

We thank the reviewer for this suggestion and added a paragraph about the link between splicing and breast cancer metastasis in the discussion on page 18.

Reviewers' comments:

Reviewer #1 (Remarks to the Author):

Despite a wealth of information, the most important endpoint remains limited to one metastasis model, with a very limited number of genes interrogated in vivo. This level of data is published in medium impact oncology journals.

Reviewer #2 (Remarks to the Author):

Nature Communications December 2018, reviewer #2 response to author rebuttal:
Second revision of "Uncovering the signaling landscape controlling breast cancer cell migration identifies novel metastasis drivers" by Koedoot, Fokkelman, van de Water et al.

Hearty thanks and congratulations are due to Koedoot, Fokkelman and contributors for their extraordinary effort and substantial improvements to the original manuscript. In this version, the authors have addressed the major suggestions either by discussing them in the manuscript text and/or providing additional supporting data, these include the impact of cell viability and the phenotypes, the principal component analyses, the novelty of the study, study bias, the use of external sources, and study conclusions. This, in addition to addressing all minor suggestions.

One minor point was left unclear, on a question which was posed in the original review, re: were there LOF or activating mutations or CNAs on genes in the pathways related to cell-specific targets? In the original review there was little discussion on the validated targets from the cell-line specific genes. Authors have corrected this by reviewing the genetic alterations on these genes, concluding that no enrichment of genetic alterations or expression in cell-line specific vs. overlapping targets, and suggest cell-line specific dependencies are the root cause of these genes being specific to each cell line. The initial question still remains unanswered which was aimed precisely to identify any putative cell-line specific dependencies the authors have remarked upon by looking at the genetic alterations of the genes within the pathways of the top targets. Again, at this junction, this is seen as a minor point which does not detract from the excellence of the revised paper as a whole

Minor general observations:

1) There are several figures Supplemental Figures 2, 11,12, 15, 16, 18, and 31 that made use of the color red and green. Particularly the shades of red and green that are difficult to distinguish by color-blind individuals. It is suggested these could be changed, particularly on the heatmaps (the bar graphs are better). Here are two websites that can help.

<http://mkweb.bcgsc.ca/colorblind/>

<https://www.tableau.com/about/blog/2016/4/examining-data-viz-rules-dont-use-red-green-together-53463>

2) In Supplemental Figure 7, would it be possible to run Ensembl VEP on the 15 cell-line mutations as to provide some information on the mutation type i.e. non-synonymous, synonymous, intronic?

Reviewer #3 (Remarks to the Author):

The authors have performed extensive work to enhance the manuscript, adding valuable experimental evidence to support several of their findings. We thank the authors for following the reviewer's comments and suggestions to improve the manuscript clarity and include much needed details on the experimental design. The resulting revised version has been much improved and

addressed most of the comments and questions. Yet, one main question has not been addressed and in our opinion could impact the findings on the study.

Major comment

1) The authors have yet to demonstrate that the siRNA knock down of PRP4B, BTPF and BUD31 does not arrest the cell cycle, and that the differences in cell migration are not simply due to the fact the cells are growth arrested and thus are not moving. This would not impact the number of tracks detected and this would not be excluded from the analysis as stated by the authors in the rebuttal. This concern regarding confounding effects of cell proliferation or cell viability on the results presented here has been raised by all three reviewers.

Many splicing factors have been shown to affect cell cycle, and can lead to increase cell death, decreased cell proliferation, or delayed G1/M entry. For example, previous siRNA screens for regulators of the pro-apoptotic proteins BCL-X and MCL1 or FAS uncovered an enrichment in spliceosomal components and splicing factors at 72h of treatment (Moore et al. Cell 2010; Papasaikas et al. Mol Cell 2015). The new supplemental data suggests that there might be an effect on cell viability: in Figure S10A,C there are fewer cells visible on the slide after PRP4B siRNA KD at 20h vs. 0h.

Measuring the number of cell tracks during the 7h of the migration experiment as shown in Fig S3 is not a reliable measure of cell growth arrest. This could be easily addressed by performing cell cycle assays or cell proliferation (MTT) assays on cells +/- siRNA at different time points over the course of the experiment (from 0h to 59hrs).

Minor comments

1) Please state more explicitly how the variables can affect the experiments in the result section (Page 5): (i) latex beads need to be phagocytosed by cells to score migration, thus a defect in phagocytosis would affect the results of the assay; (ii) difference in cell size would affect the distance migrated.

2) Please define what do you mean by "migratory phenotypes were manually classified" (page 5) – what is a manual classification?

3) Please state more explicitly the differences in the migratory assays in the results section (page 8): (i) difference between boyden chamber assays vs. fibronectin-coated plates; (ii) differences in 12h vs 7h migration; (iii) what is the time course of the siRNA treatment in the boyden chamber assay?

4) Please state more explicitly the differences between the Van der Weyden mouse in vivo screen and the current screen including the presence/absence of immune components and microenvironment (page 17).

5) Please provide a control non targeting guide for the CRISPR experiment.

6) The authors have performed significant work to analyze patient's data with regards to clinical phenotypes and genomic profiles. The figures, analysis and accompanying text could be improved to provide more clarity. Fig S15 and S16 are difficult to read, we suggest presenting the data as stacked columns representing for example % mutation in tumors with mets at diagnosis, and no mets at diagnosis. In addition a comparative analysis of selected non target genes would be useful. Please state more explicitly the limitations of the analysis including the small number of samples in the result section.

7) Please add in the result section a brief description of the splicing analysis and cut-off threshold (page 12) and explain Fig S21 – in particular how the red dots (significant events are selected).

One possible explanation for the following statement “We were unable to detect any changes in exon inclusion or 3’ or 5’ alternative splice site usage after knockdown of either PRPF4B or BUD31” is an insufficient read depth (20M vs 100M recommended for splicing analysis), as well as distinct analysis pipeline (DexSeq for intron retention and rMATS for exon inclusion) – this should be explicitly stated in the result section so that the reader is not left with the impression that PRPF4B, BUD31 and BTF only regulate intron retention. Please note that the commonly used cutoff for rMATS is $FDR < 0.05$ and that utilizing the same p-value cutoffs for DexSeq and rMATS is not appropriate as these methods are very different and their p-values are not directly comparable, and likely explains why no exon inclusion events are detected. Please provide results of the splicing analysis as a supplemental file listing all the splicing events types.

8) Please make sure that the legends of all the figures containing required details of the experiment and analysis, including what is plotted in the error bars, number of samples, statistical analysis etc.

Reviewer #2 (Remarks to the Author):

Nature Communications December 2018, reviewer #2 response to author rebuttal: Second revision of "Uncovering the signaling landscape controlling breast cancer cell migration identifies novel metastasis drivers" by Koedoot, Fokkelman, van de Water et al.

Hearty thanks and congratulations are due to Koedoot, Fokkelman and contributors for their extraordinary effort and substantial improvements to the original manuscript. In this version, the authors have addressed the major suggestions either by discussing them in the manuscript text and/or providing additional supporting data, these include the impact of cell viability and the phenotypes, the principal component analyses, the novelty of the study, study bias, the use of external sources, and study conclusions. This, in addition to addressing all minor suggestions.

One minor point was left unclear, on a question which was posed in the original review, re: were there LOF or activating mutations or CNAs on genes in the pathways related to cell-specific targets? In the original review there was little discussion on the validated targets from the cell-line specific genes. Authors have corrected this by reviewing the genetic alterations on these genes, concluding that no enrichment of genetic alterations or expression in cell-line specific vs. overlapping targets, and suggest cell-line specific dependencies are the root cause of these genes being specific to each cell line. The initial question still remains unanswered which was aimed precisely to identify any putative cell-line specific dependencies the authors have remarked upon by looking at the genetic alterations of the genes within the pathways of the top targets. Again, at this junction, this is seen as a minor point which does not detract from the excellence of the revised paper as a whole.

We thank the reviewer for the clarification of the question and we are sorry for the previous misinterpretation. As suggested by the reviewer, we performed an over-representation analysis of the cell line specific hits of our primary screen. Next, we identified the pathways that displayed the biggest enrichment differences comparing the different cell lines (Suppl. Fig. 8A). For these pathways, we determined mutations and copy number alterations in MDA-MB-231 and Hs578T cell lines using the COSMIC database. Despite the differences in over-representations, we did not observe a change in the percentage of mutations, copy number gains or copy number losses comparing the different cell lines (Suppl. Fig. 8B-D). This, in combination with the previously performed analysis (Suppl. Fig. 7) demonstrates that the cell line specific candidates cannot directly be explained by general differences in mutations or copy number variations of the candidates itself or their related pathways. These results have been discussed in the manuscript on page 7.

Supplemental Figure 8. Mutations and copy number alterations of over-represented pathways in Hs578T and MDA-MB-231 specific candidates. (A) Top pathways over-represented in primary screen candidates specific for MDA-MB-231 (left) or Hs578T (right) cell lines. (B) The percentage of genes of selected pathways mutated in Hs578T and MDA-MB-231 cells. (C) Percentage of genes of selected pathways bearing copy number (CN) gain in Hs578T or MDA-MB-231 cells. (D) Percentage of genes of selected pathways bearing CN loss in Hs578T or MDA-MB-231 cells.

Minor general observations:

1) There are several figures Supplemental Figures 2, 11,12, 15, 16, 18, and 31 that made use of the color red and green. Particularly the shades of red and green that are difficult to distinguish by color-blind individuals. It is suggested these could be changed, particularly on the heatmaps (the bar graphs are better). Here are two websites that can help.

<http://mkweb.bcgsc.ca/colorblind/>

<https://www.tableau.com/about/blog/2016/4/examining-data-viz-rules-dont-use-red-green-together-53463>

We apologize for not taking into consideration the used color schemes regarding color blindness and thank the reviewer for this advice. We changed the colors of the heatmaps and bar graphs in Figure 4 and 6 and Supplemental Figures 2, 9, 12, 15, 16, 18, 20, 25, 26 and 31.

2) In Supplemental Figure 7, would it be possible to run Ensembl VEP on the 15 cell-line mutations as to provide some information on the mutation type i.e. non-synonymous, synonymous, intronic?

According to the reviewers suggestion we added more detailed information about the mutation types and consequences using the COSMIC database in Supplemental Table 3 and page 7.

Reviewer #3 (Remarks to the Author):

The authors have performed extensive work to enhance the manuscript, adding valuable experimental evidence to support several of their findings. We thank the authors for following the reviewer's comments and suggestions to improve the manuscript clarity and include much needed details on the experimental design. The resulting revised version has been much improved and addressed most of the comments and questions. Yet, one main question has not been addressed and in our opinion could impact the findings on the study.

Major comment

1) The authors have yet to demonstrate that the siRNA knock down of PRP4B, BTPF and BUD31 does not arrest the cell cycle, and that the differences in cell migration are not simply due to the fact the cells are growth arrested and thus are not moving. This would not impact the number of tracks detected and this would not be excluded from the analysis as stated by the authors in the rebuttal. This concern regarding confounding effects of cell proliferation or cell viability on the results presented here has been raised by all three reviewers.

Many splicing factors have been shown to affect cell cycle, and can lead to increase cell death, decreased cell proliferation, or delayed G1/M entry. For example, previous siRNA screens for regulators of the pro-apoptotic proteins BCL-X and MCL1 or FAS uncovered an enrichment in spliceosomal components and splicing factors at 72h of treatment (Moore et al. Cell 2010; Papasaikas et al. Mol Cell 2015). The new supplemental data suggests that there might be an effect on cell viability: in Figure S10A,C there are fewer cells visible on the slide after PRP4B siRNA KD at 20h vs. 0h.

Measuring the number of cell tracks during the 7h of the migration experiment as shown in Fig S3 is not a reliable measure of cell growth arrest. This could be easily addressed by performing cell cycle assays or cell proliferation (MTT) assays on cells +/- siRNA at different time points over the course of the experiment (from 0h to 59hrs).

To answer the reviewer's question, we have now performed siRNA knockdown of PRPF4B, BUD31 and BPTF in Hs578T and MDA-MB-231 cell lines followed by proliferation measurements using the Sulforhodamine B (SRB) assay 72 hours after transfection, which measure total cell protein and is linear with nuclear count (1-3). At this time point, cells were fixed for the PKT assay and live cell imaging was performed in the previous experiments in the manuscript. Confirming that PRPF4B, BUD31 and BPTF mainly affect cell migration, we only observed a small non-significant decrease in cell proliferation for PRPF4B and BUD31 in MDA-MB-231 cells (Suppl. Fig. 20B-C). No effect on proliferation was observed in Hs578T cells (Suppl. Fig. 20A). NHP2L1, a splicing factor known to affect proliferation from previous experiments in our group was used as a positive control. These results are discussed in the manuscript on page 12.

1. Vichai V, Kirtikara K. Sulforhodamine B colorimetric assay for cytotoxicity screening. *Nat Protoc.* 2006;1(3):1112–1116.
2. Moerkens M, Zhang Y, Wester L, van de Water B, Meerman JH. Epidermal growth factor receptor signalling in human breast cancer cells operates parallel to estrogen receptor α signalling and results in tamoxifen insensitive proliferation. *BMC Cancer.* 2014 Apr 23;14:283.
3. Zhang Y, Wester L, He J, Geiger T, Moerkens M, Siddappa R, Helmijr JA, Timmermans MM, Look MP, van Deurzen CHM, Martens JWM, Pont C, de Graauw M, Danen EHJ, Berns EMJJ, Meerman JHN, Jansen MPH, van de Water B. IGF1R signaling drives antiestrogen resistance through PAK2/PIX activation in luminal breast cancer. *Oncogene.* 2018 Apr;37(14):1869-1884.

Supplemental Figure 20. Effect of PRPF4B, BUD31 and BPTF knockdown on cell proliferation. Proliferation of candidate knockdown compared to siKinasePool knockdown in (A) Hs578T and (B) MDA-MB-231 cells 72 hours after knockdown using the SRB assay. Mean + stdev of three biological replicates. Significance was determined using one-way ANOVA, using correction for multiple hypothesis testing. ** $p < 0.01$, *** $p < 0.001$. (C) Representative 96-well images of nuclei staining in Hs578T and MDA-MB-231 cells with candidate or control knockdown.

Minor comments

1) Please state more explicitly how the variables can affect the experiments in the result section (Page 5): (i) latex beads need to be phagocytosed by cells to score migration, thus a defect in phagocytosis would affect the results of the assay; (ii) difference in cell size would affect the distance migrated.

We thank the reviewer for these suggestions and implemented these variables now also in the results section on page 5.

2) Please define what do you mean by “migratory phenotypes were manually classified” (page 5) – what is a manual classification?

This means that we manually curated many of the tracks and based on this set the gating for the different migratory phenotypes. The migratory phenotypes based on only one parameter such as the small phenotype, could be defined by using one Z-score cutoff (in case of the small phenotype this is the net area). However, some genes clearly affected migration but showed a more complicated phenotype that could not be captured by only one parameter. For example, long smooth tracks are characterized by an increased axial ratio, increased major axis and low roughness score and would not be selected by only investigating track area. To capture the complete migratory landscape, we

decided to assign tracks to specific phenotypes by using Z-score cutoffs of a combination of parameters that were established by manual curation of many different track phenotypes. The cutoff values for the different phenotypes have been described in the methods section on page 22. We also explained this method in more detail in the results section on page 5/6.

3) Please state more explicitly the differences in the migratory assays in the results section (page 8): (i) difference between boyden chamber assays vs. fibronectin-coated plates; (ii) differences in 12h vs 7h migration; (iii) what is the time course of the siRNA treatment in the boyden chamber assay?

Below we answer the reviewer's questions point-by-point. i) We performed all other migration assays on fibronectin coated plates, while the Boyden Chamber assay was performed on non-coated polystyrene membrane which indeed might explain the difference in migratory behavior observed. ii) Also, the differences in duration of the assays might change the outcome. We used a 7h PKT assay, 12h live cell migration assay, 6-20h scratch assay (depending on the cell line) and 22h Boyden Chamber assay. Although these time lines were optimized and carefully selected, this might influence the observed effects. iii) As described in the Supplemental Methods section, 65 hours after transfection, the cells were plated in the Boyden Chambers followed by 22h incubation. Next, cells that passed through the membrane were stained with a fluorescent dye which was used for quantification of the directed cell migration. According to the reviewer's suggestion, we included the differences between the migratory assays in the results section on page 8.

4) Please state more explicitly the differences between the Van der Weyden mouse in vivo screen and the current screen including the presence/absence of immune components and microenvironment (page 17).

We thank the reviewer for this suggestion. This has now been implemented in the discussion on page 17.

5) Please provide a control non targeting guide for the CRISPR experiment.

As suggested by the reviewer, we included a control non-targeting guide in our CRISPR experiments in Supplemental Figure 9.

Supplemental Figure 9. Candidate CRISPR-Cas9 knockout and effect on live cell migration. (A) PRPF4B knockout efficiency 48 hours after Cas9 induction. (B) Migration speed of MDA-MB-231 ind-Cas9 72 hours after doxycycline exposure. Experiment was performed in biological triplicates except for sgCtrl (performed in duplicate), significance was calculated using student's t-test. * p < 0.05, ** p < 0.01, *** p < 0.001

6) The authors have performed significant work to analyze patient's data with regards to clinical phenotypes and genomic profiles. The figures, analysis and accompanying text could be improved to provide more clarity. Fig S15 and S16 are difficult to read, we suggest presenting the data as stacked columns representing for example % mutation in tumors with mets at diagnosis, and no mets at diagnosis. In addition a comparative analysis of selected non target genes would be useful. Please state more explicitly the limitations of the analysis including the small number of samples in the result section.

As suggested by the reviewer, we changed the layout of Supplemental Figures 15 and 16 and created stacked bar graphs. Moreover, we also added a comparative analysis of a similar number of genes that were included in our primary screen, but not selected as target genes (Suppl. Fig. 15C and Suppl. Fig. 16C). Next, we also discussed the limitations of the small number of samples in the results section on page 11.

Supplemental Figure 15. Candidate gene alterations in primary tumors with and without metastasis at diagnosis. (A) Log₂ RNA expression levels of candidates in primary tumors with (orange, n=22) or without (blue, n=906) distant metastases at diagnosis. (B) Candidate gene mutation rate (i), amplification rate (ii) and deletion rate (iii) in primary tumors with (orange, n=22) or without (blue, n=906) metastases at diagnosis. (C) Amplification of randomly selected non-hits in primary tumors with and without metastasis at diagnosis. Significance was calculated using the Fisher's exact test using the Benjamini-Hochberg method to correct for multiple testing. * p < 0.05, ** p < 0.01, *** p < 0.001.

Supplemental Figure 16. Candidate gene alterations in ERpos and TNBC primary breast tumors. (A) Log₂ RNA expression levels of candidates in ERpos and TNBC primary tumors. (B) Candidate gene mutation rate (i), amplification rate (ii) and deletion rate (iii) in ERpos and TNBC primary tumors. (C) Amplification of randomly selected non-hits in ERpos and TNBC primary tumors. Significance was calculated using the Fisher's exact test using the Benjamini-Hochberg method to correct for multiple testing. * $p < 0.05$, ** $p < 0.01$, *** $p < 0.001$

7) Please add in the result section a brief description of the splicing analysis and cut-off threshold (page 12) and explain Fig S21 – in particular how the red dots (significant events) are selected. One possible explanation for the following statement “We were unable to detect any changes in exon inclusion or 3’ or 5’ alternative splice site usage after knockdown of either PRPF4B or BUD31” is an insufficient read depth (20M vs 100M recommended for splicing analysis), as well as distinct analysis pipeline (DexSeq for intron retention and rMATS for exon inclusion) – this should be explicitly stated in the result section so that the reader is not left with the impression that PRPF4B, BUD31 and BTF only regulate intron retention. Please note that the commonly used cutoff for rMATS is FDR<0.05 and that utilizing the same p-value cutoffs for DexSeq and rMATS is not appropriate as these methods are very different and their p-values are not directly comparable, and likely explains why no exon inclusion events are detected. Please provide results of the splicing analysis as a supplemental file listing all the splicing events types.

The red dots are selected by an absolute intron inclusion difference of at least 10% and adjusted P-value smaller than 0.01. This was already described in the Methods section of the manuscript but is now also included in the results section on page 12. Regarding the alternative splicing events analyzed with the rMATS pipeline, we could indeed identify significant events using less strict cutoffs (FDR < 0.05 and inclusion difference > 0.1 or < -0.1). Still the number of genes affected for these events is very low compared to the intron inclusion events and also the overlap between the different cell lines is limited (Suppl. Fig. 24), indeed likely due to insufficient sequencing depth. These results have been discussed in greater detail in the results section on page 13. The significant alternative splicing events are provided in Supplemental Table 11.

Supplemental Figure 24. Overlap of genes alternatively spliced between different cell lines. A3SS = alternative 3’ splice site usage. A5SS = alternative 5’ splice site usage. MXE = mutually exclusive exon. SE = skipped exon.

8) Please make sure that the legends of all the figures containing required details of the experiment and analysis, including what is plotted in the error bars, number of samples, statistical analysis etc.

We thank the reviewer for this notification. We again looked at all the figure legends and added information when necessary.

REVIEWERS' COMMENTS:

Reviewer #2 (Remarks to the Author):

This is the third time the manuscript has been on my desk. In the latest incarnation, the authors have dutifully and thoughtfully answered all of the questions which I raised previously and also improved the Figures (particularly Fig 8) where suggested. At this point, I have no other major concerns which would hold the paper back from being published. One minor concern, which is more nebulous and therefore more difficult to address, is that in the several months over which the revisions took place, the lead author has been switched (for whatever reason) from Fokkelman to Koedoot. This in itself is not a problem, however, some of the flow, seamless storytelling, and consistency of style which the first two versions had has been lost and the paper (as a result) reads a bit more unevenly at this point. All the necessary data is there and salient points have been addressed in the text but I express a bit of remorse that the singularity of the writing has been slightly compromised. Perhaps no one else but me will notice it, as I have had the vantage point of seeing all three manuscripts. Overall, I am satisfied with the findings as they relate to newly discovered mechanisms of breast cancer metastasis, which is currently being examined in the field from all angles, i.e., genetic, epigenetic and within the context of the microenvironment's influence.

Reviewer #3 (Remarks to the Author):

We thank the authors for addressing the comments. The resulting revised version has been much improved. Congratulations on a great work.

REVIEWERS' COMMENTS:

Reviewer #2 (Remarks to the Author):

This is the third time the manuscript has been on my desk. In the latest incarnation, the authors have dutifully and thoughtfully answered all of the questions which I raised previously and also improved the Figures (particularly Fig 8) where suggested. At this point, I have no other major concerns which would hold the paper back from being published. One minor concern, which is more nebulous and therefore more difficult to address, is that in the several months over which the revisions took place, the lead author has been switched (for whatever reason) from Fokkelman to Koedoot. This in itself is not a problem, however, some of the flow, seamless storytelling, and consistency of style which the first two versions had has been lost and the paper (as a result) reads a bit more unevenly at this point. All the necessary data is there and salient points have been addressed in the text but I express a bit of remorse that the singularity of the writing has been slightly compromised.

Perhaps no one else but me will notice it, as I have had the vantage point of seeing all three manuscripts. Overall, I am satisfied with the findings as they relate to newly discovered mechanisms of breast cancer metastasis, which is currently being examined in the field from all angles, i.e., genetic, epigenetic and within the context of the microenvironment's influence.

We thank the reviewer for the comments and suggestions. We again carefully went through the manuscript and improved the flow of the manuscript.

Reviewer #3 (Remarks to the Author):

We thank the authors for addressing the comments. The resulting revised version has been much improved. Congratulations on a great work.

We thank the reviewer for the previous comments and suggestions.